# Seasonal Variability of Wake Impacts on U.S. Mid-Atlantic Offshore Wind Plant Power Production

David Rosencrans[1,2], Julie K. Lundquist[1,2,3], Mike Optis[2,4], Alex Rybchuk[2], Nicola Bodini[2], and Michael Rossol[2]

[1]Department of Atmospheric and Oceanic Sciences, University of Colorado, Boulder, 80303, USA
[2]National Renewable Energy Laboratory, Golden, 80401, USA
[3]Renewable and Sustainable Energy Institute, Boulder, 80303, USA
[4]Veer Renewables, Courtenay, V9N 9B4, Canada

*Correspondence to*: David Rosencrans (David.Rosencrans@Colorado.edu)

**Abstract**

The mid-Atlantic will experience rapid wind plant development due to its promising wind resource located near large population centers. Wind turbines and wind plants create wakes, or regions of reduced wind speed, that may negatively affect downwind turbines and plants. We evaluate wake variability and annual energy production with the first year-long modeling assessment using the Weather Research and Forecasting model, deploying 12 MW turbines across the domain at a density of 3.14 MW km$^{-2}$, matching the planned density of 3 MW km$^{-2}$. Using a series of simulations with no wind plants, one wind plant, and complete build-out of lease areas, we calculate wake effects and distinguish the effect of wakes generated internally within one plant from those generated externally between plants. We also provide a first step towards uncertainty quantification by testing the amount of added turbulence kinetic energy (TKE) by 0% and 100%. We provide a sensitivity analysis by additionally comparing 25% and 50% for a short case-study period. The strongest wakes, propagating 55 km, occur in summertime stable stratification, just when New England's grid demand peaks in summer. The seasonal variability of wakes in this offshore region is much stronger than diurnal variability of wakes. Overall, year-long simulated wake impacts reduce power output by a range between 38.2% and 34.1% (for 0%–100% added TKE). Internal wakes cause greater year-long power losses, from 29.2% to 25.7%, compared to external wakes, from 14.7% to 13.4%. The overall impact is different from the linear sum of internal wakes and external wakes due to non-linear processes. Additional simulations quantify wake uncertainty by modifying the added amount of turbulent kinetic energy from wind turbines, introducing power output variability of 3.8%. Finally, we compare annual energy production to New England grid demand and find that the lease areas can supply 58.8% to 61.2% of annual load. We note how the results of this assessment are not intended to make nor are they suitable to make commercial judgements about specific wind projects.

## 1 Introduction

The U.S. offshore wind industry is flourishing, with a target capacity of 30 GW by 2030 (FACT SHEET: Biden Administration Jumpstarts Offshore Wind Energy Projects to Create Jobs, 2023). New England features the highest population density in the United States and commensurate utility usage, making offshore wind an attractive regional electricity source. Twenty-seven active lease areas now span the mid-Atlantic Outer Continental Shelf (OCS). The OCS features low turbulence (Bodini et al., 2019) and fast winds, with 100 m winds averaging 10 m s$^{-1}$ (Musial et al., 2016). Consequently, large wind plants will be constructed to harness the ample wind resource.

Meteorological conditions and construction challenges constrain siting options for large wind plants. Because the average wind direction is southwesterly (Bodini et al., 2019), a southwest-to-northeast wind plant orientation mitigates external waking from neighboring plants. Further, preserving efficient vessel transit, upholding common fishery practices, and prioritizing safe Coast Guard search-and-rescue operations necessitates 1x1 nm corridors (W.F. Baird & Associates, 2019). Considering these constraints, wind plants will be densely packed into clusters.

Densely packed clusters produce wakes that adversely affect downwind turbines (Nygaard, 2014; Platis et al., 2018; Lundquist et al., 2019; Schneemann et al., 2020). Wakes are plumes downwind of turbines with slower wind speeds and increased turbulence. Mid-Atlantic wakes induced by large wind plants could impose wind speed deficits up to 2 m s$^{-1}$ (Pryor et al., 2021; Golbazi et al., 2022). Wind speed deficits can be replenished by wake recovery in which turbulence entrains momentum from aloft into the waked zone (Stevens et al., 2016; Gupta and Baidya Roy, 2021). However, stably stratified conditions suppress mixing for wake recovery (Fitch et al., 2013; Vanderwende et al., 2016; Porté-Agel et al., 2020). Under certain conditions, mid-Atlantic wakes could propagate 100 km or more (Pryor et al., 2021; Golbazi et al., 2022; Stoelinga et al., 2022).

Wake characteristics have been evaluated using physics-based models of varying complexity. High-fidelity methods include computational fluid dynamics models solving Reynolds-averaged Navier-Stokes equations (Antonini et al., 2020), large-eddy simulations resolving the turbine rotor as an actuator disk (Mirocha et al., 2014; Aitken et al., 2014; Shapiro et al., 2019; Arthur et al., 2020), and mesoscale models parameterizing a hub-height momentum sink, sometimes including a turbulence source (Fitch et al., 2013; Volker et al., 2015; Archer et al., 2020; Gupta and Baidya Roy, 2021), as reviewed by Fischereit et al. (2022). Pryor et al. (2021) characterized mid-Atlantic wake impacts using mesoscale modeling of 55 simulation days. They examined modified wind plant layouts of 15 MW turbines under different flow scenarios, considering power densities between 2.1 and 4.34 MW km$^{-2}$. Stoelinga et al., (2022) estimated wake impacts using 15 MW turbines and 16 simulation days under typical southwesterly flow. Golbazi et al. (2022) considered summertime wakes with three scales of turbines to consider

surface impacts. Finally, Rybchuk et al. (2022) addressed the sensitivity to wake characteristics under idealized conditions by varying planetary boundary layer (PBL) schemes.

65

**Table 1. Summary of WRF simulations.**

| Simulation Type | Acronym | Turbine Type | Period | Added TKE Amount | # Turbines |
|---|---|---|---|---|---|
| No Wind Farms | NWF | N/A | 09/2019-09/2020 | N/A | 0 |
| One Wind Farm Only | ONE | 12 MW | 09/2019-09/2020 | 0% and 100% | 177 |
| Lease Areas | LA | 12 MW | 09/2019-09/2020 | 0% and 100% | 1,418 |
| Call Areas | CA | 12 MW | 09/2019-11/2019 07/2020-09/2020 | 100% | 3,219 |

In this work, we assess intra-plant and inter-plant wakes throughout the mid-Atlantic OCS using a year-long mesoscale modeling study. The results of this assessment are not intended to make nor are they suitable to make commercial judgements about specific wind projects. The simulations use the Weather Research and Forecasting (WRF) model Version 4.2.1 (Skamarock et al., 2019). One set of simulations runs with no wind farms (NWF) as a control, validated with lidar measurements, while the others use the Fitch wind farm parameterization (WFP) (Fitch et al., 2012 with updates described by Archer et al. 2020) to incorporate turbine effects. Our simulations incorporate 12 MW turbines and a power density of 3.14 MW km$^{-2}$. Simulations employ different wind plant layouts, including one representative lease area alone (ONE) within the Rhode Island/Massachusetts (RIMA) block, all lease areas (LA), and the lease areas plus the call areas (CA), to assess different waking scenarios (Table 1). WFP simulations run separately by added turbulent kinetic energy (TKE) amount, including 0% added TKE (TKE_0) and 100% added TKE (TKE_100) to quantify the full range of uncertainty. NWF, ONE, and LA simulations run from 01 September 2019 to 01 September 2020 to capture a full year with available lidar measurement data. Due to computational costs, CA simulations focus on the summertime stable period from 01 September to 31 October 2019 and 01 July to 31 August 2020 (Table 1). This time period highlights wake impacts during months with presumed frequent stable stratification and high electricity demands (Livingston and Lundquist, 2020) as a worst-case scenario.

The remainder of this article is structured as follows: Section 2 introduces the model setup and configuration, model validation, and the analysis methods. Section 3 discusses variability in stratification, wakes, and power production. Section 4 concludes the work and offers recommendations for future work.

**2 Methods**

**WRF Modeling Setup**

We assess the effects of wakes and power production across the mid-Atlantic OCS using numerical weather prediction simulations with WRF Version 4.2.1 and the WFP (Fitch et al., 2012). Version 4.2.1 allows for modifying the amount of TKE produced by wind turbines and ensures turbulence advection (Archer et al., 2020). Two nested domains comprise 6 km and 2 km horizontal resolutions (Pronk et al., 2022; Xia et al., 2022; Bodini et al., 2023; Redfern et al., 2023), respectively, and the inner nest begins 20 grid cells into the parent domain (Figure 1). This same domain and period of study have been used to

explore interactions between power production and sea breezes (Xia et al., 2022). Fine vertical resolution (10 m) near the surface stretches aloft, with 17 levels within the lowest 200 m as recommended by Tomaszewski and Lundquist, (2020). We choose an 18 s time step in the outer domain, 54 vertical levels, a 5,000-Pa top, simple diffusion, and damping 6,000 m below the model top to prevent gravity wave reflection. Hourly 30 km initial and boundary conditions are provided by the European Centre for Medium-Range Forecasts (ECMWF) fifth-generation reanalysis (ERA5) data set (Hersbach et al., 2020). Sea

surface temperature is provided by the UK Met Office Operational Sea Surface Temperature and Sea Ice Analysis (OSTIA) data set (Donlon et al., 2012). We choose the Noah Land Surface Model (Niu et al., 2011), the Mellor-Yamada Nakanishi and Nino Level 2.5 PBL and surface layer (Nakanishi and Niino, 2006), New Thompson microphysics (Thompson et al., 2008), and the Rapid Radiative Transfer Model longwave and shortwave radiative transfer (Iacono et al., 2008) schemes. The Kain–Fritsch cumulus scheme parameterizes cloud microphysics in the outer domain only (Kain, 2004).

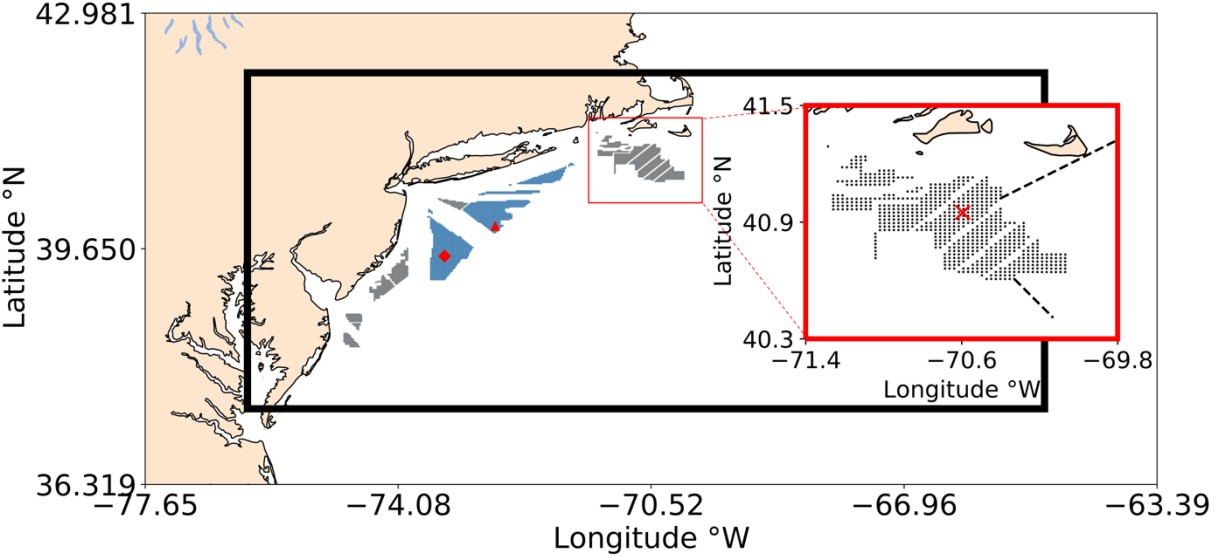

**Figure 1: Simulation Domain 1 includes the entire region, and simulation Domain 2 is outlined by the black rectangle. Each dot represents a wind turbine. Wind energy lease areas are shown in gray and call areas in blue. The red square zooms in on the**

**Rhode Island/Massachusetts block of lease areas. E05 (triangle) and E06 (diamond) floating lidars are shown in red. Atmospheric stratification is assessed at the red 'X'. Wake propagation distances are assessed along the black dashed lines.**

**Wind Turbine Layouts**

Wind turbines are sited within lease areas offshore of the U.S. East Coast (Figure 1) as defined by the Bureau of Ocean Energy Management (Renewable Energy GIS Data | Bureau of Ocean Energy Management, 2023). Following realistic deployment strategies, we site individual turbines 1 nm, or 8.6 rotor diameters, apart and an additional 0.5 nm from lease area boundaries (W.F. Baird & Associates, 2019; Beiter et al., 2020; Musial W., personal communication, Sept. 2020). This layout provides a power density of 3.14 MW km$^{-2}$. Lower power densities in U.S. waters reflect wake concerns in Europe and the need to increase turbine spacing for wake replenishment. Areas that had already been approved for development are denoted as the lease areas. Areas where competitive interest was yet to be determined are denoted as the call areas. Both lease areas and call areas are filled to spatial capacity with turbines (Figure 1), recognizing renewable energy targets (218th Legislature, 2018).

**Wind Turbine Characteristics**

For our simulations, we parameterize 12 MW turbines which are scaled by Beiter et al., (2020) from a 15 MW reference turbine with a 138 m hub height and 215 m rotor diameter. The power and thrust coefficient curves were held constant from the 15 MW machine. The rotor diameter was scaled to maintain a specific power of 332 W m$^{-2}$, which is the same as the reference 15 MW turbine. Then, the hub height was determined such that a 30 m gap was maintained between the lower bound of the rotor tip and the sea surface. No power is produced in region 1 of the power curve, from 0 m s$^{-1}$ to cut-in wind speed (3 m s$^{-1}$). Power production increases between cut-in wind speed and rated speed (11 m s$^{-1}$), region 2 of the power curve. Between rated and cut-out wind speed (30 m s$^{-1}$), region 3, an increase in wind speed no longer yields additional power production (Beiter et al., 2020) (Figure 2a).

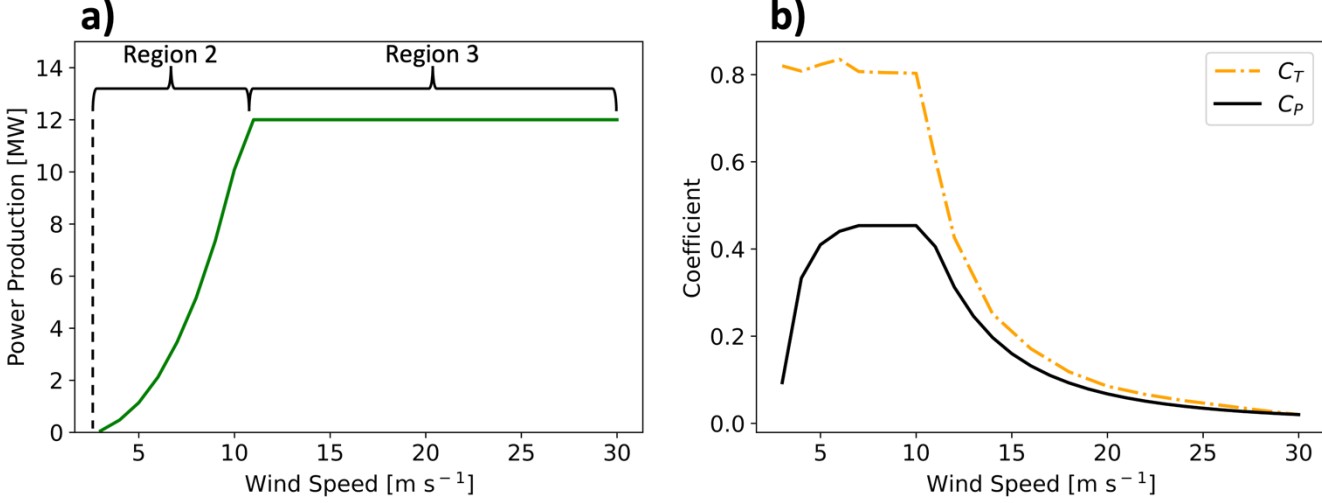

**Figure 2: Characteristics of the 12 MW scaled turbine used herein. (a) The power curve and (b) curves showing the thrust coefficients ($C_T$, dashed orange) and the power coefficients ($C_P$, solid black) with wind speed across the *x*-axis.**

**Wind Farm Parameterization**

We use the WFP (Fitch et al., 2012) to incorporate the effects of wind turbines on the 2 km grid. Horizontal wind speed reduction from turbine drag (Eq. 1), power production (Eq. 2), and turbulence generation (Fitch et al., 2012; Archer et al., 2020) (Eq. 3) are calculated in the WFP from:

$$\frac{\delta |\mathbf{V}|_{ijk}}{\delta t} = -\frac{N_{ij} C_T \big(|\mathbf{V}|_{ijk}\big) |\mathbf{V}|_{ijk}^2 A_{ijk}}{2(z_{k+1} - z_k)} \tag{1}$$

$$\frac{\delta P_{ijk}}{\delta t} = \frac{N_{ij} C_P \big(|\mathbf{V}|_{ijk}\big) |\mathbf{V}|_{ijk}^3 A_{ijk}}{2(z_{k+1} - z_k)} \tag{2}$$

$$\frac{\delta TKE_{ijk}}{\delta t} = \frac{N_{ij} C_{TKE} \big(|\mathbf{V}|_{ijk}\big) |\mathbf{V}|_{ijk}^3 A_{ijk}}{2(z_{k+1} - z_k)} \tag{3}$$

where i, j, and k represent Cartesian model coordinates, $C_T\big(|\mathbf{V}|_{ijk}\big)$ is the wind-speed-dependent thrust coefficient, $|\mathbf{V}|$ is the wind speed at turbine hub height, ρ is the air density, $A_{ijk}$ is the rotor swept area, $N_{ij}$ is the number density of turbines in grid

cell ij, $C_P\big(|\mathbf{V}|_{ijk}\big)$ is the wind-speed-dependent power coefficient, $z_k$ is the height of vertical model level k, and $C_{TKE}$ is the fraction of energy converted to TKE (Fitch et al., 2012). These values are calculated at each model level, as the use of a rotor-equivalent wind speed generally exerts a minor effect (Redfern et al., 2019).

The thrust and power coefficients ($C_T$ and $C_P$, respectively) vary with wind speed as defined by wind turbine manufacturers (Figure 2b). The thrust coefficient $C_T$ is the non-dimensionalized thrust force exerted by wind on the rotor-swept plane (Burton et al., 2011).

The power coefficient, $C_P$, governs the fraction of rotor kinetic energy converted into electrical power. This conversion is not perfectly efficient due to electrical and mechanical losses (Fitch et al., 2012; Archer et al., 2020). The leftover fraction of energy (Eq. 4) from the difference between $C_T$ and $C_p$ is transformed into turbulence, $C_{TKE}$.

$$C_{TKE} = C_T - C_P \tag{4}$$

Because electromechanical losses are not represented by the WFP, all leftover energy converts to TKE, so the TKE may be overestimated (Fitch et al., 2012; Archer et al., 2020). Some researchers suggest this TKE term is unnecessary (Volker et al., 2015), although comparisons to large-eddy simulations (Vanderwende et al., 2016) and observations (Siedersleben et al., 2020) suggest the turbine-produced TKE is critical to include. Any overestimation of TKE would enhance turbulent mixing, thereby exaggerating turbulent transport of momentum that causes wake recovery, and overestimating power production. Therefore, Archer et al. (2020) propose reducing $C_{TKE}$ to 25%. For these simulations, we bound this uncertainty by carrying out simulations with 100% and 0% added TKE (Figure A1). TKE advection is turned on.

**Observations**

We compared the NWF simulation to observations of offshore wind profiles. Two buoy-mounted meteorological ocean observing systems, denoted E05 and E06, are located within the Hudson North and Hudson South Call Areas of the New York Bight (Figure 3). Each buoy system samples line-of-sight boundary-layer wind speed and wind direction using the ZephIR ZX300M light detection and ranging (lidar) instrument. The lidars are mounted 2 m above the sea surface and take measurements at 20 m intervals up to 200 m, providing 10 min averages of wind speed and direction, which the New York State Energy Research and Development Authority (NYSERDA) has made publicly available (DNV, 2022). We use floating lidar data to validate simulations for 01 September 2019 to 01 September 2020.

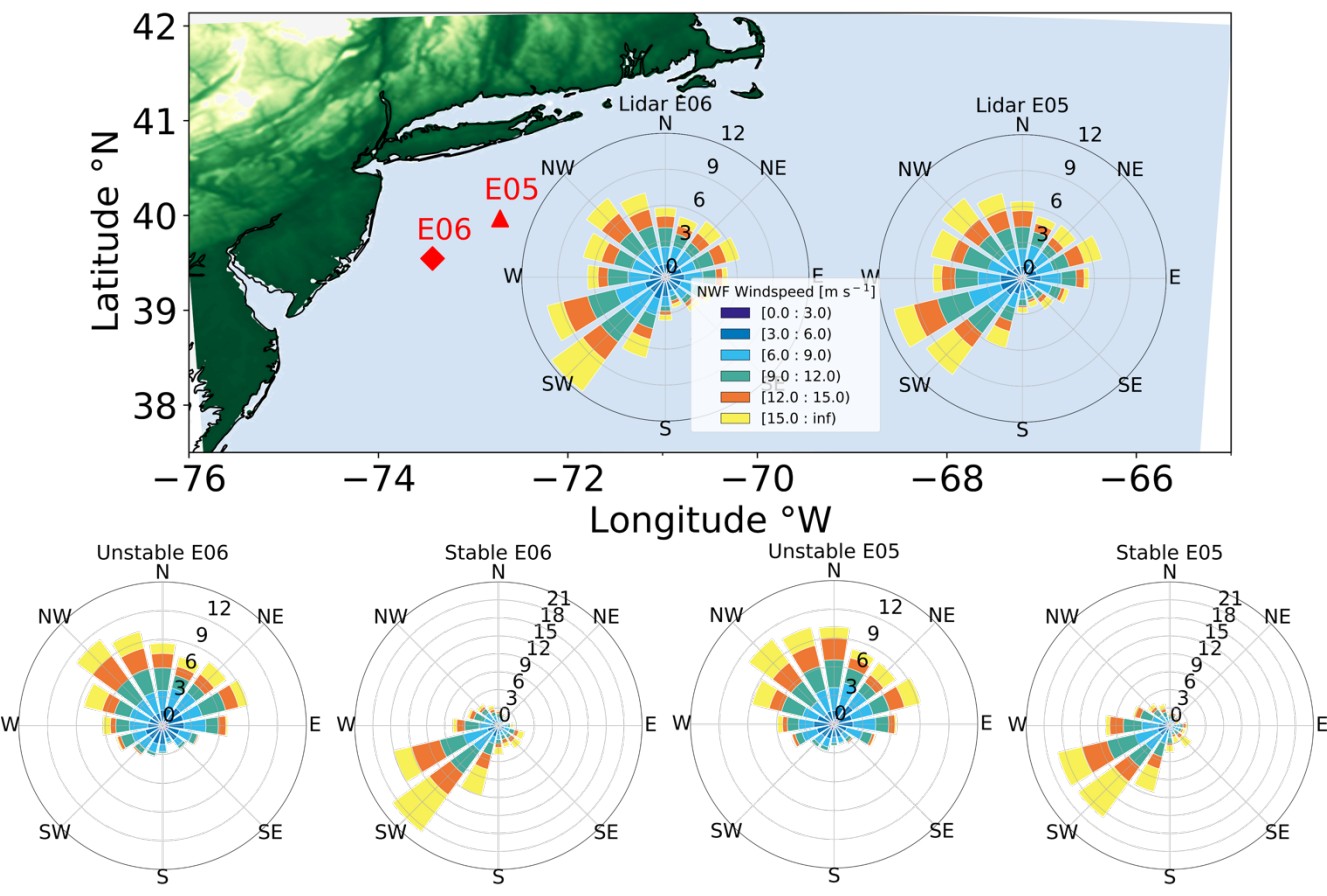

**Figure 3: Hub-height wind roses for the NYSERDA Hudson North (E05) and Hudson South (E06) floating lidars during the period 01 September 2019 to 01 September 2020. The location of E06 is shown as the red diamond and E05 as the red triangle. The bottom row shows wind roses segregated by atmospheric stratification.**

### Stability Classification

Different methods can be used to identify stratification, or atmospheric stability. Stable stratification can occur in coastal regions when warm air advects over a cooler sea surface, thereby suppressing buoyancy and turbulent mixing. Likewise, unstable stratification can occur when cool air advects over a warmer sea surface. Some observations suggest more frequent unstable stratification, based on the Obukhov length (Archer et al., 2016). The sign of the Obukhov length depends on the sign of heat flux and can be a useful metric for determining stability conditions. Other observations suggest that minimal

turbulence and strong veer can be characteristic of stable conditions (Bodini et al., 2019). Wind veer increases in stable

stratification as the influence of buoyant turbulence-induced friction decreases. Thus, winds turn to approach quasi-geostrophic flow at a quicker rate which can be further exaggerated by the presence of a low-level jet.

We calculate the Obukhov length (Monin and Obukhov, 1954) (L), representative of the height at which buoyant production of turbulence first dominates mechanical shear production of turbulence:

$$L = -\frac{u_*^3 \overline{\theta_v}}{\kappa g \overline{(w'\theta_v')}} \tag{5}$$

where $u_*$ is the friction velocity (UST from WRF output), $\theta_v$ is the virtual potential temperature, $\kappa$ is the von-Karman constant of 0.4, g is gravitational acceleration, and $\overline{w'\theta_v'}$ is the vertical turbulent heat flux (HFX from WRF output). Lengths between 0 m and −500 m are characterized as unstable stratification and lengths between 0 m and 500 m are categorized as stable

stratification (Muñoz-Esparza et al., 2012). Lengths approaching negative or positive infinity are neutral. Each timestamp from the NWF run is assigned a stability for the period 01 September 2019 to 01 September 2020 at a grid point centered on the RIMA block (Figure 1).

**Model Validation**

We validate the NWF model by comparing wind speed estimated by the turbine-free simulations with observations from E05 and E06 lidars. Model output is obtained from the grid cells containing the lidars in 20 m intervals from 60 m to 200 m following Pronk et al. (2022). Wind speeds and directions are compared using a suite of metrics recommended by Optis et al., (2020) for wind resource assessment including the correlation coefficient ($r$), centered root-mean-square error (cRMSE), and bias:

$$r = \frac{\sum_i^N (V_{W_i} - \overline{V_W})(V_{L_i} - \overline{V_L})}{N\sigma_W\sigma_L} \tag{6}$$

$$cRMSE = \sqrt{\frac{\sum_i^N \left((V_{W_i} - \overline{V_W}) - (V_{L_i} - \overline{V_L})\right)^2}{N}} \tag{7}$$

$$Bias = \frac{\sum_i^N (V_{W_i} - V_{L_i})}{N} \tag{8}$$

where V is the wind speed, N is the total number of values, $\sigma$ is the standard deviation, and subscripts *W* and *L* indicate "WRF" and "lidar", respectively. Earth Mover's Distance (EMD), or the Wasserstein metric, is calculated with a SciPy function

(Virtanen et al., 2020) as in other wind resource evaluations (Hahmann et al., 2020). Each of these metrics provides different insight into the performance of the model. For instance, the correlation coefficient illuminates how well the model captures the timing of weather systems and diurnal variability. EMD emphasizes the difference between distributions but not the timing. Bias captures the difference between measured and modeled values. Finally, cRMSE describes the random component of error.

The circularity of wind direction must be accounted for in statistical calculations. For example, computing the average between 359° and 1°, using a typical arithmetic mean, would result in 180°. However, the mean wind direction between those two values should be 360°. The SciPy (Virtanen et al., 2020) and Astropy (Price-Whelan et al., 2022) Python packages offer convenient functions which allow the user to calculate statistics for a circular variable by passing in the lower and upper bounds, in this case 0° and 360°. We calculate the mean and standard deviation of wind direction using the SciPy "circmean"

and "circstd" functions, respectively, and the correlation coefficient using the Astropy "circcorrcoef" function. The cRMSE for wind direction is then calculated following:

$$cRMSE = \sqrt{circmean\left(180° - \left|\left|(D_{W_i} - \overline{D_W}) - (D_{L_i} - \overline{D_L})\right| - 180°\right|\right)^2} \qquad (9)$$

where $D$ is wind direction and $\overline{D}$ is the circular mean of wind direction. Bias is calculated similarly to Eq. 8, except that differences between NWF and lidar values that are less than $-180°$ have 360° added and differences greater than 180° have

225 360° subtracted:

$$x = \begin{cases} x + 360° & \text{for } x < -180° \\ x - 360° & \text{for } x > 180° \end{cases} \qquad (10)$$

where $x$ is the $(D_{WRF_i} - D_{lidar_i})$ difference.

**Table 2. Percentage of data removed at 140 m due to NaN values.**

|        | Unstable | Stable | Neutral |
|--------|----------|--------|---------|
| **E05** | 1.35%   | 6.44%  | 0.33%   |
| **E06** | 3.64%   | 9.48%  | 0.62%   |

Time stamps in which the lidar returns NaN values are removed from WRF data sets during comparison (Table 2). Doing so removes 8.1% of wind speed data at 140 m at E05, made up by 1.22%, 5.76%, and 1.13% in unstable, stable, and neutral stratification, respectively. Similarly, 13.7% of wind speed data is removed at E06 and is made up by 3.20%, 9.38%, and

235 1.15% in unstable, stable, and neutral stratification, respectively. An $r^2$ value of one indicates a perfect correlation between NWF and lidar values. A value of 0 for cRMSE indicates that all values, with model bias removed, lie on the 1:1 regression line. A cRMSE value greater than 0 indicates the distance of residual points from the regression line. Negative biases indicate an underestimation from WRF while positive biases indicate overestimation. A value of 0 for EMD indicates that probability density functions from each data source are equivalent. A positive EMD indicates that the NWF wind speed distribution must

shift towards lower values to match the lidar distribution.

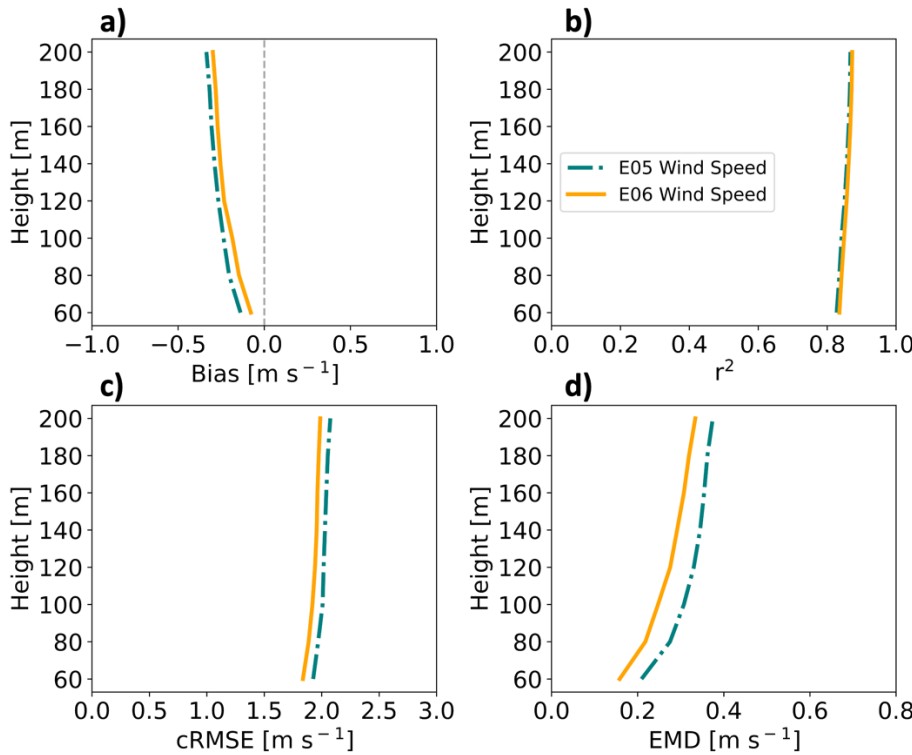

**Figure 4: Vertical profiles for wind speed comparative metrics at the E05 (teal) and E06 (orange) lidars from 01 September 2019 to 01 September 2020. Shown are (a) bias, (b) correlation, (c) cRMSE, and (d) EMD.**

NWF wind speed profiles are compared with lidar observations for the period 01 September 2019 to 01 September 2020 to assess model skill (Figure 4). Note that Pronk et al. (2022) provide validation metrics against the E05 lidar profile during the same period of study and find similar results. Negative biases (Eq. 8) increase in magnitude with height between 0 m s$^{-1}$ and $-0.5$ m s$^{-1}$ (Figure 4a), showing the model underestimates the wind speed. Strengths of variation (Eq. 6) among WRF output

and the lidars range between 0.82 and 0.86 (Figure 4b). Centered RMSE (Eq. 7) increases with height around 2 m s$^{-1}$ (Figure 4c). Finally, EMD values originate around 0.2 m s$^{-1}$ at 60 m and increase with height (Figure 4d). Comparing lidars E05 and E06, WRF performs better at E06 with a smaller bias by 0.04 m s$^{-1}$, lower cRMSE by 0.08 m s$^{-1}$, better correlation by 0.003, and smaller EMD by 0.05 m s$^{-1}$.

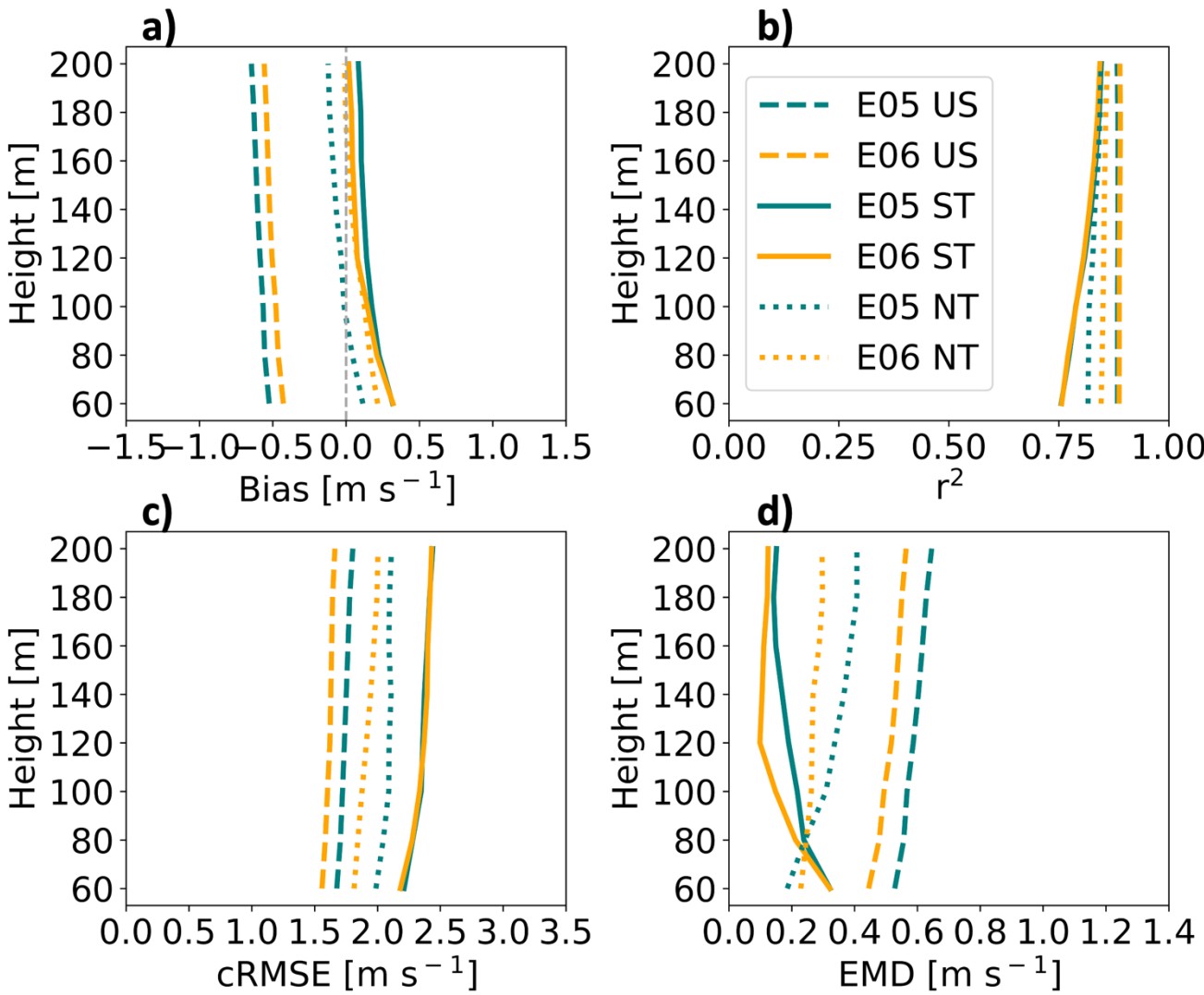

**Figure 5: Vertical profiles for wind speed comparative metrics at the E05 (teal) and E06 (orange) lidar locations subset by stratification (US = unstable, ST = stable, NT = neutral) from 01 September 2019 to 01 September 2020. Shown are (a) bias, (b) correlation, (c) cRMSE, and (d) EMD.**

We further assess the NWF performance, partitioned by stability conditions. In unstable stratification, WRF wind speeds have a negative bias that gradually increases in magnitude with height from $-0.5$ m s$^{-1}$ at 60 m (Figure 5a). In stable conditions, WRF overestimates wind speeds by roughly 0.4 m s$^{-1}$ at 60 m with biases approaching 0.0 m s$^{-1}$ further aloft (Figure 5a). In neutral conditions, WRF overestimates wind speeds by up to 0.3 m s$^{-1}$ near the surface and underestimates wind speeds further

aloft. Comparing between mean E05 and E06 profiles, WRF performs better at the E06 lidar location by 0.08 m s$^{-1}$ in unstable conditions, 0.04 m s$^{-1}$ in stable conditions, and 0.1 m s$^{-1}$ in neutral conditions.

NWF and lidar wind speeds correlate well. Correlation remains largest in unstable conditions for all heights (Figure 5b). The worst strength of relationship occurs in stable stratification although there is improvement aloft, and by 160 m, correlation between stable and neutral conditions is largely equivalent (Figure 5b). On average, WRF performance between lidar locations is the same in unstable and stable conditions and better at E06 by 0.02 in neutral conditions.

Centered RMSE profiles change with stratification. In unstable conditions, cRMSE increases somewhat with height originating from greater than 1.5 m s$^{-1}$ at 60 m (Figure 5c). In stable stratification, the cRMSE profile begins at roughly 2.3 m s$^{-1}$ at 60 m and increases with height. In neutral conditions, cRMSE increases with height from around 2 m s$^{-1}$. As before, WRF performs better at E06. On average, cRMSE is lower at E06 by 0.1 m s$^{-1}$ in unstable conditions, by a negligible amount in stable conditions, and by 0.1 m s$^{-1}$ in neutral conditions.

Earth Mover's Distance has more variability with height. It is largest in unstable stratification, increasing with height from roughly 0.5 m s$^{-1}$ at 60 m (Figure 5d). In stable conditions, EMD decreases with height and originates at around 0.35 m s$^{-1}$ at 60 m. In neutral stratification, EMD decreases with height from about 0.2 m s$^{-1}$. On average, WRF performs better at E06 by 0.07 m s$^{-1}$ in unstable conditions, by 0.04 m s$^{-1}$ in stable conditions, and by 0.06 m s$^{-1}$ in neutral conditions.

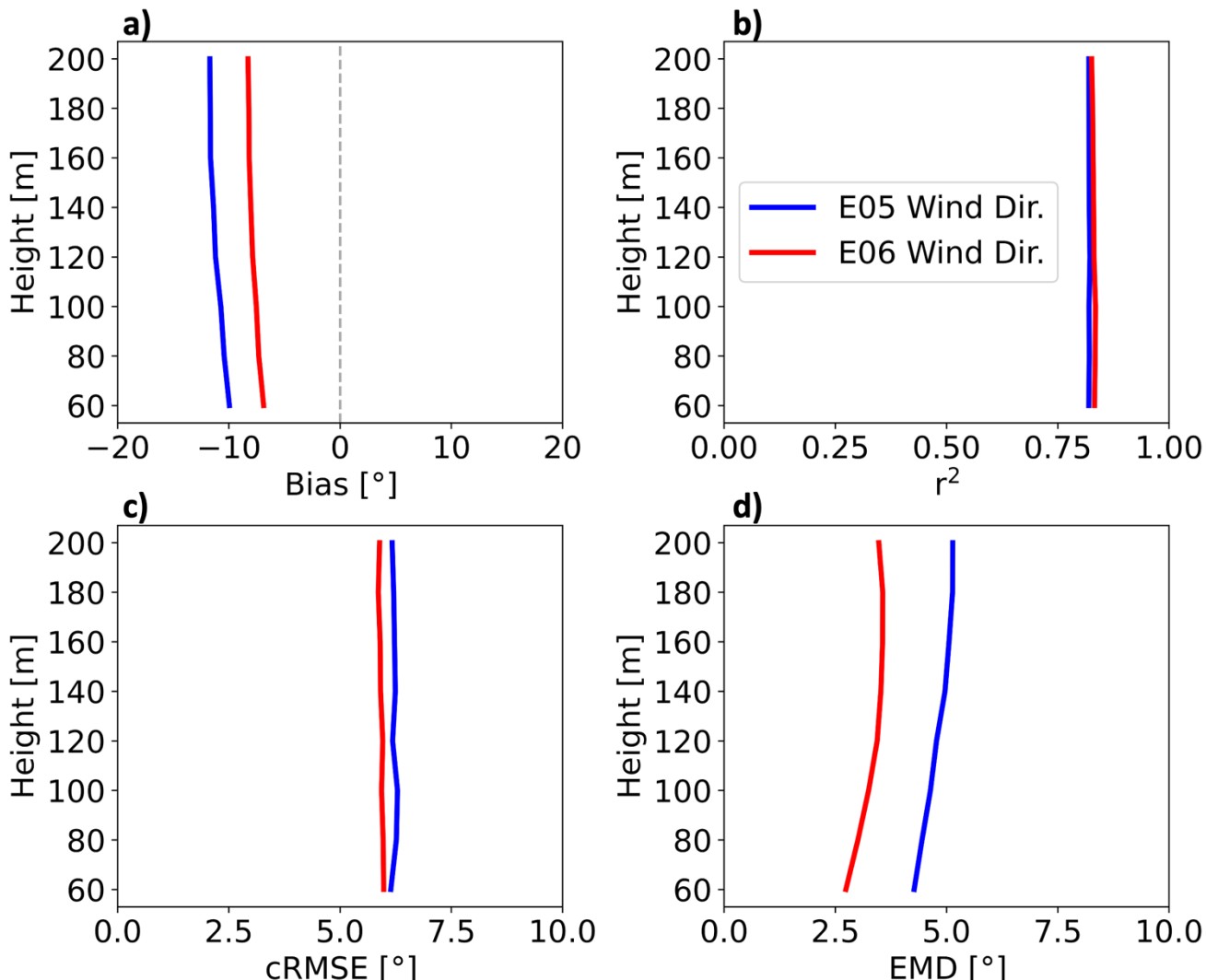

**Figure 6. Vertical profiles for wind direction comparative metrics at the E05 (blue) and E06 (red) lidar locations from 01 September 2019 to 01 September 2020.  Shown are (a) bias, (b) correlation, (c) cRMSE, and (d) EMD.**

Next, we show metrics to compare WRF-output wind direction profiles with lidar measurements.  Bias is negative, or counterclockwise, at both E05 and E06 lidar locations.  NWF output resolves wind directions better at E06 with a mean bias of -7.8° with height as compared to -11.1° at E05 (Figure 6a).  Correlation coefficients at both locations are strong, at 0.83 and 0.82 for E06 and E05, respectively (Figure 6b).  Mean cRMSE (Eq. 9) is similar between lidar locations, at 5.9° and 6.2° for E06 and E05, respectively (Figure 6c).  Finally, EMD is lower at E06, increasing with height between with an average of 3.3° (Figure 6d).  EMD is larger at E05, increasing with height between with an average of 4.8° (Figure 6d).  Overall, WRF performs better at E06 with lower absolute bias by 3.3°, lower RMSE by 0.3°, higher correlation by 0.01, and lower EMD by 1.48°.

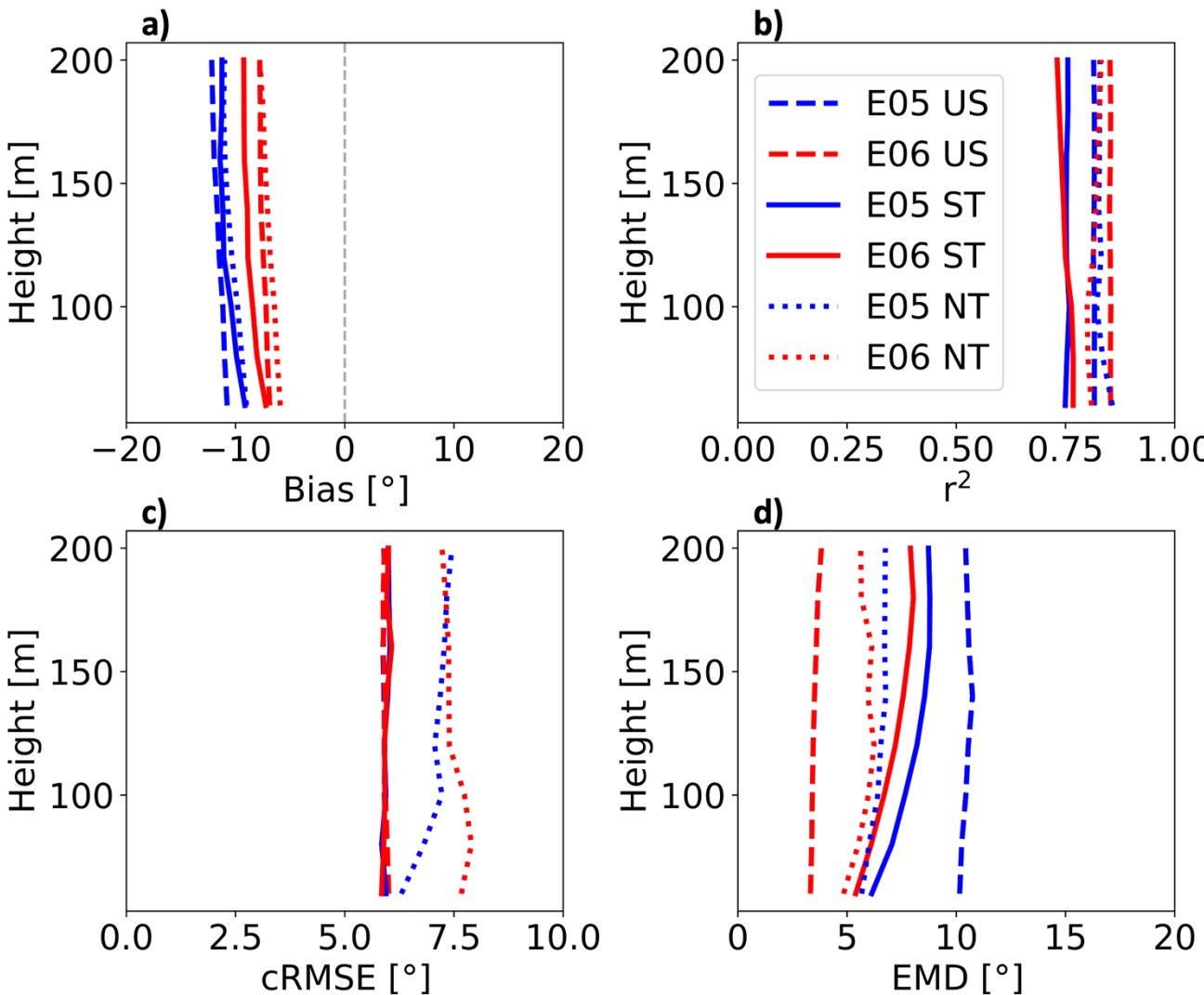

**Figure 7. Vertical profiles for wind direction comparative metrics at the E05 (blue) and E06 (red) lidar locations subset by stratification (US = unstable, ST = stable, NT = neutral) from 01 September 2019 to 01 September 2020. Shown are (a) bias, (b) correlation, (c) cRMSE, and (d) EMD.**

We use the same metrics to validate WRF against lidar-reported wind directions by stratification and begin with bias (Figure 7a). In unstable conditions, mean biases are –7.4° at E06 and –11.5° at E05. In stable stratification, bias profiles are more similar between lidar locations, reaching –8.6° at E06 and –10.7° at E05. Bias is smallest in neutral conditions at both locations, with mean values of –6.8° at E06 and –10.2° at E05. Overall, WRF performs better at E06 lidar location by 4.1° in unstable conditions, by 2.0° in stable conditions, and better at the E05 lidar location by 3.4° in neutral conditions.

The correlation between WRF-derived lidars-measured wind directions is strong in all stability conditions at both lidar locations (Figure 7b). The strength of relation in unstable conditions is 0.85 at E06 and 0.81 at E05. In stable conditions, the mean correlation is 0.75 at both E06 and E05. In neutral conditions, the strengths of relation are 0.81 at E06 and 0.83 at E05. Overall, WRF performs better at E06 by 0.03° in unstable conditions, 0.003° at E05 in stable conditions, and better at E05 by

0.01° in neutral conditions.

Profiles for cRMSE are similar in unstable and stable conditions with worse performance in neutral conditions (Figure 7c). In both unstable and stable conditions, mean cRMSE is 5.9° at both E05 and E06. In neutral conditions, mean cRMSE is 7.5° at E06 and 7.0° at E05. WRF performs the same at both lidar locations in unstable and stable conditions and is better at E05 by

315 0.4° in neutral conditions.

Large variability exists for EMD between lidar locations in WRF (Figure 7d). Unstable stratification features the largest spread between lidar locations, with EMD values of 3.5° at E06 and 10.4° at E05. In stable conditions, EMD is 7.0° at E06 and 7.9° at E05. In neutral stratification, mean EMD values are 5.7° at E06 and 6.4° at E05. On average, WRF performs best at the

320 E06 lidar location 6.9° in unstable conditions, 0.8° in stable conditions, and 0.7° in neutral conditions.

Wind speed timeseries are collected and averaged for the full year-long period from the grid cells housing lidars E05 and E06 in NWF and from the lidar measurements. The shear exponent is calculated as:

$$a = \frac{\log(V_2) - \log(V_1)}{\log(z_2) - \log(z_1)} \tag{11}$$

where $V_1$ and $V_2$ are the mean wind speeds at heights $z_1$ and $z_2$, respectively. We hold $V_1$ and $z_1$ constant at a reference height of 60 m and substitute $V_2$ and $z_2$ with values from 80 m to 200 m at 20-m intervals.

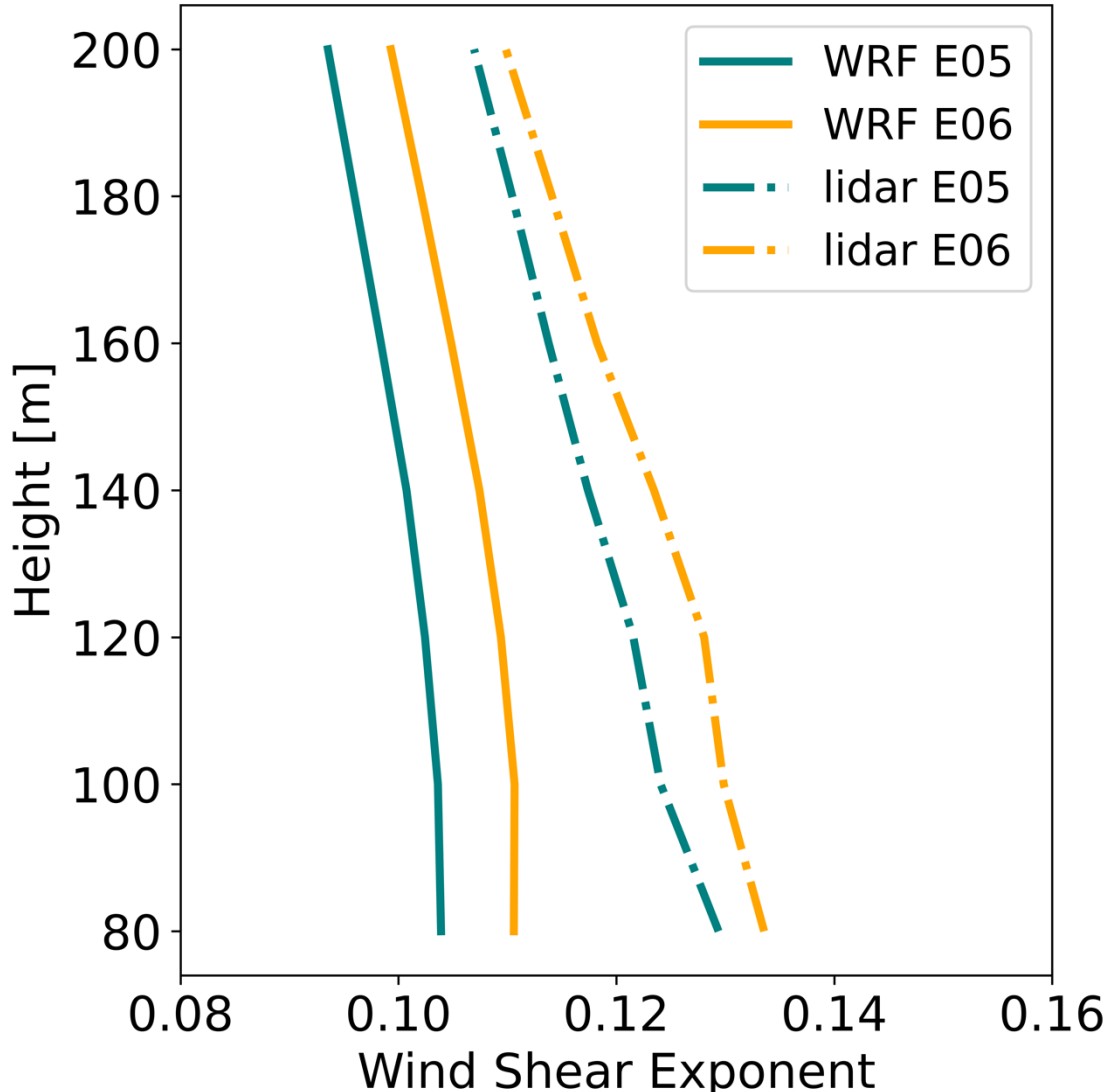

**Figure 8. Mean wind speed shear exponent by height from NWF (a) and from lidar measurements (b) from 01 September 2019 to 01 September 2020.**

Wind speed shear exponents (Eq. 11) differ between NWF and the lidar measurements. The average exponents from lidars E05 and E06 are 0.117 and 0.122, respectively, and are in good agreement with the annual average of 0.12 for both measured and modelled results in the mid-Atlantic (Viselli et al., 2018). The average exponents from WRF at grid cells housing E05

and E06 are 0.099 and 0.106, respectively. NWF-derived exponents correctly capture a decrease with height and lower coefficients at the E05 lidar. However, the exponents are smaller than those calculated from lidar measurements by -0.018

and -0.016 at E05 and E06, respectively.   Smaller exponents in NWF may result from overestimated mixing or misrepresentation of wave-induced roughness.

We calculate profiles of the Perkins Skill score ($PSS$) (Perkins et al., 2007) between NWF and lidar wind speeds. Wind speeds are considered at 20 m height intervals from 20 m to 200 m. Each wind speed timeseries is subset by all timestamps with unstable, stable, and neutral stratification, respectively. After subsetting, timestamps where lidar observations return NaN are removed from both lidar and NWF timeseries. At each height, the probability distribution functions of wind speeds are binned at 0.2 m s$^{-1}$ intervals and normalized such that the frequencies add to unity. The minimum frequency between modeled and 345   observed values for each bin is stored, and the resulting stored values are summed to calculate the score:

$$PSS = \sum_{i=1}^{n} \min\bigl(C_W(z), C_L(z)\bigr) \qquad (12)$$

where $n$ is the number of bins, $C$ is the count of normalized values in a bin, and $z$ is the height. A PSS of 1.0 suggests perfect overlap of the two distributions.

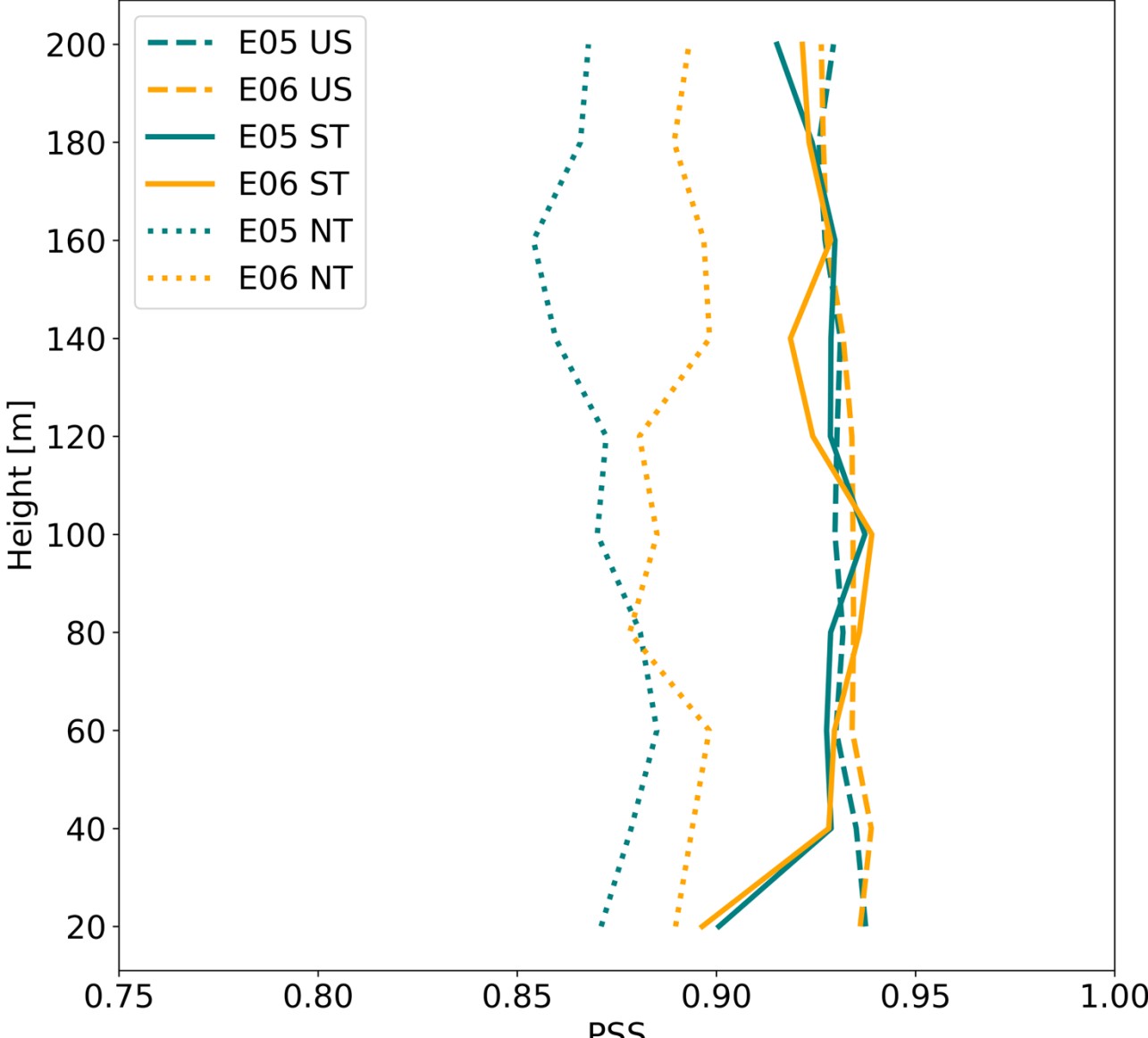

**Figure 9. Vertical profiles of the Perkins Skill Score by stratification. at the E05 (teal) and E06 (orange) lidars subset by stratification (US = unstable, ST = stable, NT = neutral).**

Profiles of PSS (Eq. 12) between NWF and lidar observations of wind speed vary by location and stratification. Performance
is generally best in unstable conditions at both E05 and E06 lidar locations with a mean value of 0.93. Performance is next best in stable conditions, starting around 0.90 at the surface and increasing to 0.93 at 120 m at E05. At E06 in stable conditions, PSS reaches a maximum value of 0.93 at 100 m. Neutral conditions exhibit worse PSS and larger spread by location. AT E05,

PSS minimizes at 0.85 at 160 m and maximizes around 0.88 at 60 m. At E06, PSS scores minimizes at 0.87 at 80 m and maximizes at 0.89 at 140 m.

### Wake Identification

The wake delineates the region downwind of turbines with a velocity deficit and turbulence enhancement. We identify the wind speed wake deficit by subtracting NWF wind speeds from WFP wind speeds at the hub height. Averaging across all times during the period 01 September 2019 to 01 September 2020 identifies the overall mean wake wind speed. Because
wakes typically propagate to the northeast during stable conditions (Figure 3), we calculate the propagation distance of wakes along a line extending northeast of the RIMA block (Figure 1) and report the distance along the line where wake wind speeds reach a threshold. In unstable conditions the prevailing wind direction is northwesterly (Figure 3), so we assess the wake propagation distance to the southeast instead. The threshold of $-0.5$ m s$^{-1}$ is chosen following Golbazi et al., (2022); Rybchuk et al., (2022). Finally, we define the areal extent of wakes as the area with a wind speed deficit less than $-0.5$ m s$^{-1}$.

### Grid Balancing

We compare model-output energy production to New England grid demand. Demand data are provided hourly (NEISO, 2023a). For comparison, we compute hourly averages of WFP power production from each set of simulations. We compare those averages to the national energy supply by acquiring the total from the U.S. Energy Information Administration (EIA,
2023).

### Power Variability

Assessing power variability is essential for addressing temporally changing grid demands. We assess the differences in electricity generation for each deployment scenario by collecting power output from grid cells containing wind turbines
separately from ONE, LA, and CA simulations. Power is summed across grid cells containing turbines and averaged at 1 day, 7 day, and 30 day intervals for comparison. We address seasonal and diurnal variability by further separating and averaging power production totals at each timestep into bins by month and hour of day. Power losses from the total, internal, and external wake effects are calculated from:

$$Loss_{tot} = 100\% - \left(\frac{P_{LA,CA}}{P_{NWF}}\right) \times 100\% \tag{13}$$

$$Loss_{int} = 100\% - \left(\frac{P_{ONE}}{P_{NWF}}\right) \times 100\% \tag{14}$$

$$Loss_{ext} = 100\% - \left(\frac{P_{LA,CA}}{P_{ONE}}\right) \times 100\% \tag{15}$$

$$Loss_{ext} = Loss_{tot} - Loss_{int} \tag{16}$$

where $P_{LA,CA}$ is the power production at ONE grid cells in the presence of wakes by either the LA or the CA, $P_{ONE}$ is the power production in the presence of internal wakes from ONE, and $P_{NWF}$ is the power production from coupling hub-height wind speeds to the power curve. These methods are performed separately by added TKE amount. We note that the upwind conditions change in a LA or CA scenario, due to external wakes, which can modify the internal losses in the numerator of Eq. 15. Thus, we provide an alternative method for calculating the external power losses as the difference between the total losses and the internal losses in Eq. 16.

Cluster-induced power deficits at ONE occur due to external wakes from the upwind lease and call areas. Power output from ONE, LA, and CA simulations are averaged in hourly windows at grid cells containing ONE turbines to reduce the effects of numerical noise (Appendix F). The resulting power averages from LA and CA simulations are divided by the averages from ONE at each time stamp. The hour of day and month of year categorize each time stamp and percentages are placed into bins accordingly. Within each bin the percentages are averaged. Only power production totals greater than 9.9 MW are considered when calculating power losses. This threshold represents the power production total when all turbines within ONE begin operating at the cut-in wind speed. For reference, the total power production for ONE at rated power is 2,124 MW. This method is repeated separately for TKE_0 and TKE_100 runs.

Individual wind turbines generate internal wakes within the ONE plant that adversely affect power production. To quantify internal wake effects at ONE, we collect NWF wind speeds at the hub height in each cell containing ONE turbines. Wind speeds are convolved with the power curve and scaled by the number of turbines per cell at 0.01 m s$^{-1}$ intervals. This method returns the amount of power that ONE would produce in the absence of wakes. Hourly power averages are obtained from both NWF and ONE runs and considered only if power production exceeds 9.9 MW. ONE power totals are divided by the NWF power estimations from the power curve. Again, each time stamp is categorized by hour of day and month of year, and percentages are binned for averaging. These steps are repeated for both TKE_0 and TKE_100 runs.

**3 Results**

**Year-Round NWF Stratification**

The predominance of NWF stability conditions changes throughout the year (Figure 10, Figure 11) as assessed using the Obukhov length (Eq. 5) centered on the RIMA block.

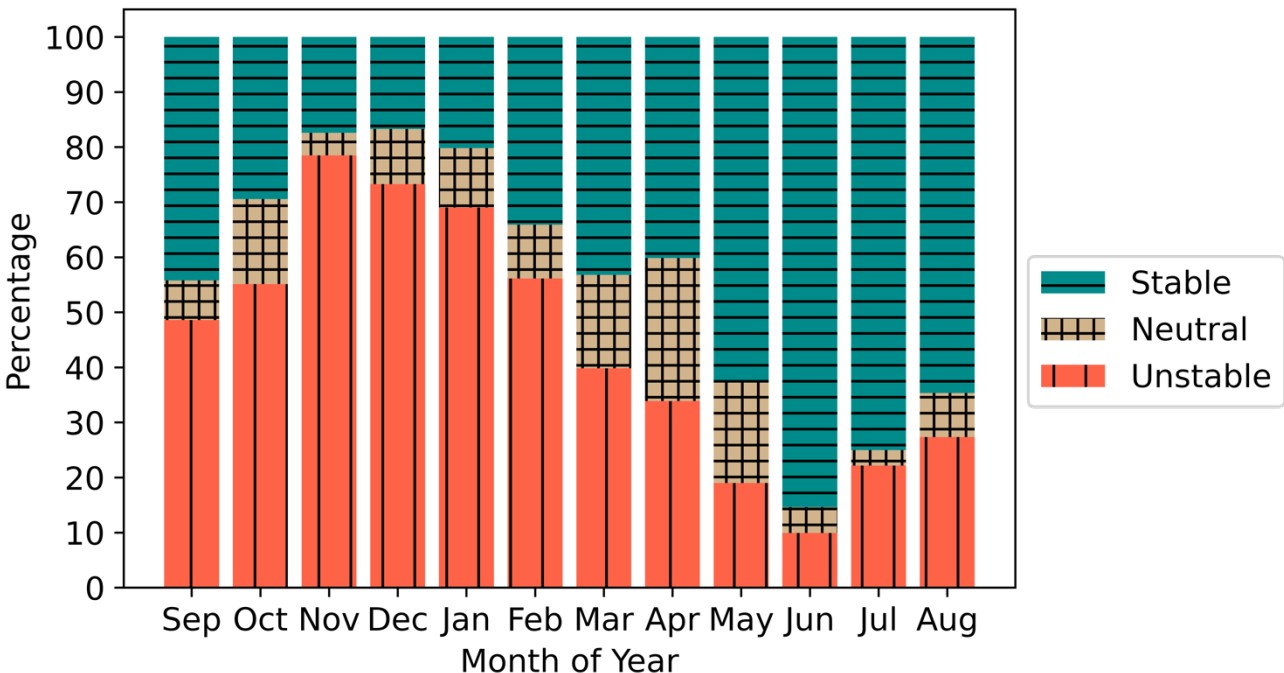

**Figure 10: Stability classification using the Obukhov length for the period 01 September 2019 to 01 September 2020 at the RIMA**
**block from NWF. Tan crosshatch represents neutral stratification, teal horizontal lines are stable stratification, and red vertical lines are unstable stratification.**

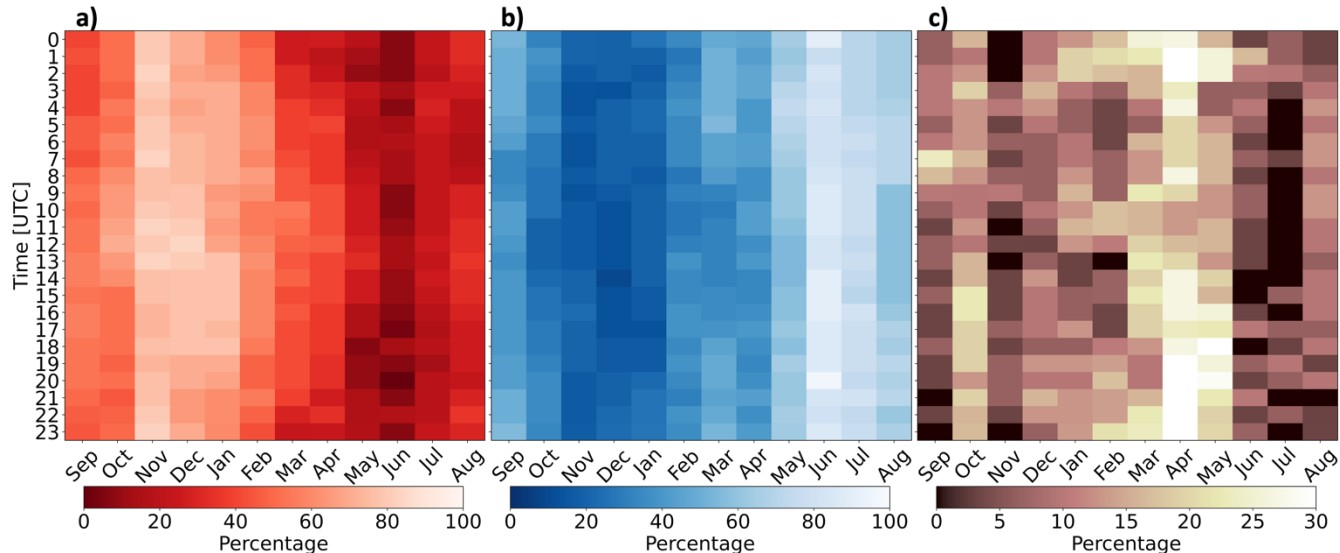

**Figure 11: Percentages of occurrence for (a) stable stratification, (b) unstable stratification, and (c) neutral stratification from 01 September 2019 to 01 September 2020.**

The winter features predominant unstable stratification whereas the summer features frequent stable stratification (Bodini et al., 2019; Optis et al., 2020) (Figure 10, Figure 11). The strong stability in summer is caused by nearby surface-heated air advecting over the colder OCS. These dynamics reverse during winter when cold air from land advects over warmer water. Overall, stratification is most frequently unstable during November and stable during June. April features the greatest percentage of neutral conditions as the springtime transition from cooler to warmer air reduces the air-sea temperature gradient. The same pattern occurs elsewhere throughout the OCS because diurnal variability in stratification is weaker than the seasonal cycle (Figure 11). The mean unstable, stable, and neutral percentages of occurrence at the RIMA block are 44.3%, 44.4%, and 11.2%, respectively, for the period 01 September 2019 to 01 September 2020. Stability calculations from the model grid cells that house lidars E05 and E06 reveal similar results (Figure B1). However, $L$ may not always represent conditions aloft (Figure C1).

**Wake Variability**

Here, we categorize wakes by the maximum wind speed deficit in space, the spatial extent, and the downwind propagation distance. While wakes remain relatively unchanged between TKE_0 and TKE_100, they drastically vary by stratification. The maximum average wake wind speed deficit occurs within the wind plant areas and intensifies from −1.5 m s$^{-1}$ to −2.8 m s$^{-1}$, moving from unstable to stable conditions for TKE_100 (Figure 12a,c). Normalized with mean NWF hub-height wind

speeds of 9.2 m s$^{-1}$ (unstable) and 11.2 m s$^{-1}$ (stable), the corresponding mean wind speed deficits are 16% and 25%. Similarly, the maximum average wind speed deficit intensifies from −1.8 m s$^{-1}$ to −3.1 m s$^{-1}$, a normalized reduction of 19% and 27%,

moving from unstable to stable at TKE_0 (Figure 12b,d). Thus, reducing TKE from 100% to 0% has a smaller impact on wake strength than increasing stability.

**Table 3. Wake wind speed reduction by stratification and TKE amount.**

|  | Unstable TKE_100 | Stable TKE_100 | Unstable TKE_0 | Stable TKE_0 |
|---|---|---|---|---|
| **Wind Speed Deficit** | −1.5 m s$^{-1}$ | −2.8 m s$^{-1}$ | −1.8 m s$^{-1}$ | −3.1 m s$^{-1}$ |
| **Normalized deficit** | 16% | 25% | 19% | 27% |

The areal extent of wakes changes by stability and added TKE. Wake deficits stronger than the −0.5 m s$^{-1}$ cutoff in unstable stratification at TKE_100 (Figure 12a) cover a total area of 7,208 km$^2$ and represent the best-case scenario where wakes impact the smallest area. In stable stratification at TKE_100 (Figure 12c), wakes cover a larger area of 15,948 km$^2$, or 2.2 times larger. A similar increase occurs using TKE_0, although areal coverage of the wake is larger due to weaker turbulence-induced wind speed replenishment from aloft. At TKE_0 in unstable conditions (Figure 12b), wakes stronger than −0.5 m s$^{-1}$ cover an

area of 7,780 km$^2$. In stable stratification, the area increases to 15,636 km$^2$ (Figure 12d), a factor of 2. The spatial extent of strong wakes spreading furthest throughout the region, representing the worst-case scenario, occurs in stable conditions at TKE_100. Wakes interact between immediate wind plant neighbors for all scenarios.

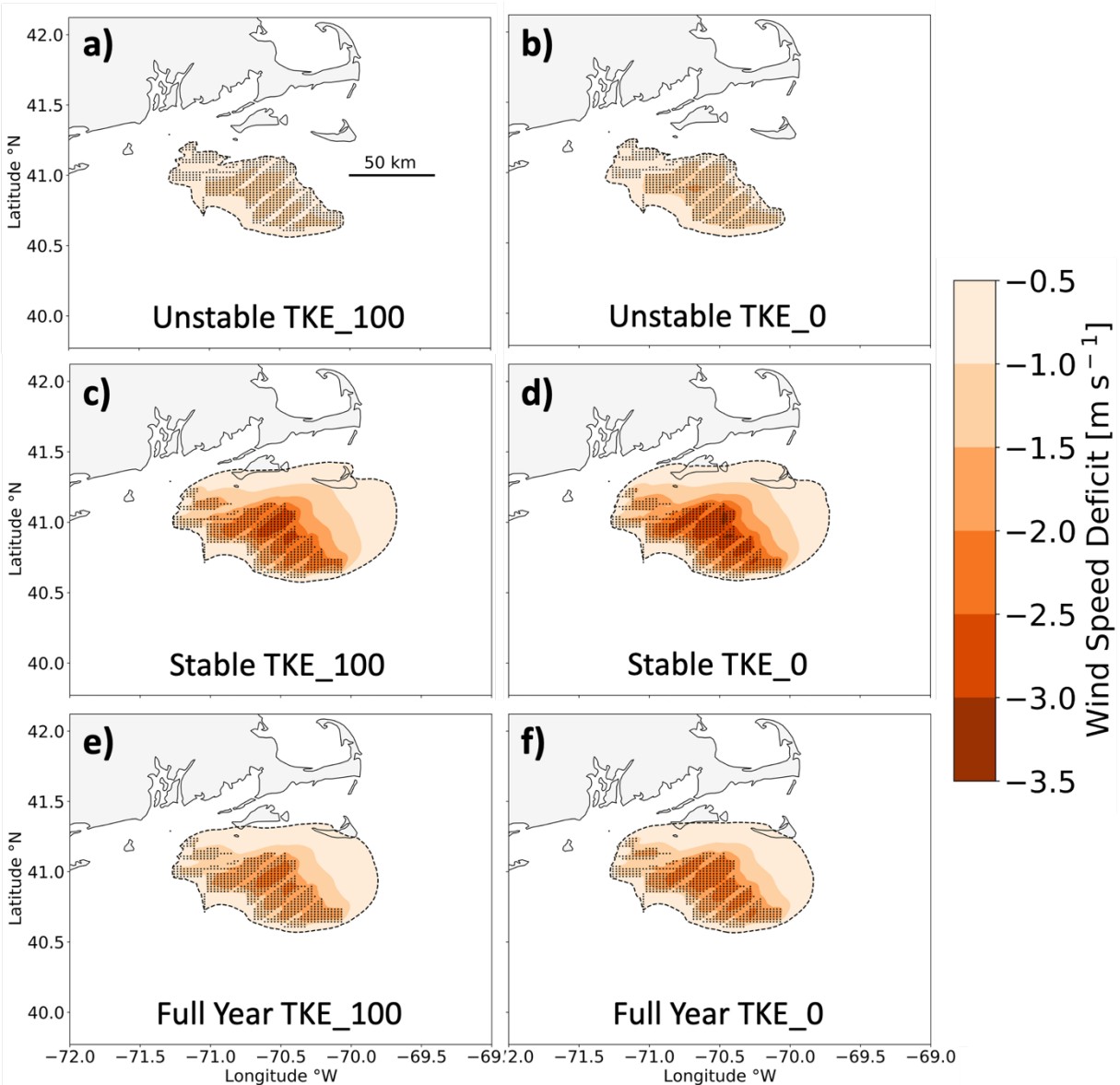

Figure 12: Average wake wind speeds among the lease areas during (a,b) unstable stratification, during (c,d) stable stratification, and (e,f) the full period 01 September 2019 to 01 September 2020. Wakes are simulated with 100% added TKE (a,c,e) or 0% added TKE (b,d,f). Wind speed deficits are shown by the colored contouring, and turbines are shown as the black dots. The −0.5 m s$^{-1}$ threshold is outlined by the black dashed line.

Stratification exerts a stronger effect on wake propagation distance than does TKE. For instance, wakes extending 3.7 km downwind in unstable conditions reach 55.4 km in stable conditions at TKE_100 (Figure 12a,c), similar to the estimate of 50 km from Golbazi et al. (2022). Likewise, wake deficits reaching 5.9 km downwind in unstable stratification reach 55.4 km

downwind in stable stratification at TKE_0 (Figure 12b,d). The same pattern exists for CA wakes (Figure D1). Overall, altering the added TKE amount has a small impact on the propagation distance of wakes relative to stratification, and

combining stable stratification with TKE_0 results in the strongest wakes.

Yearly averaged wakes show similar trends with TKE and stability (Table 4). The maximum wake strength intensifies from $-2.2$ m s$^{-1}$ to $-2.5$ m s$^{-1}$ moving from TKE_100 to TKE_0 (Figure 12e,f). Reducing TKE also increases the spatial coverage of wakes from 13,040 km$^2$ using TKE_100 (Figure 12e) to covering 13,268 km$^2$ using TKE_0 (Figure 12f). Downwind

propagation distances remain similar over the yearlong period with wakes reaching 43.4 km at TKE_100 and 41.3 km at TKE_0.

Table 4. The wake wind speed deficit, spatial extent, and downwind propagation distance by added TKE amount.

|  | Wind Speed Deficit | Spatial Extent | Propagation Distance |
|---|---|---|---|
| TKE_100 | $-2.2$ m s$^{-1}$ | 13,040 km$^2$ | 43.4 km |
| TKE_0 | $-2.5$ m s$^{-1}$ | 13,268 km$^2$ | 41.3 km |

Reduced TKE limits turbulence-induced momentum transport from aloft, thereby increasing wake strength. Counter-intuitively, longer-lasting wakes in TKE_100 develop from a larger reduction in momentum from wake recovery above the turbines (Fitch et al., 2012; Siedersleben et al., 2020), leaving less momentum available for replenishment downwind.

**Power Deficits**

 ### 3.1.1 External Wake Losses

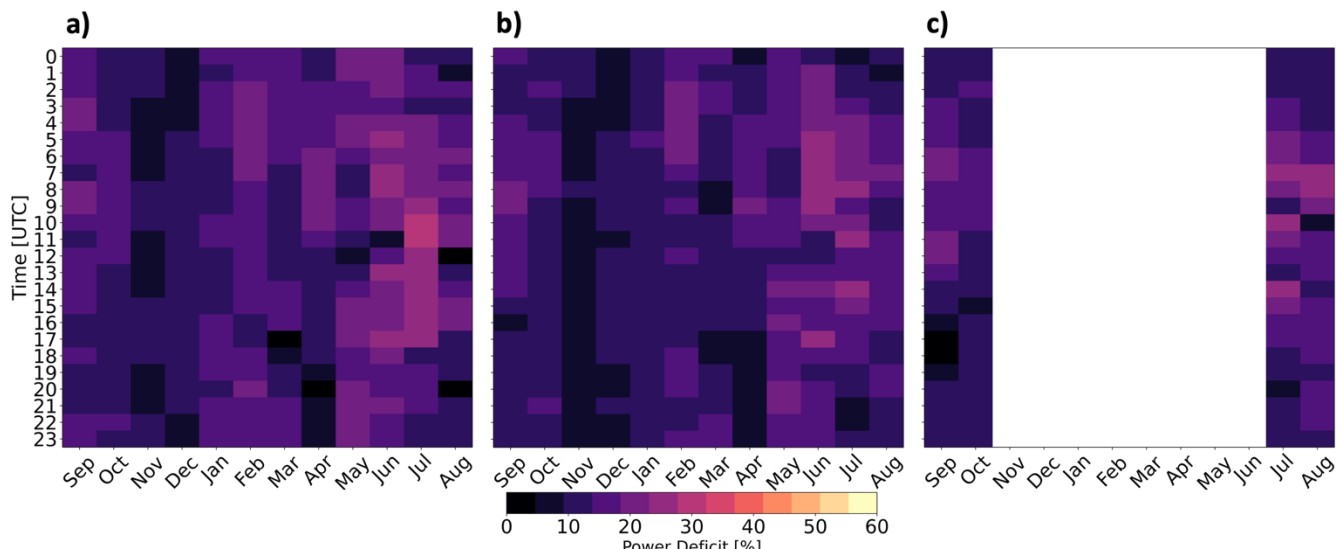

**Figure 13: The power deficit at ONE when waked by (a) the LA at TKE_0, (b) the LA at TKE_100, and (c) the CA at TKE_100. White space reflects the simulation period. The color bar is broad to facilitate comparison with losses in Figure 14.**

ONE experiences power deficits due to external wakes from the LA and the CA. Considering external wakes from the LA at TKE_0 (Eq. 15), the average yearlong power deficit at ONE is 14.7% (Figure 13a) and increases to 15.7% considering only the four stable CA months. When ONE is waked by the LA at TKE_100, the average yearlong power deficit reduces to 13.4% (Figure 13b) because increased turbulence supports faster replenishment. During the four months only, the deficit is 14.4%. When incorporating wakes from the CA (at TKE_100), the mean ONE power deficit (over four months) is 14.3% (Figure 13c).

By calculating the external power losses as the difference between total and internal losses (Eq. 16) instead, the deficits are 8.97% and 8.43% for the LA at TKE_0, and TKE_100, respectively. However, power losses vary as larger reductions from external wakes occur during summer whereas smaller reductions occur during winter.

External wake-induced losses vary both diurnally and seasonally. Larger power deficits occur more often during summer due

to stable stratification (Figure 10, Figure 11a). Smaller power deficits occur during winter (Figure 13), with faster winds that exceed rated wind speed and unstable conditions that erode wakes faster. Larger power deficits correspond with stable stratification in June and July. Conversely, smaller power deficits occur with unstable stratification throughout November and December. These patterns occur because colder air advects over warmer water in winter which causes unstable conditions that erode wakes faster. Conversely, warmer air advects over colder water during the summer, inducing stable conditions that

limit turbulent wake recovery. While wake-induced losses vary somewhat across the diurnal cycle, there is no discernible pattern. The ocean's large heat capacity suppresses daytime heating which limits changes in stratification, and by extension, the magnitude of changes in wake losses.

### 3.1.2 Internal Wake Losses

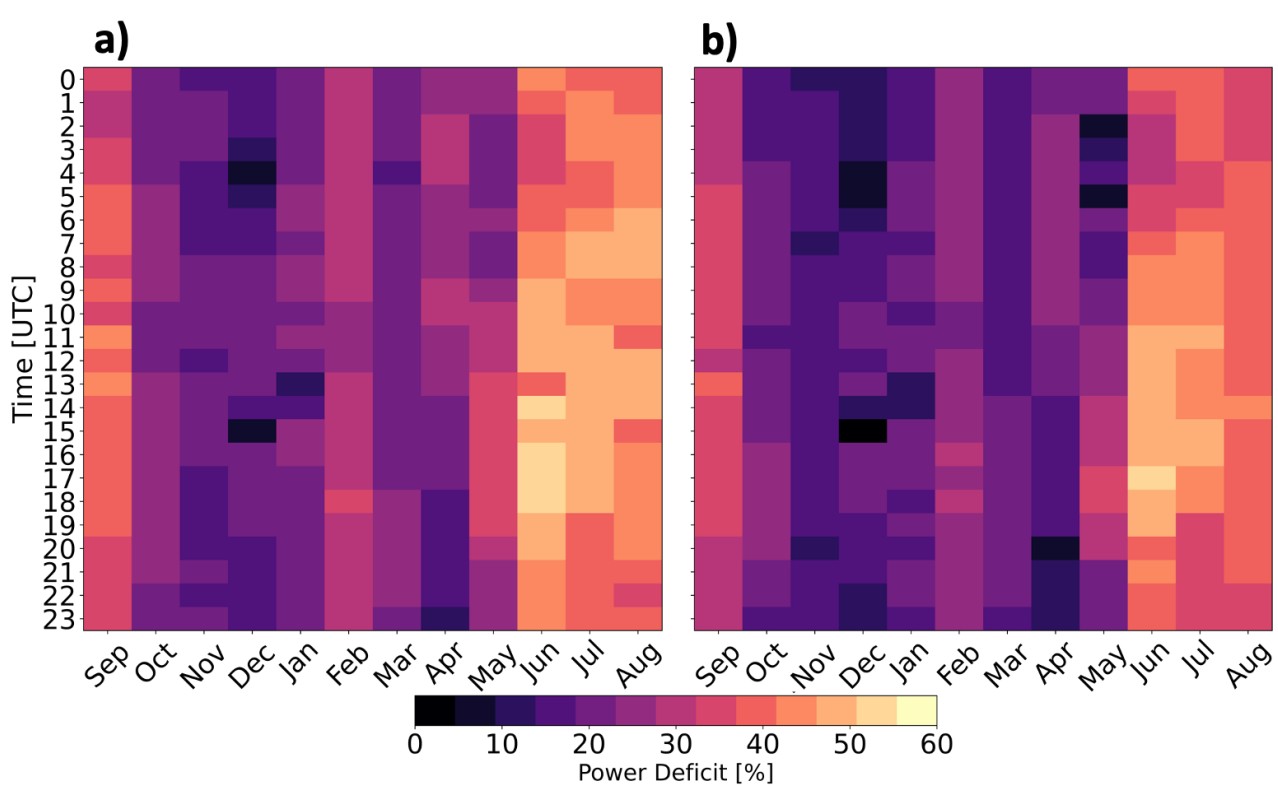

**Figure 14: The percentage of power loss at ONE from internal wakes at (a) TKE_0 and (b) TKE_100.**

Internal power deficits (Eq. 14) at ONE are at least 25% stronger than externally induced power deficits but experience similar variability with stability and TKE amount (Figure 14). Internal waking induces weaker deficits during winter and stronger 515 deficits during summer. As with external wakes, a clear diurnal pattern fails to emerge. Yearlong internal wakes from TKE_0 and TKE_100 induce power losses of 29.2% and 25.7%, respectively. During the four stable months only, the deficits increase to 36.9% and 32.9%, respectively. Using different PBL schemes with similar turbine spacing under steady-state idealized conditions, Rybchuk et al. (2022) find similar internal losses to capacity factor, up to 31.6%.

The average yearlong power deficits (Eq. 13) at ONE considering internal wakes and external wakes from the LA range between 38.2% (TKE_0) and 34.1% (TKE_100). These results concur with wake-induced losses found by Pryor et al. (2021) of 35.3% among the LA, based on 11 five day periods of different flow scenarios. Observations of wake-induced power losses have large variability over the year, ranging from as low as 5% to as high as 40% (Lee and Fields, 2021). Overall, external wakes produce yearly averaged power losses of 14.1%, whereas internal wakes induce larger losses of 27.4%. Thus, we stress

the importance of resolving region-specific and time-varying wakes for accurate energy prediction estimates.

**Annual Energy Production**

Predictions of energy supply are critical for planning, operations, and diversification of renewables. Without internal or external wake effects, ONE would produce 11.61 TWh and meet 10.02% of New England's average demand. Annual energy

production (AEP) from ONE, considering just internal wakes, reduces to 9.19 TWh (TKE_0) or 9.55 TWh (TKE_100), which could meet 7.94% to 8.24% of New England's demand. Including both internal and external wakes from the LA, ONE would produce 8.19 TWh (TKE_0) or 8.65 TWh (TKE_100), meeting 7.07% to 7.47% of demand.

Increasing the number of wind turbines increases the demand fulfilled; AEP from the LA is 68.12 TWh (TKE_0) or 70.9 TWh

(TKE_100), supplying 58.82% to 61.22% of New England's demand. On an hourly basis, the LAs fulfill demand only 24.6% (TKE_0) and 26.5% (TKE_100) of the time, highlighting the necessity for resolving accurate wake losses across the OCS. Previous work (Livingston and Lundquist, 2020) assuming a constant 20% wake loss, shown here to be underestimated, suggested that 2,000 10 MW turbines could meet New England's demand 37% of the time. In all, the LA, with 1,418 12-MW turbines, supply 68 TWh year$^{-1}$ and 71 TWh year$^{-1}$, or 1.72% (TKE_0) to 1.65% (TKE_100) of the nation's energy supply.

**Power Variability by TKE Amount**

### 3.1.3 Temporal Power Variability

While differences in wake strength between TKE amounts alter power production, wind speed exerts a larger influence. Maximum power is produced during spring with the least amount of power produced during summer (Figure 15a) for both

TKE_0 and TKE_100, because spring features faster wind speeds (Figure 15b). Power production responds to hub-height wind speeds (Figure 15) more than stability conditions (Figure 10, Figure 11). Reduced power production during summer may be problematic as New England's top-10 utility demand days since 1997 have all occurred in July or August (NEISO, 2023b).

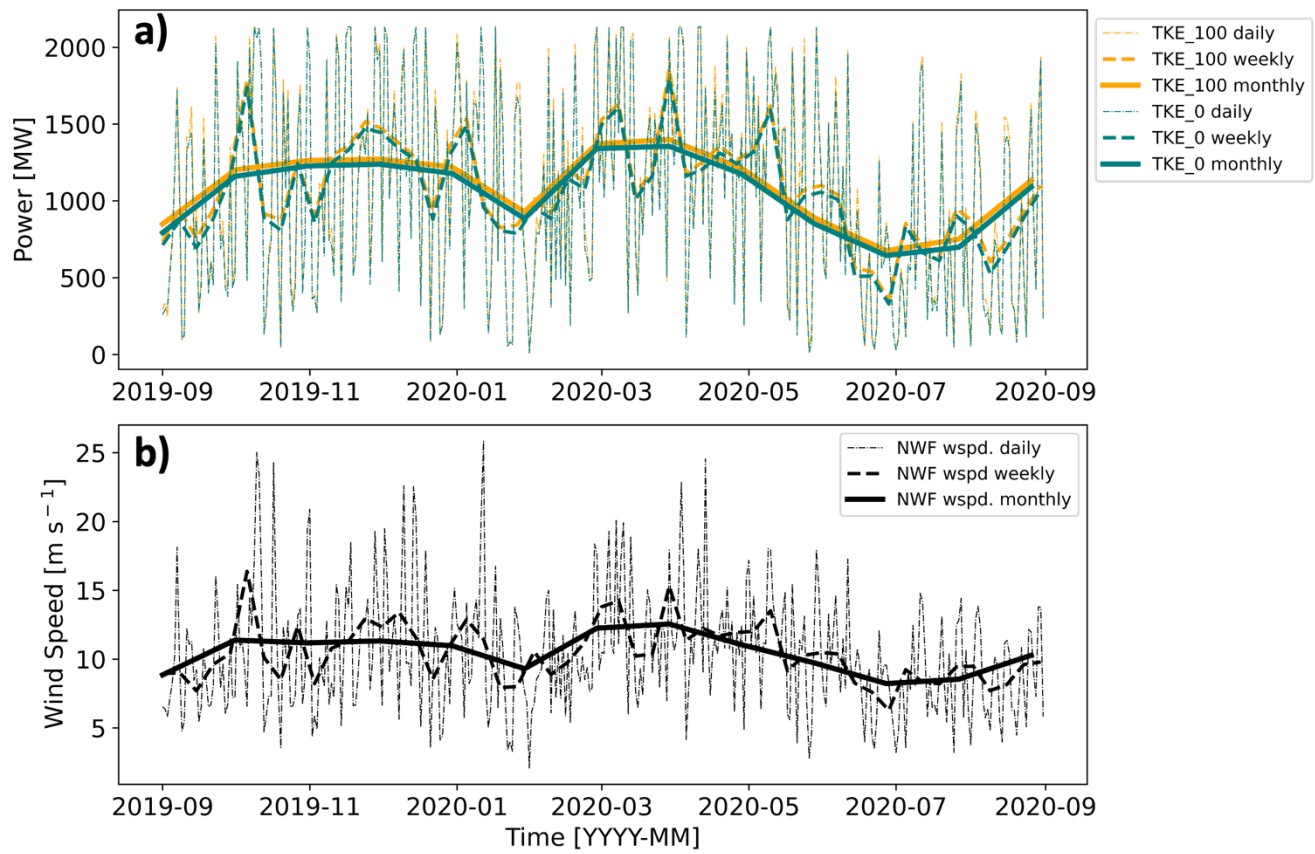

**Figure 15: (a) Total power production at ONE by TKE amount. TKE_100 power output is shown in orange and TKE_0 output is in teal. (b) Hub-height NWF wind speed at a point centered on the RIMA block. Dotted lines represent the daily average, dashed lines the 7 day average, and solid lines the 30 day average.**

Total power production varies slightly between TKE_100 and TKE_0. Due to weaker replenishment within the rotor-swept area, TKE_0 wakes are stronger, so TKE_0 produces less total power than TKE_100 (Figure 15a). Over the year, TKE_0 runs produce 96.2% (ONE) and 96.1% (LA) of the power of TKE_100. This difference does not arise from extreme outliers, as TKE_0 runs produce less power more frequently, at 71.3% (ONE) or 81.2% (LA) of the time.

 **3.1.4 Power Variability by Wind Speed**

Differences in power production (TKE_100 − TKE_0) vary by NWF hub-height wind speed (Figure 16). These differences are small at slow wind speeds, because little momentum is available for wake recovery, and at faster wind speeds within region 3 of the power curve (11−30 m s$^{-1}$) where wind speed changes do not affect power production (Figure 2a). Differences in wind speed within region 3 should have no effect on power production and are caused by numerical noise propagating through wind plant areas (Figure F1). The largest differences in power production occur in region 2 and around rated wind speed where the power curve is steep (Figure 2a, Figure 16). Additionally, large differences in power production can occur in specific meteorological conditions such as frontal propagation.

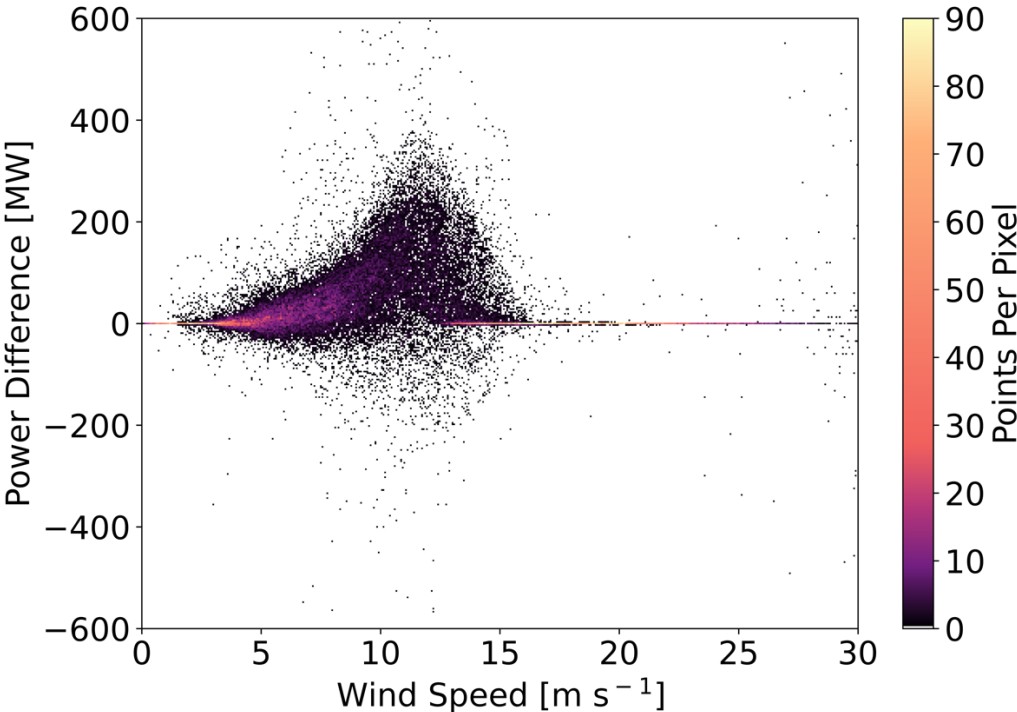

**Figure 16: The difference in power production (TKE_100-TKE_0) at ONE as a function of wind speed. Colored contouring depicts the density of scattered points per pixel. Wind speeds are obtained every 10 m from a point centered on ONE at hub height.**

Comparison of power production between TKE amounts by other meteorological variables lacked significant trends. For example, we additionally analyzed differences in power production by wind direction, following the hypothesis that northerly wind directions could transport more turbulence offshore because land has a higher roughness length than the ocean. TKE_100 runs may harness this mechanical turbulence more for wake replenishment. Analysis of differences in power production by PBL height also failed to show significant patterns. We assumed that higher PBL heights indicated a greater reservoir of

turbulence from which TKE_100 runs could replenish the wake, resulting in greater power production. Further analysis concluded by comparing power differences with the aforementioned variables' rates of change. However, we reached the same conclusions, as higher densities of scattered points existed around frequently occurring conditions such as southwesterly wind directions.

Wake strength varies spatiotemporally between TKE_0 and TKE_100 runs. While the mean difference in wind speed at hub height between TKE_100 and TKE_0 runs indicates that TKE_0 produces stronger wakes, this averaging may obscure the actual spatiotemporal variability. For example, a wind plant may have greater TKE_100 wake wind speeds while its nearby neighbor has greater TKE_0 wake wind speeds at the same point in time. Additionally, a specific wind plant may not consistently produce stronger wakes under one TKE setting. A wind plant may fluctuate between producing stronger wakes in TKE_100 runs and TKE_0 runs throughout time. This finding suggests that other boundary-layer dynamics play a role in wake strength, and the variability of power production must be explored.

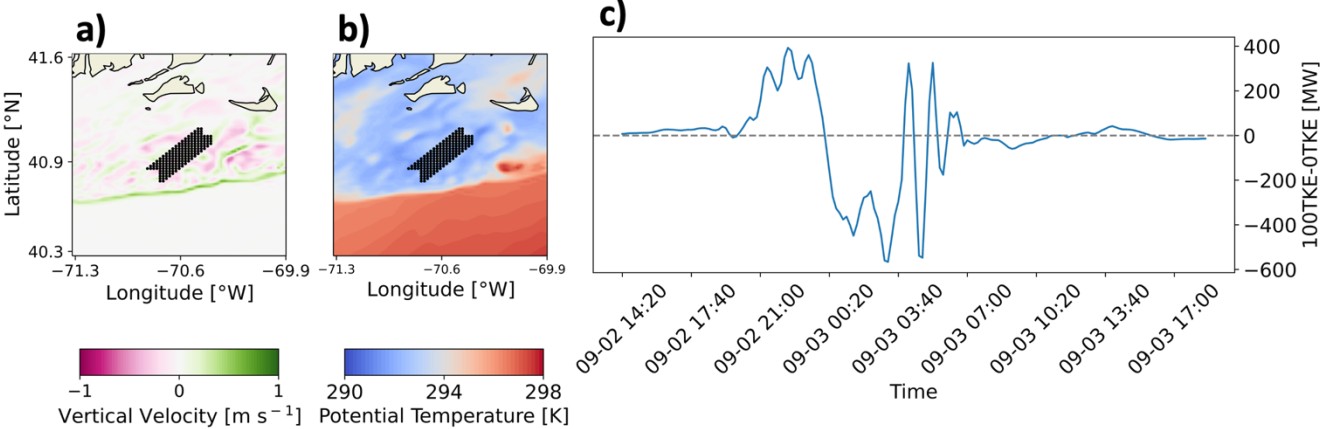

Figure 17: Propagation of a cold front through the ONE wind plant. (a) NWF vertical wind speed is shown as the colored contour with upward vertical velocities in greens and downward vertical velocities in purples. (b) NWF potential temperature is shown with lower temperatures in blues and higher temperatures in reds. In both (a) and (b), black dots indicate wind turbine locations in ONE TKE_0 and TKE_100. (c) The difference in power production between TKE_100 and TKE_0 is shown in MW, with positive values indicating that TKE_100 produces more power.

We note that wind speed and numerical noise are not the only contributors to power differences. One case study analysis shows that TKE_0 and TKE_100 separately produce more power within respective 99th percentiles over a short period of time in September (Figure 17c). Investigation reveals that a cold front propagated through the ONE wind plant from the northwest to the southeast during this period. The cold front is identified by a lenticular band of upward vertical motion at the frontal head followed by turbulent vertical motion (Figure 17a) in addition to advection of lower potential temperatures (Figure 17c). As

the cold front approaches, more power is produced by the TKE_100 simulation and is within the 99[th] percentile. When the frontal head first interacts with Vineyard Wind, more power is produced by the TKE_0 simulation and is within the 99[th] percentile. Conversely, TKE_100 produces more power following the frontal head. Frontal propagation can induce Kelvin–Helmholtz instabilities, the turbulence of which may aid wake recovery by vertically mixing momentum (Jiang, 2021). Increased turbulence in the TKE_100 simulation can harness more downward vertical transport of momentum from Kelvin–Helmholtz instabilities aft of the frontal head, increase wake replenishment, and produce more power.

## 4 Conclusions

This modeling study assesses the variability of wake effects across the mid-Atlantic OCS based on yearlong simulations, including a first step towards uncertainty quantification and approaches for distinguishing internal and external wake effects. In addition to a simulation without wind plants (NWF), validated by comparison to floating lidar observations, three wind plant layouts are explored including a representative wind plant alone (ONE), all lease areas (LA), and the lease areas plus the call areas (CA). Modifying the added TKE amount (TKE_0 or TKE_100) by turbines provides uncertainty quantification in power production estimates.

The OCS is characterized by more frequent unstable stratification during winter and stable stratification during summer (Bodini et al., 2019; Optis et al., 2020; Debnath et al., 2021). In stable conditions, wakes are stronger and propagate further downwind, (Fitch et al., 2013; Vanderwende et al., 2016; Porté-Agel et al., 2020). In the worst-case scenario where downwind wake recovery diminishes during stable stratification, mean wakes propagate 55 km downwind. While wakes may not reach downwind clusters on average, inter-cluster waking occurs intermittently. While TKE_0 produces stronger wakes than TKE_100, the downwind propagation distances do not differ.

Reduced wake wind speeds, as compared to the NWF simulation, affect power production. Yearly averaged wake losses induce power deficits at ONE from 38.2% (TKE_0) to 34.1% (TKE_100). This deficit comprises both internal and external waking. External wakes induce yearly averaged power losses of 14.7% (TKE_0) or 13.4% (TKE_100) whereas wakes from the CA induce similar losses of 14.3% over 4 months. Using an alternative method, external wakes induce losses of 8.97% and 8.43% for the LA at TKE_0, and TKE_100, respectively. Internal wakes at ONE promote larger power losses of 29.2% (TKE_0) or 25.7% (TKE_100). Wake-induced power losses vary seasonally with smaller diurnal variability. Larger power deficits occur during summer, where frequent stable conditions limit wake erosion. Although upwind clusters may generate strong external wakes among the LA, wind plant orientation with respect to prevailing winds can reduce adverse impacts from nearby neighbors. Ample distance for replenishment of external wakes by the CA moderates the negative effects. Internal wake losses remain larger due to shorter distances with limited wake recovery. Both external and internal wake-induced losses

grow in summer stably stratified conditions. These losses similarly increase in strength for TKE_0 simulations from inhibited recovery.

Resolving precise wake losses and AEP are crucial for stakeholders and grid operators. In the absence of wakes, ONE could supply 10.02% of New England's demand. Operating alone, ONE's supply reduces to 7.94% (TKE_0) or 8.24% (TKE_100). Adding external wakes from the LA, ONE's annual supply lessens to 7.07% (TKE_0) or 7.47% (TKE_100). Although wakes are stronger among the LA, the greater number of turbines can meet 58.82% (TKE_0) and 61.22% (TKE_100) of New England's demand, or roughly 1.72% and 1.65% of national demand. However, the LA only satisfy demand about 25% of the time on an hourly basis. Overall, spring features maximum power production with the fastest hub-height wind speeds. Wind speeds are slower in summer, reducing power production during July and August, which have featured New England's top-10 utility demand days since 1997 (NEISO, 2023b).

Variable TKE amounts marginally impact power generation. TKE_0 simulations average 3.8% less production than TKE_100 throughout the year, as reduced turbulence in TKE_0 limits momentum transport into the waked zone. Although differences in power production are small, both simulations exhibit large variability at short temporal periods. Improving WFP accuracy by accounting for wind shear throughout the rotor-swept region (Redfern et al., 2019) and dynamic air density may increase the variability in power production further (Wu et al., 2022). Further, different sizes of turbines may be installed in some of these regions, and the size of the turbine can influence the impacts of the turbine (Golbazi et al., 2022).

Future wind resource assessments may neglect differences between TKE_0 and TKE_100 because the power production offset is minor, although we identify a strong outlier during a frontal passage when differences in power production between TKE_100 and TKE_0 are large. While power production differences are minor, effects on other atmospheric variables may be more significant (Figure A1). Variability may be influenced by other meteorological conditions. Successive analyses should consider yearlong CA simulations to identify the full range of external wake impacts. Although we infer that the effects of CA wakes on ONE are small relative to LA wakes, yearlong estimates may show otherwise. Notably, we find that internal wakes have larger impacts on power production than those generated externally.

## 5 Appendices

### Appendix A

To assess the sensitivity of simulations to the amount of parameterized TKE, we conducted a set of 2 day test runs from 11 to 13 July 2017. This time period was chosen for its predominance of southwesterly winds, which represent typical conditions across the OCS and for the availability of Air-Sea Interaction Tower lidar observations for wind profile validation of the NWF

simulations. Test runs consist of 0% (TKE_0), 25% (TKE_25), 50% (TKE_50), and 100% (TKE_100) added TKE with the WFP.

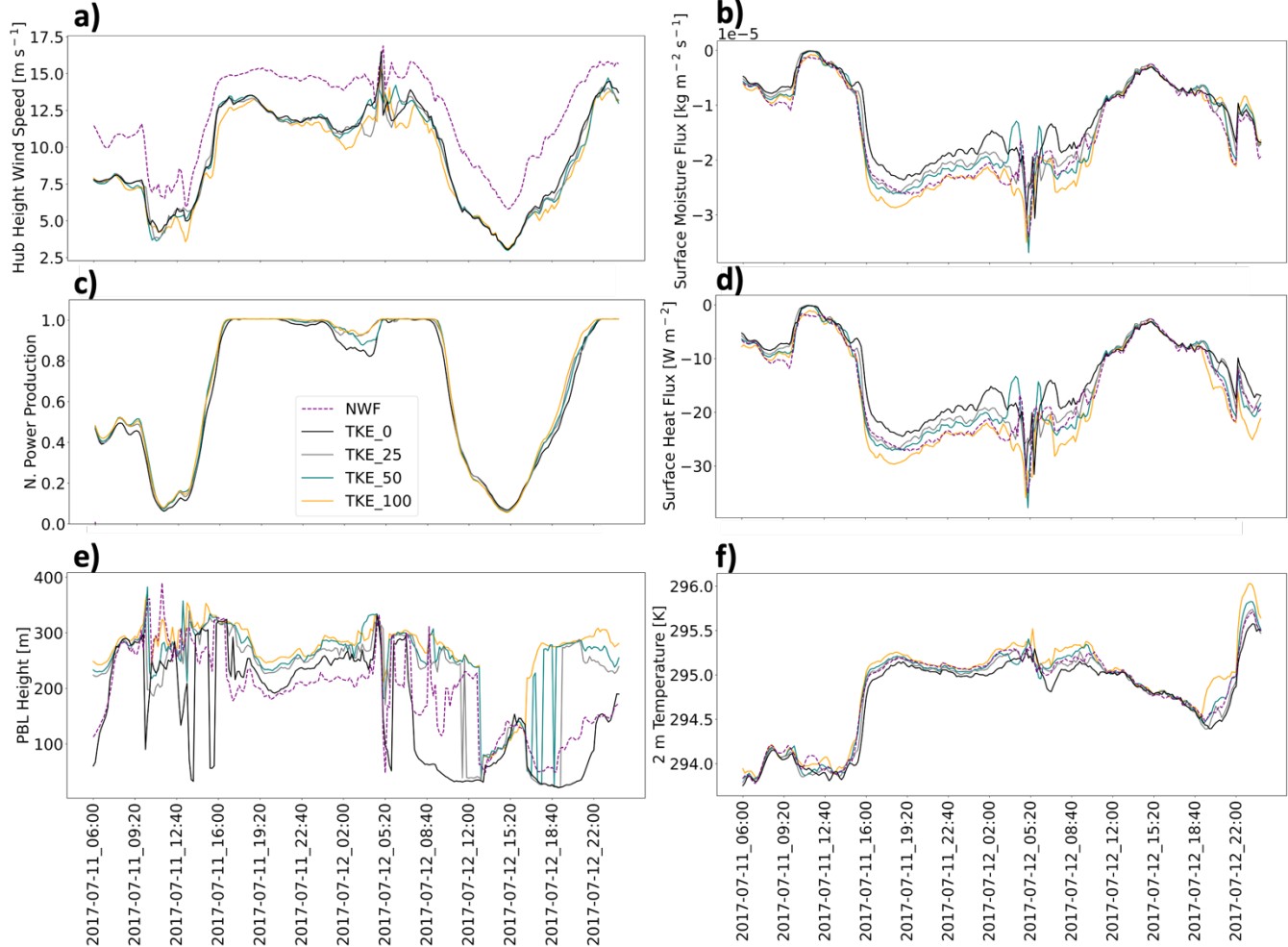

**Figure A1: The effects of modifying the amount of turbulent kinetic energy (TKE) during test runs. Panels show (a) hub-height wind speed, (b) surface moisture flux, (c) normalized power production, (d) surface heat flux, (e) planetary boundary layer (PBL) height, and (f) 2 m temperature. Values are collected from a point centered on RIMA block. Power production is the sum of all**
675 **cells containing wind turbines. TKE_100 is shown in orange, TKE_50 in blue, TKE_25 in gray, and TKE_0 in black, and NWF in purple dashes.**

Hub-height wind speeds vary by simulation type and added TKE amount (Figure A1a). Mean WFP wind speeds are always slower than NWF wind speeds, due to the momentum sink introduced by wind turbines, by 2.9 m s⁻¹. Larger variations between
680 wind speeds (Figure A1a) correspond with larger spreads in power output by TKE amount (Figure A1c). The sequencing of

power production driven by TKE amount remains consistent, namely that the differences progress from TKE_0 to TKE_25 to TKE_50 to TKE_75 to TKE_100.   Because power production totals for TKE_25 and TKE_50 are typically bounded by the totals for TKE_0 and TKE_100, production simulations incorporate TKE_0 and TKE_100 only to account for the full range of uncertainty throughout a full yearlong period from 01 September 2019 to 01 September 2020.

Although subtle, several important meteorological quantities from the model grid cell at the center of the RIMA block vary by the added TKE amount. For example, wind speeds are slower on 12 July between 12:00 and 16:00 UTC (Figure A1a). The wind speed reduction during this time period causes a corresponding decrease in turbulent transport of moisture. The mean difference in moisture fluxes throughout the full period between TKE_100 and TKE_0 is $2.84 \times 10^{-6}$ kg m$^{-2}$ s$^{-1}$ (Figure A1b).

Note that the surface moisture flux remains negative throughout the period. While maritime moisture profiles typically exhibit a decrease in concentration with height, corresponding with a positive flux, mixing from the turbines reduces the near-surface concentration and reverses the gradient.

Heat flux exhibits large variability. The mean difference in heat flux throughout the full period between TKE_100 and TKE_0 is 3.61 W m$^{-2}$ (Figure A1d). The wind speed decrease between 12:00 and 16:00 UTC reduces surface stresses and turbulent

transport of heat. The reduction in heat flux during this time period causes 2 m temperatures to decrease and exhibit less variability by TKE amount, with a mean difference of 0.26 K between TKE_100 and TKE_0 (Figure A1f).

The reduction in turbulent mixing lowers the PBL, regardless of TKE amount, to shallow heights between 30 to 80 m at 13:00

UTC (Figure A1e). The near-surface PBL height suppresses the small variations in turbulent mixing across test runs and causes fluxes to equalize. PBL heights differ the most by added TKE amount and may result from changes in weighting between two separate height determination methods present in the MYNN physics driver (Figure A1c).

## Appendix B

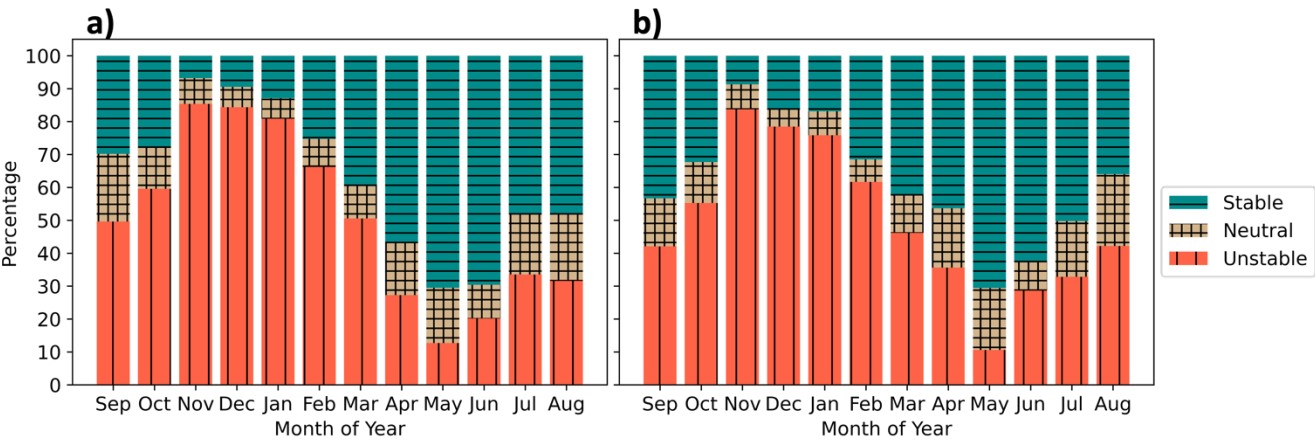

**Figure B1: Stability classification using the Obukhov length for the period 01 September 2019 to 01 September 2020 at the (a) E05 and (b) E06 lidars from NWF. Tan cross hatch are neutral stratification, blue horizontal bars are stable stratification, and red vertical bars are unstable stratification.**

Stratification at the E05 and E06 lidars (Figure B1) exhibits similar seasonal variability to the RIMA block (Figure 10). The winter months feature predominant unstable stratification caused by cold air advecting over a warm sea surface. Into the spring and early summer, stratification transitions to more common stable conditions as warm air advects over a cooler sea surface. Stratification is most commonly unstable in November and stable in May.

## Appendix C

Surface estimates of $L$ may not represent stability aloft (Figure C1) and may overestimate unstable conditions. When considering monthly averaged potential temperature profiles through the rotor layer, only November and December appear unstably stratified. While September and October appear predominantly unstable based on surface estimates, potential temperature gradients within the rotor-swept area suggest slightly stable conditions, supporting inferences that offshore conditions are stable during late summer. Therefore, our limited set of CA simulations focus on 01 September to 31 October 2019 and 01 July to 31 August 2020 for its presumed abundance of stable stratification.

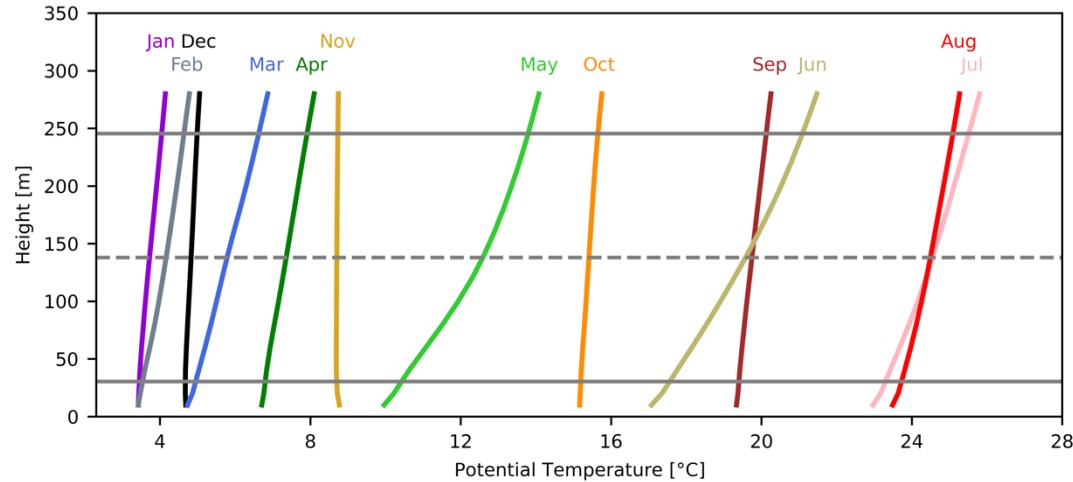

**Figure C1: Monthly averaged WRF-simulated potential temperature profiles at a point centered on the RIMA block. Horizontal gray lines indicate the levels of the hub height (dashed) and the rotor-swept area (solid).**

**Appendix D**

Wakes in the simulations with CA show similar dependence on stratification (Figure D1). Note that we simulate the CA for four months only (01 September to 31 October 2019 and 01 July to 31 August 2020) at one TKE level only (TKE_100) due to computational costs. The maximum wake strength intensifies from $-1.6$ m s$^{-1}$ to $-3.2$ m s$^{-1}$ moving from unstable to stable stratification (Figure D1b,c).

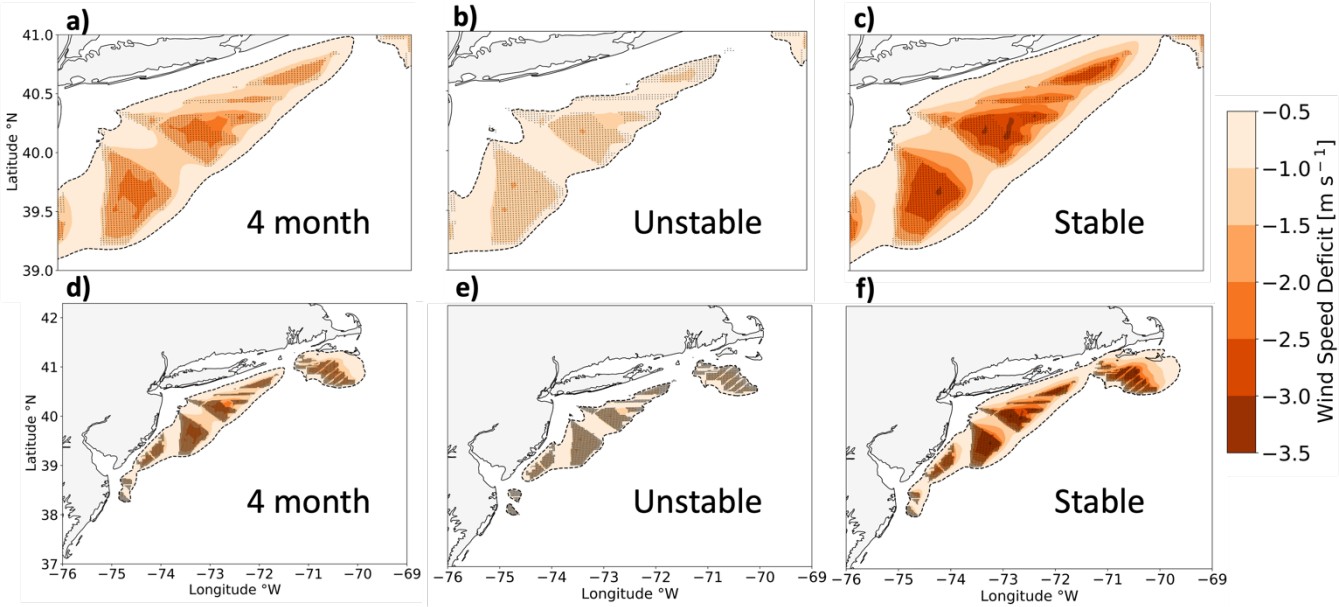

**Figure D1: Average wake wind speed deficits among the call areas (a,d) for the combined 4 month period, 01 September to 31 October 2019 and 01 July to 31 August 2020, (b,e) during unstable stratification, and (c,f) during stable stratification. All panels show 100% added TKE. Wake wind speed deficits are shown by the colored contour and turbines are shown as black dots. The upper row is zoomed in to increase granularity.**

Wake propagation distance for the call area simulation is also affected by stratification. During the 4 months considered, unstable, stable, and neutral conditions occur 38.2%, 53.4%, and 8.3% of the time, respectively. As such, there is essentially an even split between the percentage of occurrence of unstable and stable conditions. In unstable conditions, wakes from the two southernmost lease areas fail to reach neighboring downwind clusters on average, and no wakes stronger than this threshold reach the RIMA block (Figure D1e). In stable stratification, wakes from each cluster reach downwind clusters, including the RIMA block (Figure D1f). Averaged over all 4 months, wakes between LA and the CA along the New Jersey and New York Bight affect each other, but no wakes reach the RIMA block. Wakes may still interact with downwind plants at individual times and affect power production.

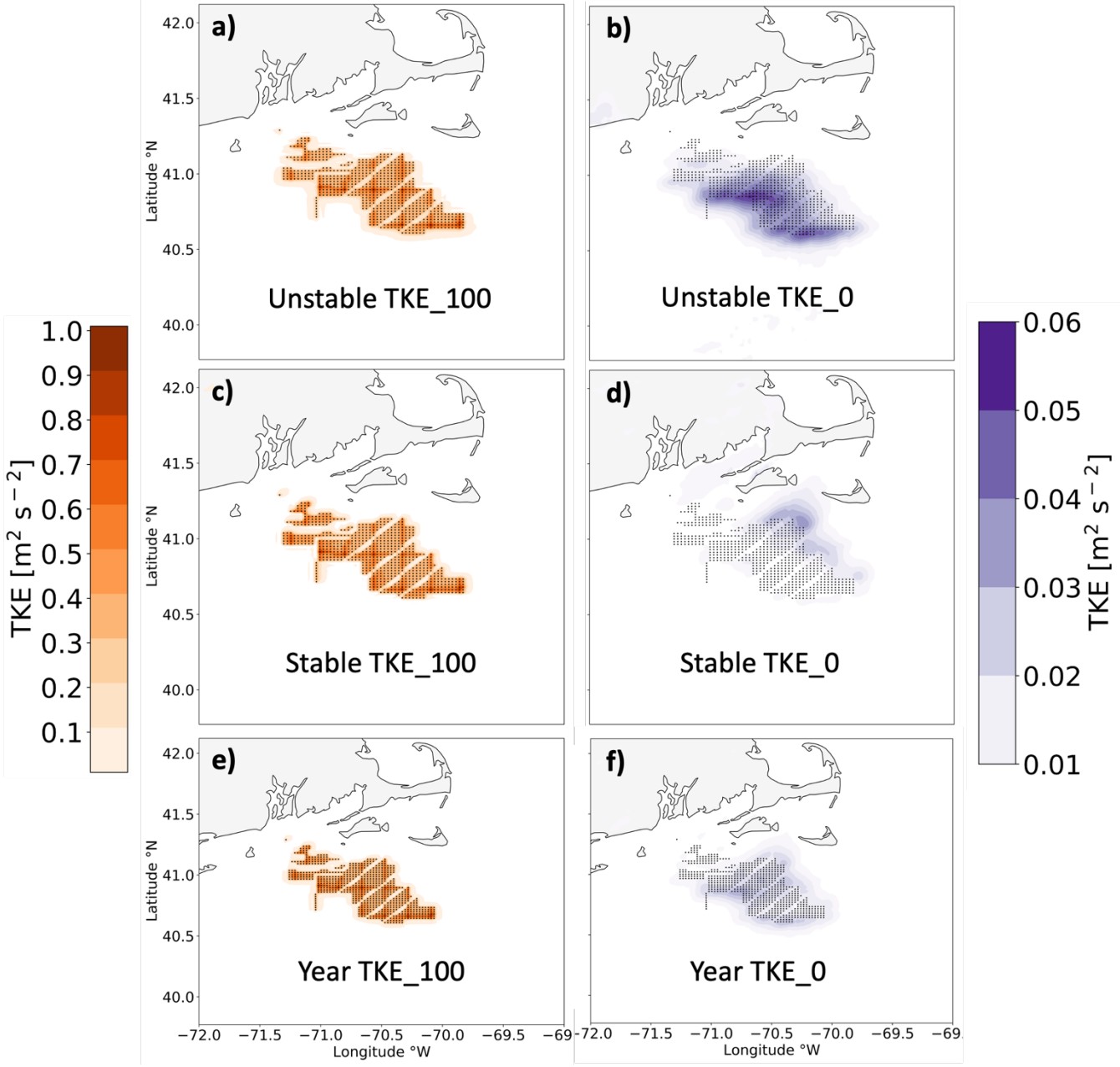

Fig. E1. Average hub-height (WFP-NWF) TKE difference among the lease areas during (a,b) unstable stratification, during (c,d) stable stratification, and (e,f) the full period 01 September 2019 to 01 September 2020. Panels show 100% added TKE (a,c,e) or 0% added TKE (b,d,f). TKE amount is shown by the colored contouring, and turbines are shown as the black dots.

Here, we characterize the (WFP-NWF) TKE differences by maximum value and by spatial extent. The maximum average TKE additions remain similar by stratification at TKE_100, reaching 1.00 m$^2$ s$^{-2}$, 1.01 m$^2$ s$^{-2}$, and 1.00 m$^2$ s$^{-2}$ during unstable conditions, stable conditions, and the full year, respectively (Fig. E1a,c,e). The amount of added TKE is not homogeneous across the wind plants in TKE_100, as the greatest contributions occur in grid cells containing more wind turbines. Some TKE is introduced in TKE_0 due to wind speed shear, although the amounts are over an order of magnitude smaller. The maximum average TKE amounts for TKE_0 are 0.05 m$^2$ s$^{-2}$, 0.03 m$^2$ s$^{-2}$, and 0.03 m$^2$ s$^{-2}$ during unstable conditions, stable conditions, and the full year, respectively. Being purely shear-induced, regions experiencing the most TKE in TKE_0 correspond more with the maximum wake wind speed deficits (Figure 12b,d,f).

We further characterize added TKE amounts by their spatial extent. We report the area encompassed by added TKE amounts greater than a threshold of 0.005 m$^2$ s$^{-2}$ because a cutoff of 0 m$^2$ s$^{-2}$ includes noise throughout the domain (Figure F1) and the spatial extent is not realistic. In TKE_100, the spatial extents are 10,724 km$^2$, 10,064 km$^2$, and 9,608 km$^2$ in unstable stratification, stable stratification, and for the full year, respectively (Fig. E1a,c,e). In TKE_0, the spatial extents are 13,888 km$^2$, 10,724 km$^2$, and 11,332 km$^2$ in unstable stratification, stable stratification, and for the full year, respectively (Fig. E1b,d,f).

**Appendix F**

Results can show evidence of numerical noise, which emerges when simulations incorporate the WFP (Ancell et al., 2018; Lauridsen and Ancell, 2018). In our simulations, these brief periods of numerical noise emerge and decay, often coincident with precipitation. While we expect differences in wake wind speed immediately downwind of power plants, it is unlikely that these differences could advect to the southeast corner of the domain, roughly 600 km southeast of the RIMA block (Figure F1a). If this numerical noise occurred in grid cells with turbines, then this noise would introduce error in power estimations.

We explored several approaches to mitigate the numerical noise, none of which succeeded. First, we increased the floating-point accuracy of numerical calculations by enabling double precision in WRF. Double precision limits the growth of rounding error to smaller magnitudes (Ancell et al., 2018). This attempt aimed to confine perturbations to smaller orders of magnitude that take longer amounts of time to become substantial. To prevent "runaway" error growth after long periods of time, we submit simulation restarts each month.

In observing a spatial correlation of numerical noise with convective precipitation during test runs, we reran test simulations with a more complex microphysics scheme. The Thompson microphysics scheme, used throughout, is double-moment with respect to cloud ice only. We substituted the Morrison microphysics scheme, which is fully double-moment with respect to cloud droplets and rain, cloud ice, snow, and graupel (Morrison et al., 2009). The use of Morrison microphysics did not improve numerical noise, so its computational cost could not be justified.

Next, we introduced a filter for shortwave numerical noise by prohibiting upgradient diffusion. Doing so requires setting the parameter diff_6th_opt to 2 in the namelist, as certain combinations of advection and diffusion orders are conducive to

790 mitigating noise around heavy precipitation (Kusaka et al., 2005). While Kusaka et al. (2005) found the combination of fifth-order advection and sixth-order diffusion to perform best, we had previously attempted this combination because default advection in WRF is fifth-order. Thus, we attempted the next best recommendation—combining sixth-order advection and diffusion. Again, this combination did not improve results.

We made a final attempt at noise reduction by running an ensemble of three members using a stochastic kinetic energy backscatter scheme. Ensemble members contain seeds with variable time steps that randomly inject kinetic energy into grid cells (Berner, 2013). These stochastic supplements replenish the kinetic energy sink from unresolvable subgrid-scale processes. We followed recommendations to perturb the stream function and potential temperature backscatter rates by $1 \times 10^{-5}$ and $1 \times 10^{-6}$, respectively. Again, while subtle differences emerged between the simulations, little improvement was found.

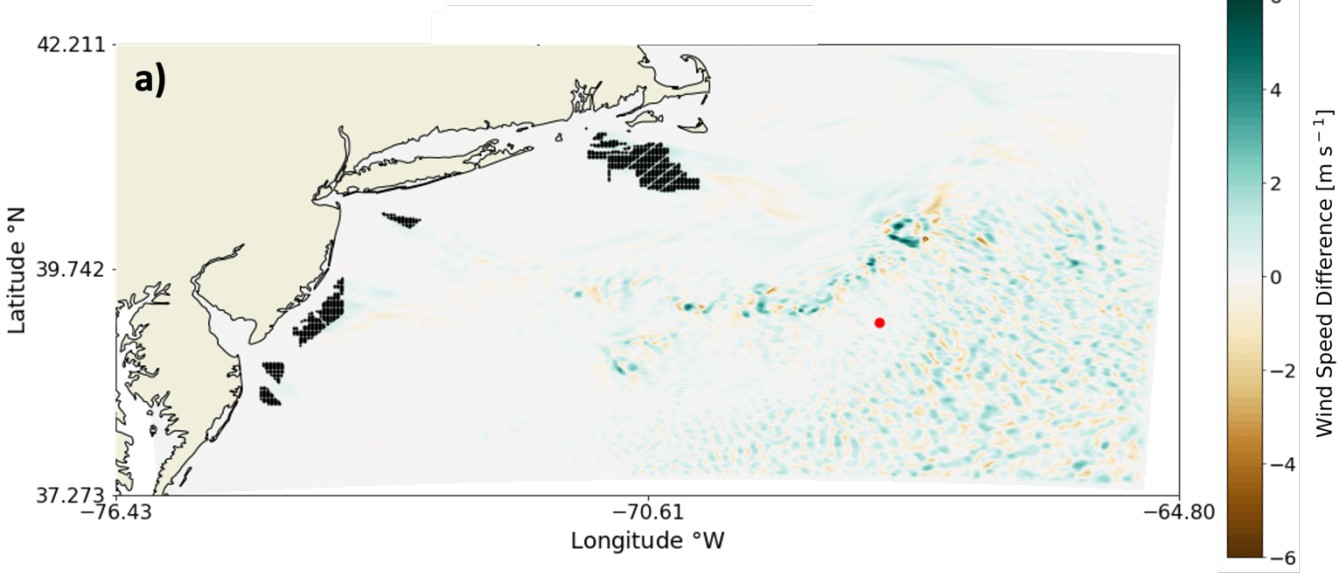

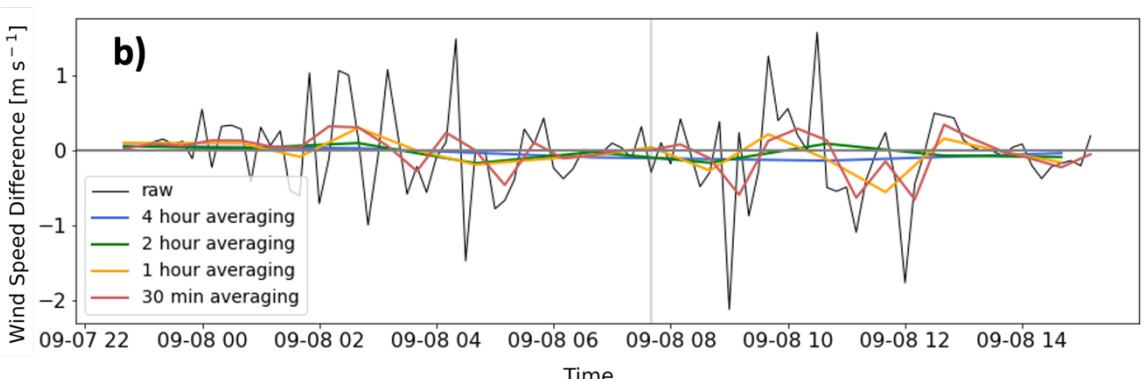

**Figure F1: (a) The wind speed difference between TKE_100 and TKE_0 at the hub height from LA runs. Wind turbines are shown as black dots. Blue contouring indicates TKE_100 produced faster wind speeds and vice versa. (b) Wind speeds obtained at the red circle in (a) are shown as a time series. The raw difference in wind speeds and averaging periods are shown as different line colors in the time series. The gray vertical line shows the time stamp of the map.**

We saw little improvement from the aforementioned preprocessing efforts. Given this lack of improvement and a need to conserve computational resources, we employed averaging during postprocessing to alleviate the effects of noise. Modifying averaging periods impacts the range of numerical noise in the wind speed field (Figure F1b). Noise occurring in grid cells containing turbines could undermine power estimation accuracy and we observed noise occurring in the southeastern portion of the domain. Subtraction of wind speeds between simulations with variable TKE amounts should only show differences within the wake, and such differences are a result of noise. Averaging periods provides greater relief. While 2 and 4 hour

averaging periods deliver the best results, these temporal scales can hide important diurnal variability. Conversely, a 30 minute averaging period can improve results, but local extrema occasionally reach magnitudes similar to the magnitudes of the raw noise. Thus, hourly averaging can mitigate noise without masking important variability. As a final note, other researchers have benefitted by employing grid nudging within this domain above the PBL (Golbazi, M., personal communication, September 2022).

## 6 Code and Data Availability

The data and files that support this work are publicly available. The ERA5 boundary conditions can be downloaded from the ECMWF Climate Data Store at https://cds.climate.copernicus.eu/cdsapp#!/dataset/reanalysis-era5-pressure-levels?tab=form. Shapefiles including the bounding extents of the lease and call areas are at https://www.boem.gov/renewable-energy/mapping-and-data/renewable-energy-gis-data. Individual turbine coordinates and their power and thrust curves are provided at https://zenodo.org/record/7374283#.Y4YZxC-B1KM. WRF namelists for NWF and WFP simulations can be obtained at https://zenodo.org/record/7374239#.Y4YaOy-B1KM. The simulation output data will be available in HDF5 format at https://data.openei.org/submissions/4500.

## 7 Author Contributions

Conceptualization: JKL and MO. Methodology: DR, JKL, and MO. Software: DR, AR, MR. Validation: DR. Formal analysis: DR. Investigation: DR and JKL. Resources: MO, NB. Writing – original draft: DR and JKL. Writing – review and editing: all co-authors. Visualization: DR. Supervision: JKL, MO, NB. Project administration: MO and NB. Funding acquisition: MO and NB.

## 8 Competing Interests

At least one of the (co-)authors is a member of the editorial board of Wind Energy Science. Furthermore, Mike Optis co-authored the submitted manuscript while an employee of the National Renewable Energy Laboratory. He has since founded Veer Renewables, which recently released a wind modeling product, WakeMap, which is based on a similar numerical weather prediction modeling framework as the one described in this manuscript. Data from WakeMap is sold to wind energy stakeholders for profit. Public content on WakeMap include a website (https://veer.eco/wakemap/), a white paper (https://veer.eco/wp-content/uploads/2023/02/WakeMap_White_Paper_Veer_Renewables.pdf) and several LinkedIn posts

promoting WakeMap. Mike Optis is the founder and president of Veer Renewables, a for-profit consulting company. Mike Optis is a shareholder of Veer Renewables and owns 92% of its stock.

## 9 Acknowledgements and Statements

This work was supported by an agreement with NREL under APUP UGA-0-41026-125. This work was authored [in part] by the National Renewable Energy Laboratory, operated by Alliance for Sustainable Energy, LLC, for the U.S. Department of Energy (DOE) under Contract No. DE-AC36-08GO28308. Funding was provided by the U.S. Department of Energy Office of Energy Efficiency and Renewable Energy Wind Energy Technologies Office and by the National Offshore Wind Research and Development Consortium under agreement no. CRD-19-16351. The views expressed in the article do not necessarily represent the views of the DOE or the U.S. Government. The U.S. Government and the publisher, by accepting the article for publication, acknowledge that the U.S. Government retains a nonexclusive, paid-up, irrevocable, worldwide license to publish or reproduce the published form of this work, or allow others to do so, for U.S. Government purposes. Neither NYSERDA nor OceanTech Services/DNV have reviewed the information contained herein and the opinions in this report do not necessarily reflect those of any of these parties. A portion of computation used the Blanca condo computing resource at the University of Colorado Boulder. Blanca is jointly funded by computing users and the University of Colorado Boulder. A portion of computation used the Summit supercomputer, which is supported by the National Science Foundation (awards ACI-1532235 and ACI-1532236), the University of Colorado Boulder, and Colorado State University. The Summit supercomputer is a joint effort of the University of Colorado Boulder and Colorado State University. A portion of this research was performed using computational resources sponsored by the DOE's Office of Energy Efficiency and Renewable Energy and located at NREL.

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
