# Peer review of "Seasonal Variability of Wake Impacts on U.S. Mid-Atlantic Offshore Wind Plant Power Production"

_Wind Energy Science, 2023_

## Referee Comment (RC1)

**Review of "Annual variability of wake impacts on Mid-Atlantic offshore wind plant developments" by Rosencrans, Lundquist, Optis, Rybchuk, Bodini, and Rossol, submitted for publication in Wind Energy Science**

This is a WRF modeling study of the wakes of future offshore wind energy areas planned along the US east coast. It considers three scenarios of offshore wind development: Vineyard Wind only, lease areas in the Mid-Atlantic, and lease + call areas in the Mid-Atlantic. An impressive modeling effort was undertaken, with nested high-resolution WRF runs for one year repeated for the three scenarios above, plus the control case with no farms, plus runs with 0% and 100% TKE added. The team had also numerical noise issues, as seems to be the norm when the wind farm parameterization is turned on, and therefore had to deal with several additional runs to take care of it. The paper is definitely worth publishing as it includes interesting and valuable results. It is exceptionally well written. It was a pleasure to read such a good paper!

I have two somewhat major issues to recommend addressing prior to publication and several minor comments.

**Major issues**

1. The study focuses too much on added TKE

The paper promises to study annual variability of wake impacts (see comment #4 below about how it is monthly and diurnal, not annual). Such a study should have a main run with certain fixed parameters, perhaps a few case studies of special interest or a validation effort, and a few case studies to assess the sensitivity to some of the parameters. Instead, in this study a lot of effort was put on the sensitivity. The parameter of focus is the amount of added TKE, which is 25% in the default settings of the WRF model, and which was found here to have a relatively small impact on the power ouput (<5%). One would expect that the main run would be with 25% TKE and then a few cases (perhaps one week in each season) would be run with 0% and 50% and 100% TKE, in addition to 25% TKE, to assess sensitivity. The main results reported in the abstract and in the conclusions would be obtained with the default 25% TKE and a sentence or two would address sensitivity to TKE.
Instead, on one hand the study focuses excessively on the sensitivity to added TKE, because all the runs have been repeated entirely for 0% and 100% TKE, when a few weeks would have been sufficient. Of the 6 figures in the paper that describe modeling results (Figures 8-13), all of them are doubled to show 0% and 100% TKE. This would be understandable if TKE had a large impact on power output, but it did not (at most 5%).
On the other hand, the team did not perform a year run with the recommended value of 25% TKE. Thus in principle every value that they report in the abstract should be a range, but it is not. Plus, no TKE results are shown (I would like to see the equivalent of Figure 8 but for added TKE).
In summary, this study focuses excessively and at the same not enough on TKE. An obvious recommendation would be to ask the team to conduct a new one-year run with 25% TKE and

rewrite the paper to focus on those results and reduce the sensitivity analysis. But I think that this would be an excessive request, plus there is already a lot of value in the current runs.

My first recommendation is therefore that the results in the abstract and conclusions, which do not report a range (i.e., the range of results with 0% and 100% TKE) but are presented without explanation as one value (see comment below), be modified by either reporting always the range, or by using an interpolation based on the few days of 25% TKE results that the authors have already run (Fig. A1). What I mean is that the team could obtain a relationship between average power output (or wind speed deficit or whatever the parameter of interest is) with 0%, 25%, and 100% TKE from the few simulated days. This relationship does not seem to be linear from Fig. A1. An example of this relationship might be something like: the power at 25% TKE is the mean of that at 0% and 100% TKE, on average. Then use that relationship to report one value (per parameter) in the abstract and conclusions, that "fitted" to 25% TKE.

The second recommendation is that the authors add a figure and discussion on the TKE distribution in the wakes with 0% and 100% TKE, like Fig. 8.

2. The calculation of the losses from external wakes may be incorrect

From the abstract, the effects of internal wakes are reported to be -27.4% and the effects of external wakes are -14.1%. The sum of the two is -41.5%. However, the combined effect is reported to be -35.9%. This is problematic. At first sight, this discrepancy may be the result of the non-linearity of the wake processes. If so, all the authors need to change is to reverse the order of two sentences and add a few words in the abstract to explain it: "Internal wakes alone cause greater year-long power losses (27.1%) compared to external wakes (14.1%). When both are present, however, the mean year-long wake impacts reduce power output by 35.9%, which is lower than the sum of the two due to non-linear processes."
However, I suspect that there might be a design issue in the way the power losses are calculated in Eq. (9) and (10). Aside from the unclear notation (see comment below), the denominator of the two equations is not the same and that may be why the discrepancy arises. Eq. (10) is correct because there is no double counting: there are no losses in the denominator and the internal losses are only in the numerator. In Eq. (9), however, there are internal losses in both the numerator and the denominator, and they are not equal. The internal losses are not equal in the VW and CA cases because, as upstream conditions change due to external wakes in the CA case, the internal wakes change too and therefore the internal wake losses do not "cancel out", there is still some influence from the internal wake losses. As such, the ratio in Eq. (9) does not quantify just external losses because it still contains the effect of internal losses; it quantifies a mix of internal and external losses.
I suggest that the authors report Eq. 10 first (Loss_internal). Then, they should replace P_VW at the denominator with P_NWF in Eq. 9, to obtain the total effect from internal and external wakes due to the CA areas (call it Loss_total). We know this value: it should be -35.9% (from l. 384). The effect of the external wakes then is the difference between the total losses and the value from Loss_internal:

Loss_external = Loss_total – Loss_internal = -35.9% - (-27.4%) = -8.5%  (Eq. 11)

This way the denominator is the same and the individual values for external and internal sum up to the correct total.

3. The stability classification is not adequate.

The authors use a very simple classification for stability based on the value of L (Eq. 8). Neutral conditions are those with abs(L)>1000 m. This is inconsistent with the published literature, e.g., Gryning et al. (2007) and Sathe et al. (2011) used 500 m, Wharton and Lundquist (2012) used 600 m, Rajewski et al. (2013) used 400 m, Archer et al. (2016) used 500 m. In fact, too few neutral cases were found here, less than 2.5% of the time (p. 29 l. 595). I am unsure what to recommend here because there is not an "accepted" value of L for neutral conditions, but the authors need to assess the sensitivity of their results to a few values, at a minimum 500 m. This could possibly help with the previous inconsistencies in the areal extent and wake length, as days that were actually neutrally stratified may have been mixed in with days with other stabilities to obfuscate some of the relationships.

**Minor issues**

4. The title needs improvements

The title suggest that the wake impacts "on" the wind farm development will be studied. This is somewhat inaccurate, as the study is about the wake impacts on offshore wind power production or output, not on the development. Development is choosing the number of turbines or their specs or their layout, which are all fixed in this study; or, development can be how the wind farm installations grow/change with time. Either way, the development here is given (3 scenarios), what changes is the power output.
Also, the title mentions the "Mid-Atlantic" as the focus area, but technically speaking the Mid-Atlantic stops as far north as New York state. From the U.S. perspective, the Vineyard Wind project is not in the Mid-Atlantic and neither are the northeastern lease areas of RI or MA. According to Wikipedia, the following states are included in the Mid-Atlantic:  Delaware, Maryland, New Jersey, New York, Pennsylvania, Virginia, West Virginia, and Washington DC. To non-U.S. readers, "Mid-Atlantic" could be the Equatorial zone, as the Atlantic Ocean extends between the two Poles. I don't have a good recommendation for an alternative, but perhaps "U.S." should be added in the title because the study focuses on the U.S. offshore areas after all.
Last, "annual" variability suggests that many years were studied to understand how the production changes from one year to the next. Instead, only one year was simulated here. Thus the variability studied here is monthly/seasonal and diurnal, but not annual.

5. Simplify naming
There is no need to add "_only" to the name of the run with only the Vineyard Wind farm. Just call it "VW."

6. Unclear notation in Eqs. (9)-(10)
These equations have already been discusses at comment #2, here I am focusing on the notation only. Eliminating "_only" will help (comment #5). P_WV_waked is not defined and uses a notation that differs from that of all other subscripts. All the other subscripts refer to a specific run, whereas "waked" refers to, I believe, a subset of grid points. But the same subset of grid points was used for all other denominators and numerators, thus the confusion. Plus the term P_VW_waked refers to run CA, I believe. I suggest something like (not including my recommendation from comment #2 above):

$$Loss_{ext} = \left(1 - \frac{P_{CA}}{P_{VW}}\right) \times 100\%$$

In the text below the equation then you specify that this equation is obtained from the grid cells over Vineyard Wind.

7. L. 318-320 ("While here … schemes"): this discussion is irrelevant and unnecessary here.

8. P. 17: some of these results are rather counter-intuitive.
if the TKE_100 runs produce weaker deficits and smaller wake areas, then the wakes should be shorter, whereas the authors report 58 km for TKE_100 and 55 km for TKE_0. The explanation provided is vague and unsupported (l. 346: "larger reduction in momentum aloft"??). The authors do not report exactly how the wake length was obtained. I suspect the method was somewhat empirical and in fact it is giving counter-intuitive results. I suggest that either the authors develop an objective and automated method for calculating the wake length and, if the inconsistency persists, they document and explain it; or that they remove any discussion of the wake length.

9. Improve Fig. A1.
Replace 10-m wind speed with 140-m wind speed in Fig. A1. In all panels (except c), add the results from NWF to appreciate the magnitude of the impacts.

10. P. 26-27: the discussion is unclear, the authors report "reductions" in several sentences, but it is unclear what is changing and what the reference is: are they discussing changes from TKE_0 to TKE_100 or from TKE_100 to TKE_0 or from NWF?

11. Fig. D1: need a legend for the colors. Also, are these wind speeds or wind speed deficits? The caption indicates wind speed.

---

## Author Comment (AC1)

**Response to Referee #1**

Referee comments appear in black and author responses appear in blue.

This is a WRF modeling study of the wakes of future offshore wind energy areas planned along the US east coast. It considers three scenarios of offshore wind development: Vineyard Wind only, lease areas in the Mid-Atlantic, and lease + call areas in the Mid-Atlantic. An impressive modeling effort was undertaken, with nested high-resolution WRF runs for one year repeated for the three scenarios above, plus the control case with no farms, plus runs with 0% and 100% TKE added. The team had also numerical noise issues, as seems to be the norm when the wind farm parameterization is turned on, and therefore had to deal with several additional runs to take care of it. The paper is definitely worth publishing as it includes interesting and valuable results. It is exceptionally well written. It was a pleasure to read such a good paper!

We thank the reviewer for reading thoroughly and providing thoughtful suggestions to improve this article.

**Major Comments**
1. **The study focuses too much on added TKE.**
The paper promises to study annual variability of wake impacts (see comment #4 below about how it is monthly and diurnal, not annual).

Thank you for pointing out our inaccurate use of word choice. Annual variability refers to multi-year studies while our research focuses on one year. You leave more title-specific suggestions, including this one, in comment #4, so we refer to all title changes in our response to comment #4.

Such a study should have a main run with certain fixed parameters, perhaps a few case studies of special interest or a validation effort, and a few case studies to assess the sensitivity to some of the parameters. Instead, in this study a lot of effort was put on the sensitivity.
The parameter of focus is the amount of added TKE, which is 25% in the default settings of the WRF model, and which was found here to have a relatively small impact on the power output (<5%). One would expect that the main run would be with 25% TKE and then a few cases (perhaps one week in each season) would be run with 0% and 50% and 100% TKE, in addition to 25% TKE, to assess sensitivity. The main results reported in the abstract and in the conclusions would be obtained with the default 25% TKE and a sentence or two would address sensitivity to TKE.

Thank you for the suggestion. In planning our simulations, we had extensive discussions about the merits of 0% vs 25% TKE, and finally decided to use 0% as a bottom limit rather than 25%. The 25% recommendation was based on only one study. Archer et al. (2020) recommended the use of 25% TKE based on idealized conditions, with neutral stratification, and for a one-wind-

turbine setup.  The 0% added TKE is more similar to the impact in the Volker et al. parameterization which has been used in several studies.

Instead, on one hand the study focuses excessively on the sensitivity to added TKE, because all the runs have been repeated entirely for 0% and 100% TKE, when a few weeks would have been sufficient. Of the 6 figures in the paper that describe modeling results (Figures 8-13), all of them are doubled to show 0% and 100% TKE. This would be understandable if TKE had a large impact on power output, but it did not (at most 5%).

Given the extensive discussion of the value of added TKE in the literature, we wanted to thoroughly document its variability. And, as pointed out above, 0% is more similar to Volker et al. and the 0-100% range therefore includes not just the Archer et al. suggestion but Volker et al. as well.

On the other hand, the team did not perform a year run with the recommended value of 25% TKE.

As noted above, the recommendation of 25% comes from one study, and we sought to provide more extensive bounds on the variability that could be introduced with a range of added TKE values. Archer et al. (2020) recommended the use of 25% TKE, and while extremely helpful to pioneer a suggestion for this issue, that recommendation is unfortunately limited in application for being run under idealized conditions, with neutral stratification, and for a one-wind-turbine setup.  For this reason, Archer et al. reported that 25% TKE was the best choice for their setup, and further investigation is still required. There is uncertainty on what the "rule of thumb" TKE amount should be in regional wind plant modeling, and our results, because we explore the whole range of possibilities from 0% to 100%, provides a useful contribution by quantifying the (small) size of the impact of the TKE term.

A few weeks of simulation time may have been sufficient for future model development choices.  However, the goal of our report was to provide the first year-long assessment of wake effects on power production, which is a highly sought-after dataset for industry partners and stakeholders.

Thus in principle every value that they report in the abstract should be a range, but it is not. Plus, no TKE results are shown (I would like to see the equivalent of Figure 8 but for added TKE). In summary, this study focuses excessively and at the same not enough on TKE. An obvious recommendation would be to ask the team to conduct a new one-year run with 25% TKE and rewrite the paper to focus on those results and reduce the sensitivity analysis. But I think that this would be an excessive request, plus there is already a lot of value in the current runs.

We appreciate that the reviewer recognizes that another set of 25% TKE simulations is computationally infeasible. Due to computational limitations, we cannot run an additional year-long simulation.

My first recommendation is therefore that the results in the abstract and conclusions, which do not report a range (i.e., the range of results with 0% and 100% TKE) but are presented without explanation as one value (see comment below), be modified by either reporting always the range, or by using an interpolation based on the few days of 25% TKE results that the authors have already run (Fig. A1).

We have modified the abstract as follows to incorporate the range of values:

> "Using a series of simulations with no wind plants, one wind plant, and complete build-out of lease areas, we calculate wake effects and distinguish the effect of wakes generated internally within one plant from those generated externally between plants. **We also vary the amount of added turbulence kinetic energy (TKE) between 0% and 100% to provide some uncertainty quantification.** The strongest wakes, propagating 55 km, occur in summertime stable stratification, just when New England's grid demand peaks in summer. The seasonal variability of wakes in this offshore region is much stronger than diurnal variability of wakes. Overall, year-long wake impacts reduce power output **by a range between 38.2% and 34.1% (for 0%-100% added TKE)**. Internal wakes cause greater year-long power losses, **from 29.2% to 25.7%,** compared to external wakes, **from 14.7% to 13.4%**. The overall impact is different from the linear sum of internal wakes and external wakes due to non-linear processes. Additional simulations quantify wake uncertainty by modifying the added amount of turbulent kinetic energy from wind turbines, introducing power output variability of 3.8%. Finally, we compare annual energy production to New England grid demand and find that the lease areas can supply **58.8% to 61.2%** of annual load."

Further, the conclusions are modified similarly:
- We now report "The average yearlong power deficits at Vineyard Wind considering internal wakes and external wakes from the LA range between 38.2% (TKE_0) and 34.1% (TKE_100)."
- Text is rewritten to include the range by "Yearly averaged wake losses induce power deficits at Vineyard Wind from 38.2% (TKE_0) to 34.1% (TKE_100)".

What I mean is that the team could obtain a relationship between average power output (or wind speed deficit or whatever the parameter of interest is) with 0%, 25%, and 100% TKE from the few simulated days. This relationship does not seem to be linear from Fig. A1. An example of this relationship might be something like: the power at 25% TKE is the mean of that at 0% and 100% TKE, on average. Then use that relationship to report one value (per parameter) in the abstract and conclusions, that "fitted" to 25% TKE.

As the reviewer has pointed out, the relationship between parameters and the amount of added TKE is a nonlinear relationship and so we have chosen to provide the range of values as above.

The second recommendation is that the authors add a figure and discussion on the TKE distribution in the wakes with 0% and 100% TKE, like Fig. 8.

We have added a section for the results and discussion of TKE at the hub height in new Appendix E, similar to Figure 8 (new Figure 11).

**2. The calculation of the losses from external wakes may be incorrect**

From the abstract, the effects of internal wakes are reported to be -27.4% and the effects of external wakes are -14.1%. The sum of the two is -41.5%. However, the combined effect is reported to be -35.9%. This is problematic. At first sight, this discrepancy may be the result of the non-linearity of the wake processes. If so, all the authors need to change is to reverse the order of two sentences and add a few words in the abstract to explain it: "Internal wakes alone cause greater year-long power losses (27.1%) compared to external wakes (14.1%). When both are present, however, the mean year-long wake impacts reduce power output by 35.9%, which is lower than the sum of the two due to non-linear processes."

Yes, we have noticed and discussed this nonlinear behavior, and have added a sentence to the abstract to explicitly note this behavior.

However, I suspect that there might be a design issue in the way the power losses are calculated in Eq. (9) and (10). Aside from the unclear notation (see comment below), the denominator of the two equations is not the same and that may be why the discrepancy arises. Eq. (10) is correct because there is no double counting: there are no losses in the denominator and the internal losses are only in the numerator. In Eq. (9), however, there are internal losses in both the numerator and the denominator, and they are not equal. The internal losses are not equal in the VW and CA cases because, as upstream conditions change due to external wakes in the CA case, the internal wakes change too and therefore the internal wake losses do not "cancel out", there is still some influence from the internal wake losses. As such, the ratio in Eq. (9) does not quantify just external losses because it still contains the effect of internal losses; it quantifies a mix of internal and external losses.

I suggest that the authors report Eq. 10 first (Loss_internal). Then, they should replace P_VW at the denominator with P_NWF in Eq. 9, to obtain the total effect from internal and external wakes due to the CA areas (call it Loss_total). We know this value: it should be -35.9% (from l. 384). The effect of the external wakes then is the difference between the total losses and the value from Loss_internal:

Loss_external = Loss_total – Loss_internal = -35.9% - (-27.4%) = -8.5% (Eq. 11)

This way the denominator is the same and the individual values for external and internal sum up to the correct total.

Thank you for this suggestion. There are several different methods for calculating the wake impact, and we have supplemented an additional method for calculating external losses as the difference between the total and internal losses, via a new equation (11).

Power losses from external, internal, and the total wake effects are calculated from:

$$Loss_{external} = 100 - \left(\frac{P_{LA,CA}}{P_{VW}}\right) * 100\% \tag{9}$$

$$Loss_{internal} = 100 - \left(\frac{P_{VW}}{P_{NWF}}\right) * 100\% \tag{10}$$

$$LOSS_{total} = 100 - \left(\frac{P_{LA,CA}}{P_{NWF}}\right) * 100\% , \tag{11}$$

where $P_{LA,CA}$ is the power production at Vineyard Wind grid cells in the presence of wakes by either the LA or the CA, $P_{VW}$ is the power production in the presence of internal wakes from VW, and $P_{NWF}$ is the power production from coupling hub-height wind speeds to the power curve. These methods are performed separately by added TKE amount. **We note that the upwind conditions change in a LA or CA scenario, due to external wakes, which can modify the internal losses in the numerator of Eq. 9. Thus, we provide an alternative method for calculating the external power losses as the difference between the total losses and the internal losses:**

$$\boldsymbol{LOSS_{external} = LOSS_{total} - LOSS_{internal}} \tag{12}$$

**3. The stability classification is not adequate.**

The authors use a very simple classification for stability based on the value of L (Eq. 8). Neutral conditions are those with abs(L)>1000 m. This is inconsistent with the published literature, e.g., Gryning et al. (2007) and Sathe et al. (2011) used 500 m, Wharton and Lundquist (2012) used 600 m, Rajewski et al. (2013) used 400 m, Archer et al. (2016) used 500 m. In fact, too few neutral cases were found here, less than 2.5% of the time (p. 29 l. 595). I am unsure what to recommend here because there is not an "accepted" value of L for neutral conditions, but the authors need to assess the sensitivity of their results to a few values, at a minimum 500 m. This could possibly help with the previous inconsistencies in the areal extent and wake length, as days that were actually neutrally stratified may have been mixed in with days with other stabilities to obfuscate some of the relationships.

As the reviewer acknowledges, there is a wide range of thresholds that have been used to determine stability regimes and there is not an accepted value of L to demarcate the line between neutral and stable or unstable conditions. The threshold of 1000 m is consistent with the published literature as Muñoz-Esparza et al. (2012) use this cutoff for neutral conditions in the offshore environment. (Most of the references cited by the reviewer were for onshore conditions). Our finding that neutral stratification occurs 2.5% of the time is only for the CA simulations, which is a subset of August-September of 2019 and June-July of 2020. For the yearlong period, our original reported number is double this value, at 4.48% of the time.

Additionally, through discussion with other WRF modelers, we learned that the WRF-output Obukhov Length (which we were using in the original calculations) is not accurate because it is calculated in the timestep before the heat flux is calculated. We have recalculated the Obukhov

length directly using model-output variables at the same location (all figures and calculations incorporating stratification have been updated). The new percentages of occurrence for unstable, stable, and neutral conditions using a 1000-m cutoff are 48.4%, 46.3%, and 5.2%, respectively (originally 53.6%, 41.9%, and 4.5%,). Using a 500 m threshold, these percentages change to 44.3%, 44.4%, and 11.2%. We choose to maintain the |L|=1000 threshold because that is consistent with offshore work (Muñoz-Esparza et al., 2012).

**Minor Issues**

**4. The title needs improvements**

The title suggest that the wake impacts "on" the wind farm development will be studied. This is somewhat inaccurate, as the study is about the wake impacts on offshore wind power production or output, not on the development. Development is choosing the number of turbines or their specs or their layout, which are all fixed in this study; or, development can be how the wind farm installations grow/change with time. Either way, the development here is given (3 scenarios), what changes is the power output.

Also, the title mentions the "Mid-Atlantic" as the focus area, but technically speaking the Mid-Atlantic stops as far north as New York state. From the U.S. perspective, the Vineyard Wind project is not in the Mid-Atlantic and neither are the northeastern lease areas of RI or MA. According to Wikipedia, the following states are included in the Mid-Atlantic: Delaware, Maryland, New Jersey, New York, Pennsylvania, Virginia, West Virginia, and Washington DC. To non-U.S. readers, "Mid-Atlantic" could be the Equatorial zone, as the Atlantic Ocean extends between the two Poles. I don't have a good recommendation for an alternative, but perhaps "U.S." should be added in the title because the study focuses on the U.S. offshore areas after all.

Last, "annual" variability suggests that many years were studied to understand how the production changes from one year to the next. Instead, only one year was simulated here. Thus the variability studied here is monthly/seasonal and diurnal, but not annual.

We keep the siting and characterizations of wind turbines constant in our work and agree that the main focus is on power production. However, our use of Mid-Atlantic is consistent with the Bureau of Ocean and Energy Management terminology. Thus, we will not change this nomenclature. We agree that annual variability implies studying multiple years, and that we should clarify the U.S. focus. We have changed the title to "Seasonal Variability of Wake Impacts on U.S. Mid-Atlantic Offshore Wind Plant Power Production".

**5. Simplify naming**

There is no need to add "_only" to the name of the run with only the Vineyard Wind farm. Just call it "VW."

All instances of "VW_only" have been changed to "VW".

**6. Unclear notation in Eqs. (9)-(10)**

These equations have already been discussed at comment #2, here I am focusing on the notation only. Eliminating "_only" will help (comment #5). P_WV_waked is not defined and uses a notation that differs from that of all other subscripts. All the other subscripts refer to a specific run, whereas "waked" refers to, I believe, a subset of grid points. But the same subset of grid points was used for all other denominators and numerators, thus the confusion. Plus the term P_VW_waked refers to run CA, I believe. I suggest something like (not including my recommendation from comment #2 above):

$$Loss_{ext} = \left(1 - \frac{P_{CA}}{P_{VW}}\right) * 100\%$$

In the text below the equation then you specify that this equation is obtained from the grid cells over Vineyard Wind.

We have changed the notation to reflect the simulation type such that $P_{LA,CA}$ refers to the power production at Vineyard Wind grid cells when exposed to external and internal wakes by either the lease or call areas, to reduce redundancy of writing the same equation twice. $P_{VW}$ refers to power production at Vineyard Wind grid cells in the presence of internal wakes in a VW simulation. $P_{NWF}$ represents power production at Vineyard Wind grid cells from coupling NWF wind speeds to the power curve.

7. **L. 318-320 ("While here ... schemes"): this discussion is irrelevant and unnecessary here.**
This sentence is also redundant and has been removed.

8. **P. 17: some of these results are rather counter-intuitive**
if the TKE_100 runs produce weaker deficits and smaller wake areas, then the wakes should be shorter, whereas the authors report 58 km for TKE_100 and 55 km for TKE_0. The explanation provided is vague and unsupported (l. 346: "larger reduction in momentum aloft"??). The authors do not report exactly how the wake length was obtained. I suspect the method was somewhat empirical and in fact it is giving counter-intuitive results. I suggest that either the authors develop an objective and automated method for calculating the wake length and, if the inconsistency persists, they document and explain it; or that they remove any discussion of the wake length.

We appreciate that, at first, this finding may seem counterintuitive. However, turbulence from the turbines enhances vertical momentum transport from aloft down to within the wake (Gupta and Baidya Roy 2021). The enhanced TKE in a TKE_100 simulation transports more momentum into the waked zone, leaving slower wind speeds above the wind plant (Fitch et al. 2012; Siedersleben et al. 2020). Reduced wind speeds above the turbines then offer a weaker reservoir of momentum available for wake recovery further downwind, leading to slightly longer wake propagation distances. References have been added to support this argument. We explain how the wake length was obtained in Section 2.8 "Wake Identification".

9. **Improve Fig. A1.**

Replace 10-m wind speed with 140-m wind speed in Fig. A1. In all panels (except c), add the results from NWF to appreciate the magnitude of the impacts.

The wind speeds shown in Fig. A1 are indeed hub height wind speed. An error in the y-label lead to a cascading effect in the text which has been fixed.  NWF values have been added to the figure.

**10. P. 26-27: the discussion is unclear, the authors report "reductions" in several sentences, but it is unclear what is changing and what the reference is: are they discussing changes from TKE_0 to TKE_100 or from TKE_100 to TKE_0 or from NWF?**

- We now clarify "The wind speed reduction **during this time period** causes a corresponding decrease in turbulent transport of moisture."
- We now clarify "The reduction in heat flux **during this time period** causes 2 m temperatures to decrease and exhibit less variability by TKE amount, with a mean difference of 0.26 K between TKE_100 and TKE_0 (Fig. A1f)."

For the final occurrence of "reduction", it is obvious to the reader that we are referring to the same time period at this point, so to not add distracting information to a topic sentence, we leave this sentence as is: "The reduction in turbulent mixing lowers the PBL, regardless of TKE amount, to shallow heights between 30 to 80 m at 13:00 UTC (Fig. A1e)."

**10. Fig. D1: need a legend for the colors. Also, are these wind speeds or wind speed deficits? The caption indicates wind speed.**

We have added a color bar to Figure D1, included a color bar title delineating that the contours represent the wind speed deficit, and specified the "wake wind speed deficit" in the caption.

References

Archer, C. L., S. Wu, Y. Ma, and P. A. Jiménez, 2020: Two Corrections for Turbulent Kinetic Energy Generated by Wind Farms in the WRF Model. *Monthly Weather Review*, **148**, 4823–4835, https://doi.org/10.1175/MWR-D-20-0097.1.

Fitch, A. C., J. B. Olson, J. K. Lundquist, J. Dudhia, A. K. Gupta, J. Michalakes, and I. Barstad, 2012: Local and Mesoscale Impacts of Wind Farms as Parameterized in a Mesoscale NWP Model. *Monthly Weather Review*, **140**, 3017–3038, https://doi.org/10.1175/MWR-D-11-00352.1.

Gupta, T., and S. Baidya Roy, 2021: Recovery Processes in a Large Offshore Wind Farm. *Wind Energy Science Discussions*, 1–23, https://doi.org/10.5194/wes-2021-7.

Muñoz-Esparza, D., B. Cañadillas, T. Neumann, and J. van Beeck, 2012: Turbulent fluxes, stability and shear in the offshore environment: Mesoscale modelling and field observations at FINO1. *Journal of Renewable and Sustainable Energy*, **4**, 063136, https://doi.org/10.1063/1.4769201.

Siedersleben, S. K., and Coauthors, 2020: Turbulent kinetic energy over large offshore wind farms observed and simulated by the mesoscale model WRF (3.8.1). *Geoscientific Model Development*, **13**, 249–268, https://doi.org/10.5194/gmd-13-249-2020.

---

## Author Comment (AC2)

**Response to Mark Stoelinga**

Referee comments appear in black and author responses appear in blue.

We thank the reviewer for his thoughtful comments.

181-182: I think centered RMSE (cRMSE) is essentially the same as what I've heard and referred to as bias-corrected RMSE (or BCRMSE), in which you first calculate the mean model bias error, subtract it from all the model values, then calculate RMSE. And, I believe both are essentially equivalent to the standard deviation of the errors as well. All that is neither here nor there. However, I do think the sentence in lines 181-182 should be clarified, to say that "a value of 0 for cRMSE indicates that all values, *after removal of the respective model or measured means*, lie on the 1:1 regression line".

Thank you for the suggestion. We have added your proposed clarification to this sentence as "A value of 0 for cRMSE indicates that all values, with model bias removed, lie on the 1:1 regression line".

189 (paragraph): Might be good to show model versus measured mean shear exponent, a metric that the wind industry uses extensively and is highly familiar with its typical range of values.

Thank you for the suggestion. We have added a new Figure 8 and a discussion of model versus measured wind shear exponent, finding that lidar-derived exponents are in good agreement with past evaluations in the mid-Atlantic and that WRF-derived exponents are underestimated.

325-326: There is an interesting result in Fig. 8 that you do not comment on, which is similar to behavior other have seen and commented on (including, I believe, one or more of you in previous work, and myself). What I'm referring to is the opposite effect of TKE amount in the near-project versus distant wake environment. Within and near the project, behavior is intuitive: higher TKE dissipates wakes and leads to smaller waked wind deficits. However, farther away, as evidenced by the distance northeastward of the first (0.5 m/s) contour, as well as the area of this contour reported in the text, it is actually slightly farther (and covers more area) with TKE than without it. In other words, at distance, higher TKE actually helps wakes, whereas near or within the project it hurts wakes. I saw the same behavior, and I'm certain you and others have commented on it previously. Do you have any new insights into this behavior?

This comment was clarified and retracted by the reviewer in a later comment posted in the online discussion.

Appendix E. The authors and I have had discussions in the past about the nature of the noise seen in difference fields (turbines minus no turbines wind speeds). I'm not opposed to the idea that they are purely numerical; I agree that is the most likely explanation. However, I still consider it possible that even the distant differences are perhaps partly physical rather than numerical. They tend to occur in an unstable boundary layer or in convective scenarios. These scenarios are characterized by small-scale, high-amplitude, chaotic structures (convective cells) whose initiation locations are

random and probably sensitive to even the smallest perturbations, which may include very subtle and fast-moving gravity waves or other disturbance triggered by the presence of the turbines.  For the purpose of energy production, though, they are probably inconsequential because they tend to cancel each other out when averaged either spatially or temporally.

Apart from noise adjacent to the farms, we have observed noise also appearing far upwind of the turbines where the introduction of wind plants should make no discernible difference to the atmospheric state (tens of kilometers upwind of the induction zone, with little or no noise in the induction zone).  Even if gravity waves were involved here, gravity wave deflection should maximize close to the wind plants before dissipating, making it more likely that these features are numerical, but we agree that numerical noise is worth looking into in future studies.

---

## Author Comment (AC3)

**Response to Referee #2**

Referee comments appear in black and author responses appear in blue.

- line 68: While 12 MW turbines seem to be similar enough to the 13 MW turbines to be installed at Vineyard Wind, I wonder how realistic the assumption is for the other lease and especially the call areas, since those will be build later than Vineyard Wind. Also how sensitive are your results to the chosen turbine type?

We had several discussions with BOEM to determine the best turbine density and the most likely turbine nameplate rating.  At that point (Fall 2019), there was little to no knowledge of the actual nameplates to be installed at each lease area, either because it was unknown or proprietary, so a blanket 12 MW was chosen through these discussions.

Repeating these simulations with different turbine types is too computationally demanding and out of scope of this investigation. Other researchers have explored this sensitivity for shorter times periods (Golbazi et al. 2022), finding that the height of the turbine can impact the surface temperature impacts (their Figure 5). A sentence acknowledging this sensitivity has been added to the conclusion: "Further, different sizes of turbines may be installed in some of these regions, and the size of the turbine can influence the impacts of the turbine (Golbazi et al., 2022)."

- line 73 - 74: Why did you choose this period and not a regular calendar year? Do you run continuously or restart the model after a certain period?

We chose this time period due to the availability of lidar measurement data.  This clarification has been added: "NWF, VW, and LA simulations run from 01 September 2019 to 01 September 2020 *to capture a full year with available lidar measurement data*". We submitted multiple restarts each month to mitigate runaway error growth as mentioned in new Appendix F.

- line 79 - 81: You don't mention section 3

These are typos where Section "n" incorrectly states Section "n+1," and these have been fixed: "Section 3 discusses variability in stratification, wakes, and power production. Section 4 concludes the work and offers recommendations for future work."

- line 95 ff: Please also provide the WRF option number in addition to the reference

The WRF namelist options for all parameters used in the study are provided in the sample namelist.input, which we provide under the Section "Code and Data Availability."

- Figure 1 caption: last sentence, double mentioning of "red"

The double mentioning has been removed. The last caption sentence now states "E05 (triangle) and E06 (diamond) floating lidars are shown in red."

- line 107 - 114: How realistic is the assumption of regular layout within the areas? To reduce internal wake effects, the turbines might be better placed in an irregular layout

While the goal of minimizing wakes might suggest an irregular layout, minimizing wakes is not the only goal of these wind farms. Cooperative use of these regions requires accommodating other uses. Therefore, our layouts for this work were determined after multiple discussions with BOEM and industry partners. The use of regular layouts in the wind energy areas is realistic, and in fact was requested by other users of this area, notably fishermen and fisherwomen, who request predictable navigable corridors with turbine installations in fixed east-to-west rows and north-to-south columns. For example, https://www.heraldnews.com/story/business/2020/01/07/fishermen-at-odds-with-developers/1945689007/ discusses how a mariner's group supported a regular layout (albeit with even more navigable corridors than proposed here).

- Figure 2a: Where is region 1? Either start numbering at 1 or mention region 1 as below cut-in

The labels for the different regions of the power curve is not something that we developed, but are widely used in the wind energy literature (specifically the controls community) (e.g. Sohoni et al. 2016). We have clarified that "No power is produced in region 1 of the power curve, from 0 m s$^{-1}$ to cut-in wind speed (3 m s$^{-1}$)."

- Figure 3: It would be nice to relate the wind rose to the "regions" in Figure 2. E.g. green could be capped at 11 (below rated power) and one color could be used for region 3. Also "m/s" should be formatted with negative exponents according to the guidelines

Thank you for the suggestion, but we find it is better to retain the granularity in wind speed so as not to limit findings.

- Line 179: Does removing the periods induce a bias? E.g. are they related to the same period / stability category?

Less than 10% of data is removed. The greatest percentage of data is removed during stable stratification, followed by unstable, and neutral conditions at both the E05 and E06 lidars. New table 2 has been added as follows:

*Table 1. Percentage of data removed at 140 m due to NaN values.*

|  | Unstable | Stable | Neutral |
|---|---|---|---|
| E05 | 1.35% | 6.44% | 0.33% |
| E06 | 3.64% | 9.48% | 0.62% |

- Line 170 - 183: Why do you choose these metrics? How do they compliment each other?
16.440.

The metrics are commonly used in these types of studies. We selected these validation metrics following (Optis et al. 2020), who asserted that these four are key for model-based wind resource assessment. These metrics have been used in subsequent similar investigations (e.g., Pronk et al. 2022). These metrics offer different insight into model performance.  For instance, a model may overestimate wind speeds but correctly capture the diurnal cycle, in which case bias would be large but correlation would be strong.  Such a setup could present less difficulty for hour-ahead power forecasting, where wind speeds could simply be derated for accurate results.  Alternatively, the model could resolve accurate mean wind speeds when compared to lidar measurements but resolve fast wind speeds too frequently.  The resulting skewness in the distribution would be captured by the Earth Mover's Distance.  Essentially, there are many ways to evaluate if a model is performing well, either temporally, by means, by distribution, etc., and these metrics capture a wide variety of model performance to guide future industry and research decision making.

- Line 189 - 196: Following up on the previous comment, how do you interpret the results that you obtain for the different error metrics? E.g. something along those lines: "the results correlate well in time but have an offset ...". This should also be discussed for the stability based analysis

We provide an interpretation of each metric two paragraphs above where each metric is introduced.  The level of detail of this description has been increased: "A CC value of one indicates a perfect correlation between NWF and lidar values.  A value of 0 for cRMSE indicates that all values, with model bias removed, lie on the 1:1 regression line.

A cRMSE value greater than 0 indicates the distance of residual points from the regression line. Negative biases indicate an underestimation from WRF while positive biases indicate overestimation. A value of 0 for EMD indicates that probability density functions from each data source are equivalent. A positive EMD indicates that the NWF wind speed distribution must shift towards lower values to match the lidar distribution."

- Line 203: You do not describe, which metric you use to classify stability. I assume you are using the same that you use in section 2.7. Consider to move section 2.7 before section 2.6 so that the reader doesn't need to guess.

We agree that it makes sense to build into the validation by providing discussion of the observations, stability classification, and then their combination for the validation. The section order has been switched as suggested.

- Section 2.7: You discuss in Appendix B that the Obukhov length only represents the surface characteristics. Why do you stick to this classification? Also Appendix B should be referenced in section 2.7. Have you estimated the sensitivity of your results to this particular metric? Platis et al. (2021) suggest that depending on the stability metric the results can vary quite a lot (Platis, A., Hundhausen, M., Lampert, A. et al. The Role of Atmospheric Stability and Turbulence in Offshore Wind-Farm Wakes in the German Bight. Boundary-Layer Meteorol 182, 441–469 (2022). https://doi.org/10.1007/s10546-021-00668-4)

We tested the sensitivity of stability metrics between the Richardson number and the Obukhov length and found differences in the percentages of occurrence of unstable, stable, and neutral stratification. We chose the Obukhov length following Archer et al. (2016), who argued that it was a suitable stability metric in the mid-Atlantic offshore region. We have added sensitivity to our choice of a 1000-m cutoff for neutral conditions by adding the percentages of occurrence for each stability class using a 500-m threshold. Also, we have improved the accuracy of the stability metric by calculating the Obkuhov length directly instead of using the WRF-generated values.

"The mean unstable, stable, and neutral percentages of occurrence at Vineyard Wind are 48.4%, 46.3%, and 5.2%, respectively, for the period 01 September 2019 to 01 September 2020, using a 1,000-m threshold for neutral conditions. Using a 500-m threshold for neutral conditions, the percentages are 44.3%, 44.4%, and 11.2%."

- Line 249 - 251: This wake length estimation seems to be too simplified: What about wake turning? What about other wind directions? Arguably the wind rose does show predominant winds from south-west, but other wind directions are also present. In

those cases the wake length will be underestimated. To understand your method it would help to draw the line in figure 1.

This wake estimation method compares the wake strength at the same point downwind between unstable and stable conditions, and is consistent with approaches used in the literature (i.e., Rybchuk et al. 2022). Altering the defined downwind line to heterogeneous wake turning or different wind directions would no longer yield a consistent comparison because more factors would be changing than just the stratification.

- Line 270: Reference Appendix E

A reference to the Appendix section (new Section F) has been added: "Power output from VW, LA, and CA simulations are averaged in hourly windows at grid cells containing Vineyard Wind turbines to reduce the effects of numerical noise (Appendix F)."

- Line 304 - 305: This sentence is difficult to understand. Please revise.
This sentence has been revised to "The same pattern occurs elsewhere throughout the OCS because diurnal variability in stratification is weaker than the seasonal cycle".

- Line 311 - 319: These results could be much more neatly presented in a table instead of text form.

Thank you for the suggestion; new Table 3 summarizing the results of this paragraph has been added:

*Table 2. Wake wind speed reduction by stratification and TKE amount.*

|  | Unstable TKE_100 | Stable TKE_100 | Unstable TKE_0 | Stable TKE_0 |
|---|---|---|---|---|
| Wind Speed Deficit | $-1.5$ m s$^{-2}$ | $-2.8$ m s$^{-2}$ | $-1.8$ m s$^{-2}$ | $-3.1$ m s$^{-2}$ |
| Normalized deficit | 16% | 25% | 19% | 27% |

- Line 325: "although areal coverage is larger from reduced wind speed replenishment". What do you mean by this?

Because turbulence is weaker in TKE_0, there is less vertical transport of momentum into the waked region from aloft. Accordingly, the spatial extent of wakes grows larger

when compared with TKE_100: "although areal coverage of the wake is larger due to weaker turbulence-induced wind speed replenishment from aloft."

- Line 326 - 327: According to the numbers that you present for stable stratification the waked area is actually larger for TKE_100 (16404 km²) compared to TKE_0 (16060 km²). This contradicts with your conclusion in this sentence. Please clarify.

Thank you for pointing this out. This sentence has been revised to state that the largest spatial area of wakes occurs in stable conditions in TKE_100.

- Line 341 - 345: Again a table would facilitate a comparison between scenarios

New table 4 has been added underneath the text for easier comparison:

*Table 3. The wake wind speed deficit, spatial extent, and downwind propagation distance by added TKE amount.*

|          | Wind Speed Deficit      | Spatial Extent          | Propagation Distance |
|----------|-------------------------|-------------------------|----------------------|
| TKE_100  | $-2.2$ m s$^{-1}$       | 13,040 km$^2$           | 43 km                |
| TKE_0    | $-2.5$ m s$^{-1}$       | 13,268 km$^2$           | 41 km                |

- Line 349: You reference D1 here, but D1 only shows TKE_100 and thus the differences due to different TKE levels cannot be assessed.

Figure D1 facilitates comparison between stability conditions. This sentence has been clarified accordingly: "The same pattern exists for CA wakes (**Error! Reference source not found.**)."

- Figure 9: Sub-figure titles are (a) for all

Thank you for catching this typo. Sub-figure titles for new Figure 12 have been revised to include (b) and (c).

- Line 361 - 362: Can you provide the power losses averaged over the four month for VW_only and VW_waked for comparison?

The external power losses from the lease areas during the four stable months have been added: "Considering external wakes from the LA at TKE_0 (Eq. 9), the average yearlong power deficit at Vineyard Wind is 14.7% (Fig. 12a) and increases to 15.7% considering only the four stable CA months." The internal losses over the four stable months have also been added: "During the four CA months only, the deficits increase to 36.9% and 32.9%, respectively."

- Section 3.3.1: You show also diurnal variations, but these are not discussed. Please add this.

We have added clarification with the following: "While wake-induced losses vary somewhat across the diurnal cycle, there is no discernible pattern.  The ocean's large heat capacity suppresses daytime heating which limits changes in stratification, and by extension, the magnitude of changes in wake losses."

- Line 383 - 398: It seems a bit counter-intuitively that losses are not additive, i.e. internal losses + external losses != total losses. While the proposed loss estimates (9) and (10) do make sense, they do not share the same reference (P_VW_only vs P_NWF), which makes it more difficult to compare.

The total wake losses are not additive between internal and external losses, primarily because of nonlinear interactions but also because the denominators are different. We have added an alternative method for calculating external losses, represented as the subtraction between total and internal losses, which share the same denominator in new equation (12).

- Line 402: I understand the energy demand estimates are taken for present day? Are there estimates on how the energy demand will change until CA and LA are build?

New York ISO provides several estimates for future energy demand which vary considerably by the scenario type.  A high-load future demand scenario would represent greater implementation of electrification, such as electric vehicles and wintertime heating, and slower adoption of grid independence, such as on-site solar generation. The low-load scenario essentially represents the opposite. (https://www.nyiso.com/documents/20142/2226333/2021-Gold-Book-Final-Public.pdf/b08606d7-db88-c04b-b260-ab35c300ed64).  The difference between the high-load and low-load scenarios could reach a spread of about 100,000 GWh by 2053 (https://www.nyiso.com/documents/20142/37320118/2023-Gold-Book-Forecast-Graphs.pdf/ad7db043-ea01-dc3b-b917-ca4cd1d7cd8f).  Reporting the amount of demand that the LA and CA layouts could supply in the future would inherit a large amount of uncertainty, which is why we choose to compare supply with current demand.

- Line 408 - 409: Could you add another line in figure 11 representing the stability conditions. This would make it easier to see that the power production is indeed more closely linked to hub-height wind than stability.

Unfortunately, a timeseries plot of stability conditions at the same granularity (as seen below) does not easily facilitate comparison. This is why we have chosen to show stability with longer temporal averages using bar charts and grid plots.

[Figure]

*Timeseries of the Obukhov Length over the yearlong period.*

- Line 424: Reference figure 2 here again to remind the reader of the definition of region 2 and 3.

A reference to the power curve in Figure 2a has been added: " These differences are small at slow wind speeds, because little momentum is available for wake recovery, and at faster wind speeds within region 3 of the power curve (11−30 m s$^{-1}$) where wind speed changes do not affect power production (**Error! Reference source not found.**a)."

- Figure 13 caption: "black dots indicate turbine locations": suggesting to add "in TKE_0 and TKE_100", since in NWF they are not included

The caption (for what is now Figure 16) has been rewritten to "black dots indicate turbine locations in VW TKE_0 and TKE_100".

- Line 508: It would be interesting to discuss, how the difference due to added TKE amount compares to the difference due to different PBL schemes. You mention Rybchuk et al. (2022) at some places through the paper, but don't compare the effects due to PBL schemes and added TKE amount directly.

While we would also find this an interesting discussion, direct comparisons are not possible as Rybchuk et al. focus on idealized scenarios and the present study is for real

scenarios. We are currently working on winning funding to carry out a more direct comparison of the present work with simulations with the 3DPBL scheme. Detailed discussion is not within the scope of this work, and so we refer the readers to Rybchuk et al. (2022).

- Line 537: What do you mean by "the differences … are precise"?

This sentence has been rewritten to "The sequencing of power production driven by TKE amount remains consistent, namely that the differences always progress from TKE_0 to TKE_25 to TKE_50 to TKE_75 to TKE_100." "Consistent" is used as a lead into the next sentence where we discuss that power production values are typically bookended by TKE_0 and TKE_10.

- Appendix A: The mixture of discussion on variability due to added TKE amount and the special case during calm winds between 12:00 and 15:20 on 12 July is confusing. These two aspects should be kept separate.

We attempted to rewrite the section on the special case-study period separately as the reviewer suggested, but upon reading, we determined it was even more confusing to bounce back and forth between meteorological variables (wind speed, heat flux, etc), and decided that in this appendix, we will keep each idea in its own respective paragraph.

- Line 555: the first sentence is a bit difficult to understand. The difference between TKE_0 and TKE_25 seems to be more than 15 to 20 m

15-20 m here refers to the actual (very shallow) boundary layer height for a specific time period and not the difference between runs.  This sentence has been rewritten to "The reduction in turbulent mixing lowers the PBL, regardless of TKE amount, to very shallow heights between 15 to 20 m from 12-15:20 UTC (**Error! Reference source not found.**e)."

- Line 565: The way you reference figures is sometimes confusing to me. For instance, I would reference Figure B1 here as "stratification at the E05 and E06 (Fig. B1) lidars exhibits similar seasonal variability to Vineyard Wind (Fig. 6)". Since vineyard wind is shown in Fig. 6 and not in Fig. B1. Please also check other parts of the manuscript. Note also that you wrote "E05" twice.

Thank you for pointing this out.  A figure reference should go directly after the point being made. This recommendation has been implemented.

- Figure D1: Colorbar is missing; is the upper row just a zoom of the lower row?
Yes, it is just a zoomed in version of the figure. We have added the explanation that "The upper row is zoomed in to increase granularity" in the figure caption. A colorbar has also been added.

- Figure E1: "at which the map occurs" -> suggestion "of the map"

The caption text has been revised according to the suggestion in new Figure F1: "The gray vertical line shows the time stamp of the map."

- Line 643 - 646: Difficult to understand. What do you mean by "poses a threat to power estimations". I don't understand the contrast "although ..., we show noise occurring in the SE ..." and why this "underscores the point that ... should only show differences within the wake". Please clarify.

- The first sentence has been clarified to: "Noise occurring in grid cells containing turbines could undermine power estimation accuracy and we observed noise occurring in the southeastern portion of the domain."

- The second sentence was changed to be more concise: "Subtraction of wind speeds between simulations with variable TKE amounts should only show differences within the wake, and such differences are a result of noise."

- Line 660: Is there a link missing for "OpenEI_link"?
We are still working on getting the data ported for public access. A url will be inserted here once the data is uploaded.

- Line 715: Missing DOI
A DOI has been added

- Line 717: Missing URL
A url has been added

- Line 839: Missing URL
A url has been added

- Line 844: Missing DOI
A url has been added

References

Archer, C. L., B. A. Colle, D. L. Veron, F. Veron, and M. J. Sienkiewicz, 2016: On the predominance of unstable atmospheric conditions in the marine boundary layer offshore of the U.S. northeastern coast. *Journal of Geophysical Research: Atmospheres*, **121**, 8869–8885, https://doi.org/10.1002/2016JD024896.

Golbazi, M., C. L. Archer, and S. Alessandrini, 2022: Surface impacts of large offshore wind farms. *Environ. Res. Lett.*, **17**, 064021, https://doi.org/10.1088/1748-9326/ac6e49.

Optis, M., N. Bodini, M. Debnath, and P. Doubrawa, 2020: *Best Practices for the Validation of U.S. Offshore Wind Resource Models*. National Renewable Energy Lab. (NREL), Golden, CO (United States),.

Pronk, V., N. Bodini, M. Optis, J. K. Lundquist, P. Moriarty, C. Draxl, A. Purkayastha, and E. Young, 2022: Can reanalysis products outperform mesoscale numerical weather prediction models in modeling the wind resource in simple terrain? *Wind Energy Science*, **7**, 487–504, https://doi.org/10.5194/wes-7-487-2022.

Rybchuk, A., T. W. Juliano, J. K. Lundquist, D. Rosencrans, N. Bodini, and M. Optis, 2021: The Sensitivity of the Fitch Wind Farm Parameterization to a Three-Dimensional Planetary Boundary Layer Scheme. *Wind Energy Science Discussions*, 1–39, https://doi.org/10.5194/wes-2021-127.

——, ——, ——, ——, ——, and ——, 2022: The sensitivity of the fitch wind farm parameterization to a three-dimensional planetary boundary layer scheme. *Wind Energy Science*, **7**, 2085–2098, https://doi.org/10.5194/wes-7-2085-2022.

Sohoni, V., S. C. Gupta, and R. K. Nema, 2016: A Critical Review on Wind Turbine Power Curve Modelling Techniques and Their Applications in Wind Based Energy Systems. *Journal of Energy*, **2016**, e8519785, https://doi.org/10.1155/2016/8519785.

---

## Referee Report (RR1)

**Second review of "Seasonal Variability of Wake Impacts on U.S. Mid-Atlantic Offshore Wind Plant Power Production" by Rosencrans et al., submitted to WES**

The author have done an excellent job at addressing my concerns. I only a few minor requests.

1. The 25% TKE correction value is the default in WRF. It does not matter that only one study to date (Archer et al. 2020) has supported it. What matters is that *everybody* is using that value unless they have extensive knowledge of the issues associated with it and change it manually. I disagree that "The 0% added TKE is more similar to the impact in the Volker et al. parameterization which has been used in several studies." Volker et al. used a totally different approach for TKE and therefore I disagree that they should be cited in support of 0% TKE. Other studies have used 0% TKE, but only as sensitivity. One of the first papers by Fitch et al. demonstrated that adding 0% TKE was indeed inaccurate for example. We all know that some TKE is indeed added by the turbines and therefore 0% is not a representative value. 100% is the old default and there is value in using it. But not having anything in between is not good. I wish that the authors had done some runs at 25%, but it's too late for that. But the sentence in the abstract: "We also vary the amount of added turbulence kinetic energy (TKE) between 0% and 100% to provide some uncertainty quantification" is untrue. You did not "vary" the added TKE between the two extremes, you only used the two extremes. Thus the sentence should be rephrased as "To provide some uncertainty quantification, we tested two values of added TKE: 0% and 100%."

2. I suggest a small change in the order of the equations for the losses, to improve readability (shorter acronyms) and be more consistent (no mix of Loss and LOSS):

$$Loss_{tot} = 100\% - \left(\frac{P_{LA,CA}}{P_{NWF}}\right) \times 100\% \ (9)$$

$$Loss_{int} = 100\% - \left(\frac{P_{VW}}{P_{NWF}}\right) \times 100\% \ (10)$$

$$Loss_{ext} = 100\% - \left(\frac{P_{LA,CA}}{P_{VW}}\right) \times 100\% \ (11)$$

$$Loss_{ext} = Loss_{tot} - Loss_{int} \ (12)$$

3. I disagree that L=1000 m is a good choice here. The authors provide one reference only for it, Munoz-Esparza et al. (2012), which I am not familiar with. I provided 5 or 6 references for shorter values and recommended 500 m, which is in the ballpark of all of them. Dismissing them because not all of them are offshore is not convincing; not to mention that at least one, Archer et al. (2016), was offshore, more recent than Munoz-Esparza et al. (2012), and obtained from measurements in the Nantucket Sound, which is in the area of interest here. Lastly, using L=500 m the authors obtained a more typical frequency of neutral cases (11.2% versus the previous low value of 4.5%). As such, I have to insist that the calculations be modified using L=500 m.

---

## Author Response (AR2)

**Referee 1**

Referee comments appear in black and author responses appear in blue.

This paper is well-written, generally well designed and deserving of publication. I noticed that there have been previous review rounds, but I was not involved in those. However, I would like to raise some points that, in my opinion, should be addressed to enhance the paper's quality.

We thank the new reviewer for assisting with the review process and providing feedback to enhance this research.

1. Firstly, it would be beneficial to provide more information about the model's grid setup. The distance between the outer edges of the small domain and the outer edges of the large domain is rather small, and I wonder if the lateral nesting zone of the small domain is not in the spatial spin-up zone of the larger domain. Even though the model setup relies on a previous study, it would be good to clarify this.

While it appears that there is not much distance between the inner and outer domain boundaries, this is a very large domain. There are 20 cells between the outer and inner domain boundaries, which allow 4 waves to be resolved before entering the inner-most domain. We acknowledge the rule of thumb that the inner domain boundaries should be at a distance of roughly 1/3 of the outer domain's total extent, but our setup is 4 times the minimum requirement.

We have modified the text on lines 108-110 to: "Two nested domains comprise 6 km and 2 km horizontal resolutions (Pronk et al. 2022; Xia et al. 2022; Bodini et al. 2023; Redfern et al. 2023), respectively, *and the inner nest begins 20 grid cells into the parent domain* (Figure 1)."

2. I hold the opinion that using (bias-corrected) Root Mean Square (RMS) and correlation as metrics for evaluating model performance may not be the most suitable choice. Even though spectral nudging is applied in the outer domain, the flow in the inner domain can evolve freely. A small deviation in a weather system's position or timing could result in a double penalty effect. Therefore, for this type of studies, it is crucial to focus on getting the statistics right, rather than precisely timing weather systems. In my view, bias and quantification of distribution metrics are more relevant for such studies. I recommend a brief discussion of this issue when describing the evaluation metrics.

We believe that reporting more metrics than fewer metrics benefits a wider readership. We have added a discussion of how these validation metrics benefit model validation on new lines 237-240:

"Each of these metrics provides different insight into the performance of the model. For instance, the correlation coefficient illuminates how well the model captures the timing of weather systems and diurnal variability. EMD emphasizes the difference between distributions but not the timing. Bias captures the difference between measured and modeled values. Finally, cRMSE describes the random component of error."

3. I would argue that there should be only one definition for external loss. The total loss can be accurately represented by equation 11, as it is defined based on the reference no-wind farm simulation. Similarly, the internal loss for only the Vineyard Wind farm is correctly calculated, as it also involves dividing by the no-wind farm simulation. However, I have reservations about the correctness of equation 9. Your reference here is the power from the Vineyard Wind, rather than the no-wind farm simulation. The definition of external losses should be the difference between the run that includes all wind farms (combines internal and external loss) and the run with only the Vineyard Wind Farm (internal loss), divided by the power from the no-wind farm simulation (which is the reference). This is equivalent with Equation 12.

While we appreciate the reviewer's request for a streamlined definition for external loss, our experience in presenting this work in several industry and academic venues suggests that different communities request different definitions of external loss. Each of these definitions can be justified, and so we feel that it is important to include multiple measures. We have received requests for the loss definition as defined in the former equation 9 (new equation 15). We thus choose to present multiple definitions, clearly defined, so that readers can use the definition most suited to their interests. We believe it is helpful to include as an alternative method for calculating the external power losses.

4. The averaging of percentual power loss might be incorrect: I would argue that the percentual power losses, is the total power loss over a period considered (could be total year, or the January 0-1UTC periods), divided by the power of the reference situation (no wind farm) over the same period. However you describe that first you calculate the percentage loss over an hourly window and then these percentages are averaged. This is not the same.

Yes, we agree that using the percentage losses over an hourly window would not be the same. We choose to average over an hourly window first before all subsequent calculations to reduce the effects of numerical noise.

5. In many climatological studies, the Perkin Skill Score is utilized to compare distributions from climate models for various variables. Both the Earth Mover's Distance (EMD) and the Perkin Skill Score are closely related metrics. It would be valuable to include a brief description of how those two compare. Personally, I find the Perkin Skill Score very intuitive because it is dimensionless and assesses the overlap between distributions.
We have added the following text to the methods section on lines 459-478:

We calculate profiles of the Perkins Skill score ($PSS$) (Perkins et al., 2007) between NWF and lidar wind speeds. Wind speeds are considered at 20 m height intervals from 20 m to 200 m. Each wind speed timeseries is subset by all timestamps with unstable, stable, and neutral stratification, respectively. After subsetting, timestamps where lidar observations return NaN are removed from both lidar and NWF timeseries. At each height, the probability distribution functions of wind speeds are binned at 0.2 m s$^{-1}$ intervals and normalized such that the

frequencies add to unity. The minimum frequency between modeled and observed values for each bin is stored, and the resulting stored values are summed to calculate the score (Eq. 12):

$$PSS = \sum_{i=1}^{n} \min(C_{NWF}(z), C_{lidar}(z)) \tag{12}$$

where $n$ is the number of bins, $C$ is the count of normalized values in a bin, and $z$ is the height. A PSS of 1.0 suggests perfect overlap of the two distributions.

[Figure]

Figure 9. Vertical profiles of the Perkins Skill Score by stratification. at the E05 (teal) and E06 (orange) lidars subset by stratification (US = unstable, ST = stable, NT = neutral).

Profiles of PSS between NWF and lidar observations of wind speed vary by location and stratification. Performance is generally best in unstable conditions at both E05 and E06 lidar

locations with a mean value of 0.93. Performance is next best in stable conditions, starting around 0.90 at the surface and increasing to 0.93 at 120 m at E05. At E06 in stable conditions, PSS reaches a maximum value of 0.93 at 100 m. Neutral conditions exhibit worse PSS and larger spread by location. AT E05, PSS minimizes at 0.85 at 160 m and maximizes around 0.88 at 60 m. At E06, PSS scores minimizes at 0.87 at 80 m and maximizes at 0.89 at 140 m.

I wonder how the EMD, which is expressed in meters per second, depends on the wind speed itself. Do distributions with higher wind speed also have higher EMD even though the overlap is similar? Does this influence the comparison between stable and unstable that do have different wind speeds?

Distributions with faster wind speeds will not have a higher EMD value. For instance, adding 100 m s$^{-1}$ to every WRF and lidar value, (i.e., keeping the distributions the same but shifting them towards faster values) and recomputing the EMD results in the same EMD value. So, for instance, faster wind speeds in stable stratification will not cause the EMD to be larger, because the same amount of "work" would be required to move modeled data points onto the observed data points.

6. In your analysis about the different stability classes, you mix the effect of the wake with the fact that the wind rose is likely rather different during the different stability classes. I would appreciate a clearer distinction between the effects caused by variations in wake behavior and those resulting from different wind patterns. To analyse this, you have to include wind roses for both unstable and stable conditions. I expect substantial differences, which could be attributed to seasonal variations. To gain deeper insights, stratifying the data according to wind directions might be a valuable addition.

Thank you for the suggestion. We now assess the wake propagation distance to the southeast of the RIMA block in unstable conditions to account for the predominant northwesterly winds.

We have modified Figure 3 to show the relationship between stability and wind direction:

[Figure]

We have modified the methods section to describe the changes on new lines 483-488: "Because wakes typically propagate to the northeast during stable conditions (Figure 3), we calculate the propagation distance of wakes along a line extending northeast of the RIMA block (Figure 1) and report the distance along the line where wake wind speeds reach a threshold. In unstable conditions the prevailing wind direction is northwesterly (Figure 3), so we assess the wake propagation distance to the southeast instead. The threshold of $-0.5$ m s$^{-1}$ is chosen following Golbazi et al., (2022); Rybchuk et al., (2022)."

We now report wake propagation distances of 3.7 km (TKE_100) and 5.9 km (TKE_0) in unstable conditions on new lines 652 and 654.

7. I find the Turbulent Kinetic Energy (TKE) coefficient sensitivity study interesting and informative. However, I would argue that it does not fully represent uncertainty quantification. There are numerous other uncertain parameters, particularly in wind farm parameterization, which require a more comprehensive approach for uncertainty quantification. I suggest to remove the term uncertainty quantification in abstract and conclusions. You can add a discussion of how this would be a first step towards uncertainty quantification in a discussion.

We agree that a more extensive uncertainty quantification could be considered, but the TKE coefficient receives considerable attention in the literature and therefore addressing its bounding values (0% and 100%) provides a range of variation in wakes due to this parameter

and how it affects boundary-layer dynamics and therefore more uncertainty quantification than has yet been addressed. The reviewer's suggestion of emphasizing the "first step" towards a more complete uncertainty quantification is a good one, so we incorporate this language into the abstract and conclusions. New text in the abstract is as follows:

"We also provide a first step towards uncertainty quantification by testing the amount of added turbulence kinetic energy (TKE) by 0% and 100%. We provide a sensitivity analysis by additionally comparing 25% and 50% for a short case-study period."

New text in the first sentence of the conclusions (line 831) is as follows:

"This modeling study assesses the variability of wake effects across the mid-Atlantic OCS based on yearlong simulations, including a first step towards uncertainty quantification and approaches for distinguishing internal and external wake effects."

8. Appendix A, in my view, is very interesting but not yet mature enough for inclusion in the scientific paper. While it is interesting to study the effect of Turbulent Kinetic Energy (TKE) on surface heat and moisture flux, as well as planetary boundary layer heights, it is a different topic that would benefit from additional model evaluation, particularly concerning those variables.

We appreciate that there are other parameters that must be explored for an in-depth sensitivity analysis. The findings shown in Appendix A can provide context for additional questions and research into the sensitivity by TKE amount which has been requested by previous reviewers. This case-study analysis can lay the groundwork for future development. The placement of this material in an appendix rather than the main paper emphasizes that it is supporting material rather than full mature analysis, and so we choose not to remove the section.

9. It would be helpful to provide references or an explanation of the procedure for scaling the 12 MW turbine to the 15 MW turbine.

This scaling was performed by Beiter et al. (2020). We have made this point clearer and have added a brief description and reference on new lines 142-146:
"For our simulations, we parameterize 12 MW turbines *which are* scaled *by Beiter et al., (2020)* from a 15 MW reference turbine with a 138 m hub height and 215 m rotor diameter. *The power and thrust coefficient curves were held constant from the 15 MW machine. The rotor diameter was scaled to maintain a specific power of 332 W m$^{-2}$, which is the same as the reference 15-MW turbine. Then, the hub height was determined such that a 30-m gap was maintained between the lower bound of the rotor tip and the sea surface.*"

10. I wonder how the wind direction statistics have been conducted, especially given that wind directions range from 0 to 360 degrees. A brief explanation or clarification in this regard would enhance the reader's understanding.

We have added the following text to clarify wind direction statistics on new lines 251-263:

The circularity of wind direction must be accounted for in statistical calculations. For example, computing the average between 359° and 1°, using a typical arithmetic mean, would result in 180°. However, the mean wind direction between those two values should be 360°. The SciPy (Virtanen et al. 2020) and Astropy (Price-Whelan et al. 2022) Python packages offer convenient functions which allow the user to calculate statistics for a circular variable by passing in the lower and upper bounds, in this case 0° and 360°, respectively. We calculate the mean and standard deviation of wind direction using the SciPy "circmean" and "circstd" functions, respectively, and the correlation coefficient using the Astropy "circcorrcoef" function. The cRMSE for wind direction is then calculated using Eq. 9:

$$cRMSE = \sqrt{circmean\left(180° - \left|\left|\left(D_{WRF_i} - \overline{D_{WRF}}\right) - \left(D_{lidar_i} - \overline{D_{lidar}}\right)\right| - 180°\right|\right)^2} \quad (9)$$

where $D$ is wind direction and $\overline{D}$ is the circular mean of wind direction. Bias is calculated similarly to Eq. 6, except that differences between NWF and lidar values that are less than −180° have 360° added and differences greater than 180° have 360° subtracted (Eq. 10):

$$x = \begin{cases} x + 360° & \text{for } x < -180° \\ x - 360° & \text{for } x > 180° \end{cases} \quad (10)$$

where $x$ is the $(D_{WRF_i} - D_{lidar_i})$ difference.

11. Regarding Figure 16 and the related description, I am unsure of the significance of the difference between TKE=%100 and TKE=0%. This difference could potentially be attributed to variations in the timing or positioning of weather systems unrelated to the parameterisation. The explanation of Kelvin-Helmholtz instabilities appears speculative and requires a more detailed analysis. I recommend the removal of this analysis from the paper unless more substantial evidence and analysis can be provided.

While we appreciate the concern, the timing of the gust front is dictated by the initial and boundary conditions (which are the same for both simulations).

12. P20: the maximum average wake wind speed deficit: it is unclear what this refers to: is it the maximum deficit in space? Is this within the wind farm or outside? And why do you divide by the average hub-height wind speed to get a percentage rather than obtain the maximum percentual average wind speed deficit? This is not necessarily the same. Would be good to add some clarifications.
Yes, the maximum average wake wind speed deficit is the maximum deficit in space for the mean wakes shown in the figures. The maximum average wake occurs inside the wind farms. Our goal is to show the relationship between the maximum deficit in space and the typical wind speed, as opposed to how large the percentual reduction is when averaged over every timestamp.

We have added clarification on new lines 612-613:
"Here, we categorize wakes by the maximum wind speed deficit *in space*, the spatial extent, and the downwind propagation distance."

We have also added clarification on new lines 614-615: "The maximum average wake wind speed deficit *occurs within the wind plant areas* and intensifies from −1.5 m s$^{-1}$ to −2.8 m s$^{-1}$, moving from unstable to stable conditions for TKE_100 (Figure 11a,c)."

**Referee 2**

Referee comments appear in black and author responses appear in blue.

The authors have done an excellent job at addressing my concerns. I only have a few minor requests.

We thank the reviewer for reading the modified article thoroughly and providing suggestions to improve this work.

1. The 25% TKE correction value is the default in WRF. It does not matter that only one study to date (Archer et al. 2020) has supported it. What matters is that *everybody* is using that value unless they have extensive knowledge of the issues associated with it and change it manually. I disagree that "The 0% added TKE is more similar to the impact in the Volker et al. parameterization which has been used in several studies." Volker et al. used a totally different approach for TKE and therefore I disagree that they should be cited in support of 0% TKE.

   We do not wish to claim that 0% added TKE mirrors the approach used by Volker et al., but that Volker et al. rely on wind speed shear to induce turbulence, which shares similarity in concept to the 0% added TKE approach. Regardless, our reference to Volker was only in response to the reviewer and is not used in the manuscript.

   Other studies have used 0% TKE, but only as sensitivity. One of the first papers by Fitch et al. demonstrated that adding 0% TKE was indeed inaccurate for example. We all know that some TKE is indeed added by the turbines and therefore 0% is not a representative value. 100% is the old default and there is value in using it. But not having anything in between is not good. I wish that the authors had done some runs at 25%, but it's too late for that.

   We agree that performing a more in-depth assessment of the variation by TKE amount would be more thorough, however our computational resources limited us to two simulations only. Fortunately, we include a short-term run using 25% and 50% added TKE in the Appendix, which the readers may use to make comparisons and jump start future research.

   But the sentence in the abstract: "We also vary the amount of added turbulence kinetic energy (TKE) between 0% and 100% to provide some uncertainty quantification" is untrue. You did not "vary" the added TKE between the two extremes, you only used the two extremes. Thus the sentence should be rephrased as "To provide some uncertainty quantification, we tested two values of added TKE: 0% and 100%."

   We agree that no TKE values between 0% and 100% were used for the main findings of this study and have modified the sentence according to the reviewer's suggestion on lines 18-

19: "We also provide a first step towards uncertainty quantification by testing the amount of added turbulence kinetic energy (TKE) by 0% and 100%."

2. I suggest a small change in the order of the equations for the losses, to improve readability (shorter acronyms) and be more consistent (no mix of Loss and LOSS):

$$Loss_{tot} = 100\% - \left(\frac{P_{LA,CA}}{P_{NWF}}\right) \times 100\% \tag{9}$$

$$Loss_{int} = 100\% - \left(\frac{P_{VW}}{P_{NWF}}\right) \times 100\% \tag{10}$$

$$Loss_{ext} = 100\% - \left(\frac{P_{LA,CA}}{P_{VW}}\right) \times 100\% \tag{11}$$

$$Loss_{ext} = Loss_{tot} - Loss_{int} \tag{12}$$

We thank the reviewer for their attention to detail regarding "LOSS" vs. "Loss" and agree that keeping external loss equations next to each other improves readability. We have incorporated the reviewer's suggestions, in new Eqs. 13-16, having replaced the "VW" acronym with "ONE", to easily recognize that one wind plant by itself is used, as can be seen throughout the article:

$$Loss_{tot} = 100\% - \left(\frac{P_{LA,CA}}{P_{NWF}}\right) \times 100\% \tag{13}$$

$$Loss_{int} = 100\% - \left(\frac{P_{ONE}}{P_{NWF}}\right) \times 100\% \tag{14}$$

$$Loss_{ext} = 100\% - \left(\frac{P_{LA,CA}}{P_{ONE}}\right) \times 100\% \tag{15}$$

$$Loss_{ext} = Loss_{tot} - Loss_{int} \tag{16}$$

3. I disagree that L=1000 m is a good choice here. The authors provide one reference only for it, Munoz-Esparza et al. (2012), which I am not familiar with. I provided 5 or 6 references for shorter values and recommended 500 m, which is in the ballpark of all of them. Dismissing them because not all of them are offshore is not convincing; not to mention that at least one, Archer et al. (2016), was offshore, more recent than Munoz- Esparza et al. (2012), and obtained from measurements in the Nantucket Sound, which is in the area of interest here. Lastly, using L=500 m the authors obtained a more typical frequency of neutral cases (11.2% versus the previous low value of 4.5%). As such, I have to insist that the calculations be modified using L=500 m.

We thank the reviewer for their suggestion, as our recalculated validation metrics using L=500 m show improvement across most statistics. As such, we have redone all calculations that rely on stratification using the L=500 m threshold.

References

Beiter, P., W. Musial, P. Duffy, A. Cooperman, M. Shields, D. Heimiller, and M. Optis, 2020: The Cost of Floating Offshore Wind Energy in California Between 2019 and 2032. *Renewable Energy*, 113, https://doi.org/10.2172/1710181.

Bodini, N., and Coauthors, 2023: The 2023 National Offshore Wind data set (NOW-23). *Earth System Science Data Discussions*, 1–57, https://doi.org/10.5194/essd-2023-490.

Perkins, S. E., A. J. Pitman, N. J. Holbrook, and J. McAneney, 2007: Evaluation of the AR4 Climate Models' Simulated Daily Maximum Temperature, Minimum Temperature, and Precipitation over Australia Using Probability Density Functions. *Journal of Climate*, **20**, 4356–4376, https://doi.org/10.1175/JCLI4253.1.

Price-Whelan, A. M., and Coauthors, 2022: The Astropy Project: Sustaining and Growing a Community-oriented Open-source Project and the Latest Major Release (v5.0) of the Core Package*. *ApJ*, **935**, 167, https://doi.org/10.3847/1538-4357/ac7c74.

Pronk, V., N. Bodini, M. Optis, J. K. Lundquist, P. Moriarty, C. Draxl, A. Purkayastha, and E. Young, 2022: Can reanalysis products outperform mesoscale numerical weather prediction models in modeling the wind resource in simple terrain? *Wind Energ. Sci.*, **7**, 487–504, https://doi.org/10.5194/wes-7-487-2022.

Redfern, S., M. Optis, G. Xia, and C. Draxl, 2023: Offshore wind energy forecasting sensitivity to sea surface temperature input in the Mid-Atlantic. *Wind Energy Science*, **8**, 1–23, https://doi.org/10.5194/wes-8-1-2023.

Virtanen, P., and Coauthors, 2020: SciPy 1.0: fundamental algorithms for scientific computing in Python. *Nat Methods*, **17**, 261–272, https://doi.org/10.1038/s41592-019-0686-2.

Xia, G., C. Draxl, M. Optis, and S. Redfern, 2022: Detecting and characterizing simulated sea breezes over the US northeastern coast with implications for offshore wind energy. *Wind Energy Science*, **7**, 815–829, https://doi.org/10.5194/wes-7-815-2022.

---

## Author Response (AR3)

Editor suggestions appear in black and author responses appear in blue.

I have reviewed the revised version of your manuscript, and I am happy to see that you have implemented most of the two reviewer suggestions. Congratulations on a job well done!
However, I have some further suggestions to help improve the readability of your article and correct some technical problems:

     We thank the editor for their attention to detail and for providing suggestions to enhance the readability of this article.

1. Caption Fig. 3: The location of E06 is shown as the red diamond and E05 as the red triangle.

     We have added "The location of" to the figure caption.

2. L160: Should be 'TKE advection is turned on.'

     We have substituted the acronym "TKE" for "turbulence" in this sentence.

3. You use r^2 in Figure 6 but defined CC in Equation 6. I would use r^2 (or /rho) for correlation, as is commonly used.

     We now define "r" in Equation 6 instead of "CC". We have also made this substitution in the text on lines 204 and 239.

4. You use a big V for wind in some places (Eqs 6-8) and then a little v in the shear. It would be best to be consistent.

     We have replaced a lowercase v with an uppercase V in the shear exponent equation and in the text on lines 330-331.

5. L190: "production of turbulence (Eq. 5): " remove Eq. 5. A number is already on the equation. Also, in Eq 6-8 and 9. and so on.

     We now reference the equations at appropriate locations in the text on lines 255-257, 295, 337, 360, and 528.

6. I think long underscores in equations are very ugly and difficult to read. Why not use "O" (or "L") for observations (lidar) and "M" ("W") for modeled (or WRF-simulated)? The same applies to TKE (usually k in equations).

     We have substituted "W" for WRF and "L" for lidar in Eqs. 6-9 and 12.

7. Many of the figure captions are incomplete:
1. Captions of Figures 5,6, 7, 8, and 9: "at the E05 (blue) and E06 (red) lidar locations.'

     "locations" has been added after "lidar" in Figures 5, 6, 7, and 8.

2. Time periods are missing in Figs 5, 6

     The time period under consideration has been added to Figures 5, 6, 7, and 8.

3. It would be easier to compare lidar and WRF simulated shear if the curves were in the same graphic.

     WRF and lidar shear exponents are now shown in one figure panel for ease of comparison.

8. Again, in Eqs 13-16, could you shorten the names and underscores? Could the equations be condensed into a single one?

Thank you for the suggestion. However, considering that the calculation of this power always generates a lot of questions in presentations and several questions from reviewers, we think that explicitness is more critical than concision here. Therefore, we have chosen to leave equations 13-16 as is: it must be extremely clear to readers exactly how power is calculated. Combining the equations (and the two different methods for power calculation) could lead to confusion.

9. Figure 17: The difference in power production between TKE_100 and TKE_0 is shown in MW

We have changed "megawatts" to "MW" in the figure caption.

10. As a final recommendation, please follow the WES style guidelines at https://www.wind-energy-science.net/submission.html. Some of your journal abbreviations need to be corrected (this will save you time later in the article production).

We have corrected journal abbreviations in the references using the Web of Sciences journal title abbreviations.